# ROBUST FEATURE LEARNING FOR MULTI-INDEX MODELS IN HIGH DIMENSIONS

**Alireza Mousavi-Hosseini[1,2], Adel Javanmard[3], Murat A. Erdogdu[1,2]**
[1]University of Toronto, [2]Vector Insitute, [3]University of Southern California
`{mousavi,erdogdu}@cs.toronto,edu, ajavanma@usc.edu`

## ABSTRACT

Recently, there have been numerous studies on feature learning with neural networks, specifically on learning single- and multi-index models where the target is a function of a low-dimensional projection of the input. Prior works have shown that in high dimensions, the majority of the compute and data resources are spent on recovering the low-dimensional projection; once this subspace is recovered, the remainder of the target can be learned independently of the ambient dimension. However, implications of feature learning in adversarial settings remain unexplored. In this work, we take the first steps towards understanding adversarially robust feature learning with neural networks. Specifically, we prove that the hidden directions of a multi-index model offer a Bayes optimal low-dimensional projection for robustness against $\ell_2$-bounded adversarial perturbations under the squared loss, assuming that the multi-index coordinates are statistically independent from the rest of the coordinates. Therefore, robust learning can be achieved by first performing standard feature learning, then robustly tuning a linear readout layer on top of the standard representations. In particular, we show that adversarially robust learning is just as easy as standard learning. Specifically, the additional number of samples needed to robustly learn multi-index models when compared to standard learning, does not depend on dimensionality.

## 1 INTRODUCTION

A crucial capability of neural networks is their ability to hierarchically learn useful features, and to avoid the curse of dimensionality by *adapting* to potential low-dimensional structures in data through empirical risk minimization (ERM) (Bach, 2017; Schmidt-Hieber, 2020). Recently, a theoretical line of work has demonstrated that gradient-based training, which is not a priori guaranteed to implement ERM due to non-convexity, also demonstrates similar behavior and efficiently learns functions of low-dimensional projections (Wei et al., 2019; Damian et al., 2022; Bietti et al., 2022; Barak et al., 2022; Ba et al., 2022; Mousavi-Hosseini et al., 2023a) or functions with certain hierarchical properties (Abbe et al., 2022; 2023; Dandi et al., 2023). These theoretical insights provided a useful avenue for explaining standard feature learning mechanisms in neural networks.

On the other hand, it has been empirically observed that deep neural networks trained with respect to standard losses are susceptible to adversarial attacks; small perturbations in the input may not be detectable by humans, yet they can significantly alter the prediction performed by the model (Szegedy et al., 2014). To overcome this issue, a popular approach is to instead minimize the adversarially robust empirical risk (Madry et al., 2018). However, unlike its standard counterpart, achieving successful generalization of deep neural networks on robust test risk has been particularly challenging, and even the standard performance of the model can degrade once adversarial training is performed (Tsipras et al., 2018). Therefore, one may wonder if robust neural networks are still adaptive to certain problem structures that improve generalization. By focusing on hidden low-dimensionality as a well-known example of such structure, we aim at answering the following fundamental question:

> *Can neural networks maintain their statistical adaptivity to low dimensions*
> *when trained for robustness against adversarial perturbations?*

We answer this question positively by providing the following contributions.

- When considering $\ell_2$-constrained perturbations, Bayes optimal predictors can be constructed by projecting the input data onto the low-dimensional subspace defined by the target function. In this sense, the optimal low-dimensional projection remains unchanged compared to standard learning.

- Consequently, provided that they have access to an oracle that is able to recover the low-dimensional target subspace, neural networks can achieve a sample complexity that is *independent of the ambient dimension* when robustly learning multi-index models. This is achieved by minimizing the empirical adversarial risk with respect to the second layer. While the basic definition of empirical adversarial risk implies computational complexity dependent on input dimension, by simply projecting the inputs onto the low-dimensional target subspace, the computational complexity can also be made independent of the input dimension.

- An oracle for recovering the low-dimensional target subspace can be constructed by training the first layer of a two-layer neural network with a standard loss function, as demonstrated by many prior works. By combining our results with two particular choices of oracle implementation (Damian et al., 2022; Lee et al., 2024), we provide end-to-end guarantees for robustly learning multi-index models with gradient-based algorithms.

## 1.1 RELATED WORKS

**Feature Learning for Single/Multi-Index Models.** Many recent works have focused on proving benefits of feature learning, allowing the neural network weights to travel far from initialization, as opposed to freezing weights around initialization in lazy training (Chizat et al., 2019) which is equivalent to using the Neural Tangent Kernel (Jacot et al., 2018). When using online SGD on the squared loss, Ben Arous et al. (2021) showed that the complexity of learning single-index models with known link function depends on a quantity called information exponent. Gradient-based learning of single-index models with information exponent 1 was studied in Ba et al. (2022); Mousavi-Hosseini et al. (2023a), and Damian et al. (2022) considered multi-index polynomials where the equivalent of information exponent is at most 2. For general information exponent, Bietti et al. (2022) provided an algorithm for gradient-based learning using two-layer neural networks. A feature learning analysis faithful to SGD without modifications was presented in Glasgow (2024) for learning the XOR. The counterpart of information exponent for multi-index models, the leap exponent, was introduced in Abbe et al. (2023). Considering SGD on the squared loss as an example of a Correlational Statistical Query (CSQ) algorithm, Damian et al. (2023) provided CSQ-optimal algorithms for learning single-index models. Further improvements to the isotropic sample complexity were achieved by either considering structured anisotropic Gaussian data (Ba et al., 2023; Mousavi-Hosseini et al., 2023b), or the sparsity of the hidden direction (Vural & Erdogdu, 2024). The benefits of feature learning have also been considered for multitask learning (Collins et al., 2024) and in networks with depth larger than 2 (Nichani et al., 2023; Wang et al., 2024b). More recently, it was observed that gradient-based learning can go beyond CSQ algorithms by reusing batches (Dandi et al., 2024; Lee et al., 2024; Arnaboldi et al., 2024), or by changing the loss function (Joshi et al., 2024). In such cases, the algorithm becomes an instance of a Statistical Query (SQ) learner, and the sample complexity is characterized by the generative exponent of the link function (Damian et al., 2024).

While the above works mostly exist in a narrow-width setting where the interaction between neurons is largely ignored, another line of research focused on the mean-field or wide limits of two-layer neural networks (Chizat & Bach, 2018; Rotskoff & Vanden-Eijnden, 2018; Mei et al., 2018) for providing learnability guarantees (Wei et al., 2019; Chizat & Bach, 2020; Abbe et al., 2022; Telgarsky, 2023; Mahankali et al., 2023; Chen & Ge, 2024). In particular, the mean-field Langevin algorithm provides global convergence guarantees for two-layer neural networks (Chizat, 2022; Nitanda et al., 2022), leading to sample complexity linear in an effective dimension for learning sparse parities (Suzuki et al., 2023; Nitanda et al., 2024) and multi-index models (Mousavi-Hosseini et al., 2024).

**Adversarially Robust Learning.** The existence of small worst-case or adversarial perturbations that can significantly change the prediction of deep neural networks was first demonstrated in Szegedy et al. (2014). Among many defences proposed, one effective approach is adversarial training introduced by Madry et al. (2018), which is based on solving a min-max problem to perform robust optimization. One observation regarding this algorithm is that it tends to decrease the standard performance of the model (Tsipras et al., 2018). Therefore, the following works studied the hardness of robust learning and established a statistical separation in a simple mixture of Gaussians setting (Schmidt et al., 2018), or computational separation by proving statistical query lower bounds (Bubeck et al.,

2019). Further studies focused on exact characterizations of the robust and standard error, as well as the fundamental and the algorithmic tradeoffs between robustness and accuracy in the context of linear regression (Javanmard et al., 2020), mixture of Gaussians classification (Javanmard & Soltanolkotabi, 2022), and in the random features model (Hassani & Javanmard, 2024). Closer to our work, Javanmard & Mehrabi (2024) show that this tradeoff is mitigated when the data enjoy a low-dimensional structure. However, they focus on the population adversarial risk in binary classification and generalized linear models, where the features live on a low-dimensional manifold with known structure. Here, we consider a multi-index model wherein the response depends on a low-dimensional projection of inputs, and derive finite-sample bounds for adversarial risk.

In this work, we provide an alternative narrative compared to the line of work above by showing that in a high-dimensional regression setting, learning multi-index models that are robust against $\ell_2$ perturbations can be as easy as standard learning. We achieve this result by focusing on the feature learning capability of neural networks, i.e. their ability to capture low-dimensional projections.

**Notation.** For Euclidean vectors, $\langle \cdot, \cdot \rangle$ and $\|\cdot\|$ denote the Euclidean inner product and norm respectively. For tensors, $\|\cdot\|_{\mathrm{F}}$ and $\|\cdot\|$ denote the Frobenius and operator norms respectively. We use $\mathbb{S}^{k-1}$ for the unit sphere in $\mathbb{R}^k$, and $\tau_k$ denotes the uniform probability measure on $\mathbb{S}^{k-1}$. For quantities $a$ and $b$, $a = \mathcal{O}(b)$ means there is an absolute constant $C$ such that $a \leq Cb$, and $\Omega$ is similarly defined. $\tilde{\mathcal{O}}$ and $\tilde{\Omega}$ allow $C$ to grow polylogarithmically with problem parameters.

## 2 PROBLEM SETUP: FEATURE LEARNING AND ADVERSARIAL ROBUSTNESS

**Statistical Model.** Consider a regression setting where the input $\boldsymbol{x} \in \mathbb{R}^d$ and the target $y \in \mathbb{R}$ are generated from a distribution $(\boldsymbol{x}, y) \sim \mathcal{P}$. For a prediction function $f : \mathbb{R}^d \to \mathbb{R}$, its population adversarial risk, where we assume the adversary can perform a worst-case perturbation on the input with a budget of $\varepsilon$ measured in $\ell_2$-norm, before passing it to the model, is defined as

$$\mathrm{AR}(f) := \mathbb{E}\left[\max_{\|\boldsymbol{\delta}\| \leq \varepsilon} (f(\boldsymbol{x} + \boldsymbol{\delta}) - y)^2\right], \tag{2.1}$$

where the expectation is over all random variables inside the brackets. Given a (non-parametric) family of prediction functions $\mathcal{F}$, our goal is to learn a predictor that achieves the optimal adversarial risk given by

$$\mathrm{AR}^* := \min_{f \in \mathcal{F}} \mathrm{AR}(f), \tag{2.2}$$

We focus on learners of the form of two-layer neural networks with width $N$, given as

$$f(\boldsymbol{x}; \boldsymbol{a}, \boldsymbol{W}, \boldsymbol{b}) = \boldsymbol{a}^\top \sigma(\boldsymbol{W}\boldsymbol{x} + \boldsymbol{b}), \tag{2.3}$$

where $\boldsymbol{a} \in \mathbb{R}^N$ is the second layer weights and $\boldsymbol{W} \in \mathbb{R}^{N \times d}$ and $\boldsymbol{b} \in \mathbb{R}^N$ are the first layer weights and biases. To avoid overloading the notation we use $\mathrm{AR}(f(\cdot; \boldsymbol{a}, \boldsymbol{W}, \boldsymbol{b})) = \mathrm{AR}(\boldsymbol{a}, \boldsymbol{W}, \boldsymbol{b})$. Given access to $n$ i.i.d. samples $\{\boldsymbol{x}^{(i)}, y^{(i)}\}_{i=1}^n$ from $\mathcal{P}$, the goal is to learn the network parameters $\boldsymbol{a}, \boldsymbol{W}$, and $\boldsymbol{b}$ in such a way that the quantity $\mathrm{AR}(\boldsymbol{a}, \boldsymbol{W}, \boldsymbol{b})$ is close to the optimal adversarial risk $\mathrm{AR}^*$.

A long line of recent works has shown that neural networks are particularly efficient in regression tasks when the target is a function of a low-dimensional projection of the input, see e.g. Bach (2017). Throughout the paper, we also make the same assumption that the data follows a *multi-index model*,

$$\mathbb{E}[y \mid \boldsymbol{x}] = g(\langle \boldsymbol{u}_1, \boldsymbol{x} \rangle, \dots, \langle \boldsymbol{u}_k, \boldsymbol{x} \rangle), \tag{2.4}$$

for all $\boldsymbol{x} \in \mathbb{R}^d$, where $g : \mathbb{R}^k \to \mathbb{R}$ is the link function, and we assume $\boldsymbol{u}_1, \dots, \boldsymbol{u}_k$ are orthonormal without loss of generality. Let $\boldsymbol{U} \in \mathbb{R}^{k \times d}$ be an orthonormal matrix whose rows are given by $(\boldsymbol{u}_i)$; we use the shorthand notation $g(\langle \boldsymbol{u}_1, \boldsymbol{x} \rangle, \dots, \langle \boldsymbol{u}_k, \boldsymbol{x} \rangle) := g(\boldsymbol{U}\boldsymbol{x})$. In the special case where $k = 1$, this model reduces to a *single-index model*. In this paper, we consider the setting where $k \ll d$, and in particular $k = \mathcal{O}(1)$.

**Feature Learning.** In the context of training two-layer neural networks when learning multi-index models, feature learning refers to recovering the target directions $\boldsymbol{U}$ via the first layer weights $\boldsymbol{W}$. Successful feature learning reduces the effective dimension of the problem from the input dimension $d$ to the number of target directions $k$, and circumvents the curse of dimensionality when $k \ll d$.

The complexity of recovering $\boldsymbol{U}$ depends on multiple factors such as the choice of algorithm as well as the properties of the link function. We will provide an overview of some existing results for recovering $\boldsymbol{U}$ with neural networks in Section 4.2, along with several concrete examples.

## 3 OPTIMAL REPRESENTATIONS FOR ROBUST LEARNING

In this section, we demonstrate that under $\ell_2$-constrained perturbations, the optimal low-dimensional representations for robust learning coincides with those in standard setting, both of which are given by the target directions $\boldsymbol{U}$. Crucially, our result relies on the following assumption on input distribution.

**Assumption 1.** *Suppose $\tilde{\boldsymbol{U}} \in \mathbb{R}^{(d-k) \times d}$ is any orthonormal matrix whose rows complete the rows of $\boldsymbol{U}$ into a basis of $\mathbb{R}^d$. Then, $y$ and $\boldsymbol{U}\boldsymbol{x}$ are jointly independent from $\tilde{\boldsymbol{U}}\boldsymbol{x}$.*

The above assumption is quite general. For example, with the notation $\boldsymbol{x}_{\parallel} := \boldsymbol{U}\boldsymbol{x}$ and $\boldsymbol{x}_{\perp} := \tilde{\boldsymbol{U}}\boldsymbol{x}$, it holds when $y = g(\boldsymbol{x}_{\parallel}) + \varsigma$ where $\varsigma$ is independent zero-mean noise, and $\boldsymbol{x} = \boldsymbol{U}^{\top}\boldsymbol{U}\boldsymbol{z}_1 + \tilde{\boldsymbol{U}}^{\top}\tilde{\boldsymbol{U}}\boldsymbol{z}_2$ for independent vectors $\boldsymbol{z}_1, \boldsymbol{z}_2 \in \mathbb{R}^d$. We now present a central result below along with its proof. We discuss the necessity of Assumption 1 and $\ell_2$ constrained attacks to obtain this result in Appendix B.1.

**Theorem 1.** *Suppose Assumption 1 holds and (2.2) admits a minimizer. Then, there exists a function $f^* : \mathbb{R}^d \to \mathbb{R}$ of the form $f^*(\boldsymbol{x}) = h(\boldsymbol{U}\boldsymbol{x})$ with $h : \mathbb{R}^k \to \mathbb{R}$ given by $h(\boldsymbol{z}) = \mathbb{E}[f(\boldsymbol{x}) \,|\, \boldsymbol{U}\boldsymbol{x} = \boldsymbol{z}]$ for some $f \in \mathcal{F}$, such that*

$$.\mathrm{AR}(f^*) \leq \mathrm{AR}^*, \tag{3.1}$$

*with equality when $f^* \in \mathcal{F}$.*

Consequently, to achieve $\mathrm{AR}^*$, one only needs to $(i)$ learn the target directions $\boldsymbol{U}$, and $(ii)$ approximate functions in a $k$-dimensional subspace rather than $d$. For two-layer neural networks, the first layer $\boldsymbol{W}$ recovers $\boldsymbol{U}$, and the remaining parameters $\boldsymbol{a}$ and $\boldsymbol{b}$ are used to approximate the optimal $h$. While this recipe is general, we provide specific implications in the next section.

**Proof.** We will show that for every $f \in \mathcal{F}$, $h(\boldsymbol{z}) = \mathbb{E}[f(\boldsymbol{x}) \,|\, \boldsymbol{U}\boldsymbol{x} = \boldsymbol{z}]$ gives $\mathrm{AR}(h(\boldsymbol{U}\cdot)) \leq \mathrm{AR}(f)$. Then, choosing $f$ to be some minimizer of $\mathrm{AR}$ yields the desired result.

Define the residuals $r_y(\boldsymbol{x}_{\parallel}, \boldsymbol{\delta}_{\parallel}) := y - h(\boldsymbol{x}_{\parallel} + \boldsymbol{\delta}_{\parallel})$, and $r_f(\boldsymbol{x}, \boldsymbol{\delta}) := f(\boldsymbol{x} + \boldsymbol{\delta}) - h(\boldsymbol{x}_{\parallel} + \boldsymbol{\delta}_{\parallel})$. Then, by a decomposition of the squared loss and the tower property of conditional expectation,

$$\mathrm{AR}(f) = \mathbb{E}_{(\boldsymbol{x}_{\parallel}, y)}\left[ \mathbb{E}_{\boldsymbol{x}_{\perp}}\left[ \max_{\|\boldsymbol{\delta}\| \leq \varepsilon} r_y(\boldsymbol{x}_{\parallel}, \boldsymbol{\delta}_{\parallel})^2 + r_f(\boldsymbol{x}, \boldsymbol{\delta})^2 - 2r_y(\boldsymbol{x}_{\parallel}, \boldsymbol{\delta}_{\parallel})r_f(\boldsymbol{x}, \boldsymbol{\delta}) \,\Big|\, \boldsymbol{x}_{\parallel}, y \right] \right]$$

$$\geq \mathbb{E}_{(\boldsymbol{x}_{\parallel}, y)}\left[ \max_{\|\boldsymbol{\delta}\| \leq \varepsilon} r_y(\boldsymbol{x}_{\parallel}, \boldsymbol{\delta}_{\parallel})^2 + \mathbb{E}_{\boldsymbol{x}_{\perp}}\left[ r_f(\boldsymbol{x}, \boldsymbol{\delta})^2 \,\big|\, \boldsymbol{x}_{\parallel}, y \right] - 2r_y(\boldsymbol{x}_{\parallel}, \boldsymbol{\delta}_{\parallel}) \mathbb{E}_{\boldsymbol{x}_{\perp}}\left[ r_f(\boldsymbol{x}, \boldsymbol{\delta}) \,\big|\, \boldsymbol{x}_{\parallel}, y \right] \right]$$

$$\geq \mathbb{E}_{(\boldsymbol{x}_{\parallel}, y)}\left[ \max_{\{\|\boldsymbol{\delta}\| \leq \varepsilon, \boldsymbol{\delta}_{\perp} = 0\}} r_y(\boldsymbol{x}_{\parallel}, \boldsymbol{\delta}_{\parallel})^2 + \mathbb{E}_{\boldsymbol{x}_{\perp}}\left[ r_f(\boldsymbol{x}, \boldsymbol{\delta})^2 \,\big|\, \boldsymbol{x}_{\parallel}, y \right] - 2r_y(\boldsymbol{x}_{\parallel}, \boldsymbol{\delta}_{\parallel}) \mathbb{E}_{\boldsymbol{x}_{\perp}}\left[ r_f(\boldsymbol{x}, \boldsymbol{\delta}) \,\big|\, \boldsymbol{x}_{\parallel}, y \right] \right].$$

Since $y|\boldsymbol{x}_{\parallel}$ is independent from $\boldsymbol{x}_{\perp}$, for any fixed $\boldsymbol{\delta}$, we have $\mathbb{E}\left[ r_f(\boldsymbol{x}, \boldsymbol{\delta}) \,|\, \boldsymbol{x}_{\parallel}, y \right] = \mathbb{E}\left[ r_f(\boldsymbol{x}, \boldsymbol{\delta}) \,|\, \boldsymbol{x}_{\parallel} \right]$. Thus, using the notation $f(\boldsymbol{x}) = f(\boldsymbol{x}_{\parallel}, \boldsymbol{x}_{\perp})$, provided that $\boldsymbol{\delta}_{\perp} = 0$, Assumption 1 yields

$$h(\boldsymbol{z} + \boldsymbol{\delta}_{\parallel}) = \mathbb{E}\left[ f(\boldsymbol{x}) \,|\, \boldsymbol{x}_{\parallel} = \boldsymbol{z} + \boldsymbol{\delta}_{\parallel} \right] = \mathbb{E}\left[ f(\boldsymbol{z} + \boldsymbol{\delta}_{\parallel}, \boldsymbol{x}_{\perp} + \boldsymbol{\delta}_{\perp}) \right] = \mathbb{E}\left[ f(\boldsymbol{x} + \boldsymbol{\delta}) \,|\, \boldsymbol{x}_{\parallel} = \boldsymbol{z} \right],$$

for all $\boldsymbol{z} \in \mathbb{R}^k$. Plugging in $\boldsymbol{z} = \boldsymbol{x}_{\parallel}$ gives $\mathbb{E}\left[ r_f(\boldsymbol{x}, \boldsymbol{\delta}) \,|\, \boldsymbol{x}_{\parallel} \right] = 0$. Therefore,

$$\mathrm{AR}(f) \geq \mathbb{E}_{(\boldsymbol{x}_{\parallel}, y)}\left[ \max_{\{\|\boldsymbol{\delta}\| \leq \varepsilon, \boldsymbol{\delta}_{\perp} = 0\}} r_y(\boldsymbol{x}_{\parallel}, \boldsymbol{\delta}_{\parallel})^2 + \mathbb{E}_{\boldsymbol{x}_{\perp}}\left[ r_f(\boldsymbol{x}, \boldsymbol{\delta})^2 \,|\, \boldsymbol{x}_{\parallel}, y \right] \right]$$

$$\geq \mathbb{E}_{(\boldsymbol{x}_{\parallel}, y)}\left[ \max_{\{\|\boldsymbol{\delta}\| \leq \varepsilon, \boldsymbol{\delta}_{\perp} = 0\}} (y - h(\boldsymbol{U}(\boldsymbol{x} + \boldsymbol{\delta})))^2 \right] = \mathrm{AR}(h(\boldsymbol{U}\cdot)),$$

where we dropped the constraint $\boldsymbol{\delta}_{\perp} = 0$ as it does not contribute, which concludes the proof. $\square$

**A discussion on robust/non-robust feature decomposition.** Many prior works in classification assume that features can be divided into robust and non-robust groups (see e.g. Tsipras et al. (2018); Ilyas et al. (2019); Kim et al. (2021); Li & Li (2024).) We do not rely on this robust/non-robust decomposition. Instead, the $k$ relevant features for predicting $y$ can be either robust or non-robust. The robust training of the second layer ensures the model utilizes the robust subset of these $k$ features, if such a subset exists, while the first layer performs dimensionality reduction.

Before moving to the next section, we provide the following remark on proper scaling of $\varepsilon$. Since $\mathbb{E}[\|\boldsymbol{x}\|]$ grows with $\sqrt{d}$, it may seem natural to scale the adversary budget $\varepsilon$ with dimension as well. However, we provide a simple argument on the contrary. Consider the single-index case $y = g(\langle \boldsymbol{u}, \boldsymbol{x} \rangle)$, and let $h$ be the optimal function constructed in Theorem 1, providing the prediction function $\boldsymbol{x} \mapsto h(\langle \boldsymbol{u}, \boldsymbol{x} \rangle)$. One can then observe that even a constant order $\varepsilon$ is sufficient to incur a large change in the input of $h$, e.g., choosing $\boldsymbol{\delta} = \varepsilon \boldsymbol{u}$ perturbs the input of the predictor by $\varepsilon$. Thus, this justifies the regime where $\varepsilon$ is of constant order compared to the input dimension, which is the focus in the rest of the paper.

## 4 LEARNING PROCEDURE AND GUARANTEES

As outlined in the previous section, to robustly learn the target model, standard representations $\boldsymbol{U}$ suffice. In this section, we consider concrete examples of how a standard feature learning oracle combined with an adversarially robust second layer training leads to robust learning. We assume access to the following *feature learning oracle* to recover $\boldsymbol{U}$. We will provide instances of practical implementations of this oracle using standard gradient-based algorithms in Section 4.2.

**Definition 2** (DFL). *An $\alpha$-Deterministic Feature Learner (DFL) is an oracle that for every $\zeta > 0$, given $n_{\mathrm{DFL}}(\zeta)$ samples from $\mathcal{P}$, returns a weight matrix $\boldsymbol{W} = (\boldsymbol{w}_1, \ldots, \boldsymbol{w}_N)^\top \in \mathbb{R}^{N \times d}$ with unit-norm rows, such that for all $\boldsymbol{u} \in \mathrm{span}(\boldsymbol{u}_1, \ldots, \boldsymbol{u}_k)$ with $\|\boldsymbol{u}\| = 1$, for some $\alpha > 0$ we have*

$$\frac{|\{i : \langle \boldsymbol{w}_i, \boldsymbol{u} \rangle \geq 1 - \zeta\}|}{N} \geq \alpha \zeta^{(k-1)/2}.$$

An $\alpha$-DFL oracle returns weights such that roughly an $\alpha$-proportion of them align with (and sufficiently cover) the target subspace. By a packing argument, we can show that the best achievable ratio is $\alpha \leq c(k)$ for some constant $c(k) > 0$ depending only on $k$, which is why we use the normalizing factor $\zeta^{(k-1)/2}$ above. We show in Section 4.2 that the definition above with a constant order $\alpha$ is attainable by standard gradient-based algorithms. That said, in the multi-index setting, it is possible to improve our learning guarantees by considering the following stochastic oracle.

**Definition 3** (SFL). *An $(\alpha,\beta)$-Stochastic Feature Learner (SFL) is an oracle that for every $\zeta > 0$, given $n_{\mathrm{SFL}}(\zeta)$ samples from $\mathcal{P}$, returns a random weight matrix $\boldsymbol{W} = (\boldsymbol{w}_1, \ldots, \boldsymbol{w}_N)^\top \in \mathbb{R}^{N \times d}$ with unit-norm rows, such that there exists $S \subseteq [N]$ with $|S|/N \geq \alpha$ satisfying $\|\boldsymbol{w}_i - \boldsymbol{U}^\top \boldsymbol{U} \boldsymbol{w}_i\|^2 \leq \zeta$ for $i \in S$. Further, $\left(\frac{\boldsymbol{U} \boldsymbol{w}_i}{\|\boldsymbol{U} \boldsymbol{w}_i\|}\right)_{i \in S} \overset{\mathrm{i.i.d.}}{\sim} \mu$, and $\frac{\mathrm{d}\mu}{\mathrm{d}\tau_k} \geq \beta$ for some $\beta > 0$, where $\mu$ is some measure and $\tau_k$ is uniform, both supported on $\mathbb{S}^{k-1}$.*

The above oracle essentially defines a random features model in the smaller target subspace, where a subset of the weights are sampled independently from a distribution that supports all target directions $\boldsymbol{U}$. The lower bound on $\frac{\mathrm{d}\mu}{\mathrm{d}\tau_k}$ ensures sufficient coverage of the low-dimensional space of target directions. We note that an $(\alpha, \beta)$-SFL oracle can be used to directly implement an $\alpha$-DFL oracle; by a standard union bound argument, one can show $N = \tilde{\Theta}(1/(\alpha\beta\zeta^{(k-1)/2}))$ guarantees the output of $(\alpha, \beta)$-SFL satisfies Definition 2 with high probability. Therefore, while its definition is slightly more involved, $(\alpha, \beta)$-SFL is a more specialized oracle compared to $\alpha$-DFL.

Once the first layer representation is provided by above oracles, we can fix the biases at some random initialization, and train the second layer weights $\boldsymbol{a}$ by minimizing the empirical adversarial risk

$$\widehat{\mathrm{AR}}(\boldsymbol{a}, \boldsymbol{W}, \boldsymbol{b}) = \frac{1}{n_{\mathrm{FA}}} \sum_{i=1}^{n_{\mathrm{FA}}} \max_{\|\boldsymbol{\delta}^{(i)}\| \leq \varepsilon} (f(\boldsymbol{x}^{(i)} + \boldsymbol{\delta}^{(i)}; \boldsymbol{a}, \boldsymbol{W}, \boldsymbol{b}) - y^{(i)})^2. \tag{4.1}$$

We formalize the training procedure with two-layer neural networks in Algorithm 1. We highlight that keeping biases at random initialization while only training the second layer $\boldsymbol{a}$ performs non-linear function approximation, and has been used in many prior works on feature learning (Damian et al., 2022; Mousavi-Hosseini et al., 2023b; Oko et al., 2024). Further, while $\boldsymbol{a} \mapsto \widehat{\mathrm{AR}}(\boldsymbol{a}, \boldsymbol{W}, \boldsymbol{b})$ is convex for fixed $\boldsymbol{W}$ and $\boldsymbol{b}$ since it is a maximum over convex functions, exact training of $\boldsymbol{a}$ in practice may not be straightforward since the inner maximization is not concave and does not admit a closed-form solution. In practice, some form of gradient descent ascent algorithm is typically used when training $\boldsymbol{a}$ (Madry et al., 2018). We leave the study of the computational aspect of that part to future work.

We will make the following standard tail assumptions on the data distribution.

---

**Algorithm 1** Adversarially robust learning with two-layer NNs.

---

**Input:** $\zeta, r_a, r_b, \{\boldsymbol{x}^{(i)}, y^{(i)}\}_{i=1}^{n_{\mathrm{FL}}(\zeta)+n_{\mathrm{FA}}}$, $\mathrm{FL} \in \{\alpha\text{-DFL}, (\alpha,\beta)\text{-SFL}\}$.

1: **Phase 1: Feature Learning**

2:     $\boldsymbol{W} = \mathrm{FL}\Big(\zeta, \{\boldsymbol{x}^{(i)}, y^{(i)}\}_{i=n_{\mathrm{FA}}+1}^{n_{\mathrm{FA}}+n_{\mathrm{FL}}(\zeta)}\Big).$

3: **Phase 2: Robust Function Approximation**

4:     $b_j \overset{iid}{\sim} \mathrm{Unif}(-r_b, r_b)$ for $1 \leq j \leq N$.

5:     $\hat{\boldsymbol{a}} = \arg\min_{\|\boldsymbol{a}\| \leq \frac{r_a}{\sqrt{N}}} \widehat{\mathrm{AR}}(\boldsymbol{a}, \boldsymbol{W}, \boldsymbol{b}).$

6: **return** $(\boldsymbol{a}, \boldsymbol{W}, \boldsymbol{b})$

---

**Assumption 2.** *Suppose $\boldsymbol{x}$ has zero mean and $\mathcal{O}(1)$ subGaussian norm. Furthermore, for all $r \geq 1$, it holds that $\mathbb{E}[|y|^r]^{1/r} \leq \mathcal{O}(r^{p/2})$ for some constant $p \geq 1$.*

Note that the condition on $y$ above is mild; for example, it holds for a noisy multi-index model $y = g(\boldsymbol{U}\boldsymbol{x}) + \varsigma$, where $\varsigma$ has $\mathcal{O}(1)$ subGaussian norm and $g$ grows at most polynomially, i.e., $|g(\cdot)| \lesssim 1 + |\cdot|^p$. Similarly, we also keep the function class $\mathcal{F}$ quite general and provide our first set of results for a class of pseudo-Lipschitz functions which is introduced below.

**Assumption 3.** *We assume $\mathcal{F}$ is a class of functions that are pseudo-Lipschitz along the target coordinates. Specifically, using the notation $f(\boldsymbol{x}) = f(\boldsymbol{x}_\|, \boldsymbol{x}_\perp)$ and defining $\varepsilon_1 := 1 \vee \varepsilon$, we have*

$$|f(\boldsymbol{z}_1, \boldsymbol{x}_\perp) - f(\boldsymbol{z}_2, \boldsymbol{x}_\perp)| \leq L(\boldsymbol{x}_\perp)\big(\varepsilon_1^{1-p}\|\boldsymbol{z}_1\|^{p-1} + \varepsilon_1^{1-p}\|\boldsymbol{z}_2\|^{p-1} + 1\big)\|\boldsymbol{z}_1 - \boldsymbol{z}_2\|$$

*for all $f \in \mathcal{F}$, all $\boldsymbol{z}_1, \boldsymbol{z}_2 \in \mathbb{R}^k$, and some constants $L$ and $p \geq 1$ such that $\mathbb{E}[L(\boldsymbol{x}_\perp)] \leq L$.*

**Remark.** The prefactor $\varepsilon_1^{1-p}$ is justified intuitively since the optimal function of the form $h(\boldsymbol{z}) = \mathbb{E}[f(\boldsymbol{x}) \,|\, \boldsymbol{U}\boldsymbol{x} = \boldsymbol{z}]$ should satisfy $\mathbb{E}\big[\max_{\|\boldsymbol{\delta}\| \leq \varepsilon}(y - h(\boldsymbol{U}(\boldsymbol{x} + \boldsymbol{\delta})))^2\big] = \mathrm{AR}^*$, which is bounded, and does not grow with $\varepsilon$ beyond a certain point. This implies that $h$ must be sufficiently smooth while its input is perturbed, and in particular, its (local) Lipschitz constant should remain bounded while $\varepsilon$ grows, hence the introduction of the prefactor.

The first result of this section assumes access to $\alpha$-DFL oracle.

**Theorem 4.** *Suppose Assumptions 1,2,3 hold and the ReLU activation is used. For a tolerance $\epsilon > 0$ define $\tilde{\epsilon} := \epsilon \wedge (\epsilon^2/\mathrm{AR}^*)$, and for the adversary budget $\varepsilon$ recall $\varepsilon_1 := 1 \vee \varepsilon$. Consider Algorithm 1 with $\mathrm{FL} = \alpha$-DFL oracle, $r_a = \tilde{\mathcal{O}}\big((\varepsilon_1/\sqrt{\tilde{\epsilon}})^{k+1+1/k}/\alpha\big)$ and $r_b = \tilde{\mathcal{O}}\big(\varepsilon_1(\varepsilon_1/\sqrt{\tilde{\epsilon}})^{1+1/k}\big)$. Then, if the number of second phase samples $n_{\mathrm{FA}}$, the number of neurons $N$, and $\alpha$-DFL error $\zeta$ satisfy*

$$n_{\mathrm{FA}} \geq \tilde{\Omega}\left(\frac{\varepsilon_1^4}{\alpha^4 \epsilon^2}\left(\frac{\varepsilon_1}{\sqrt{\tilde{\epsilon}}}\right)^{\mathcal{O}(k)}\right), \quad N \geq \tilde{\Omega}\left(\frac{1}{\alpha\zeta^{(k-1)/2}}\left(\frac{\varepsilon_1}{\sqrt{\tilde{\epsilon}}}\right)^{\mathcal{O}(k)}\right), \quad \zeta \leq \tilde{\mathcal{O}}\left(\left(\frac{\tilde{\epsilon}}{\sqrt{\varepsilon_1}}\right)^{\mathcal{O}(k)}\right),$$

*we have $\mathrm{AR}(\hat{\boldsymbol{a}}, \boldsymbol{W}, \boldsymbol{b}) \leq \mathrm{AR}^* + \epsilon$ with probability at least $1 - n_{\mathrm{FA}}^{-c}$ where $c > 0$ is an absolute constant.*

**Remark.** The total sample complexity of Algorithm 1 is given by the sum of complexities of the two phases, i.e., $n_{\mathrm{total}} = n_{\mathrm{FA}} + n_{\mathrm{DFL}}(\zeta)$. We will provide bounds on $n_{\mathrm{DFL}}(\zeta)$ in Propositions 8 and 10 to ultimately characterize $n_{\mathrm{total}}$ in Corollaries 9 and 11.

The above theorem states that once the feature learning oracle has recovered the target subspace, the number of samples and neurons needed for robust learning is independent of the ambient dimension $d$. Thus, in a high-dimensional setting, statistical complexity is dominated by the feature learning oracle, implying that adversarially robust learning is statistically as easy as standard learning.

Arguing about computational complexity is more involved. While the number of neurons required is independent of $d$, in its naive implementation, Phase 2 of Algorithm 1 needs to solve inner maximization problems over $\mathbb{R}^d$, which may be costly. However, once $\boldsymbol{U} \in \mathbb{R}^{k \times d}$ is estimated in Phase 1, we can reduce the dimension from $d$ to $k$ by projecting onto $\mathrm{span}(\boldsymbol{u}_1, \ldots, \boldsymbol{u}_k)$, i.e.,

$$\sum_{j=1}^{N} a_j \sigma(\langle \boldsymbol{w}_j, \boldsymbol{x} \rangle + b_j) \approx \sum_{j=1}^{N} a_j \sigma(\langle \boldsymbol{U}\boldsymbol{w}_j, \boldsymbol{U}\boldsymbol{x} \rangle + b_j).$$

With this modification, we only need to consider worst-case perturbations over $\mathbb{R}^k$, thus the computational complexity of Phase 2 will also be independent of the ambient dimension $d$.

It is possible to remove the dependence on $\zeta$ in the number of neurons by instead assuming access to an $(\alpha,\beta)$-SFL oracle, as outlined below.

**Theorem 5.** *Consider the same setting as Theorem 4, except that we use the $(\alpha,\beta)$-SFL oracle in Algorithm 1 with $r_a = \tilde{\mathcal{O}}\big((\varepsilon_1/\sqrt{\tilde{\epsilon}})^{k+1+1/k}/(\alpha\beta)\big)$. Then, the sufficient number of second phase samples $n_{\mathrm{FA}}$, neurons $N$, and oracle error $\zeta$ are given as*

$$n_{\mathrm{FA}} \geq \tilde{\Omega}\left(\frac{\varepsilon_1^4}{\alpha^4\beta^4\epsilon^2}\left(\frac{\varepsilon_1}{\sqrt{\tilde{\epsilon}}}\right)^{\mathcal{O}(k)}\right), \quad N \geq \tilde{\Omega}\left(\frac{1}{\alpha\beta^2}\left(\frac{\varepsilon_1}{\sqrt{\tilde{\epsilon}}}\right)^{\mathcal{O}(k)}\right), \quad \zeta \leq \tilde{\mathcal{O}}\left(\beta^2\left(\frac{\tilde{\epsilon}}{\sqrt{\varepsilon_1}}\right)^{\mathcal{O}(k)}\right).$$

Under a Gaussian input assumption, there exist $\alpha$-DFL and $(\alpha,\beta)$-SFL oracles that rely only on standard gradient-based training such that for a small constant $\zeta$, $n_{\mathrm{DFL}}(\zeta)$ and $n_{\mathrm{SFL}}(\zeta)$ both scale with some polynomial of $d$, where the exponent depends on certain properties of the link function, termed as the *information or generative exponent* (Ben Arous et al., 2021; Damian et al., 2024). We will provide explicit examples of such algorithms in Section 4.2 to characterize the total sample complexity $n_{\mathrm{total}} = n_{\mathrm{FA}} + n_{\mathrm{DFL/SFL}}$. For the interested reader, we restate Theorems 4 and 5 in Appendix B.2 with explicit exponents.

## 4.1 COMPETING AGAINST THE OPTIMAL POLYNOMIAL PREDICTOR

In this section, we restrict $\mathcal{F}$ to only polynomials, which allows us to derive more refined bounds on the number of samples and neurons. Specifically, we make the following assumption.

**Assumption 4.** *Suppose $\mathcal{F}$ is the class of d-variate polynomials of degree $p$ for some constant $p > 0$. Further, $\sigma$ is either a polynomial of degree $q \geq p$, or the ReLU activation for which we define $q = (p-1) \vee 1$.*

While the ReLU activation is sufficient for function approximation, we also consider polynomial activations in Assumption 4 since using those, recent works have been able to achieve sharper theoretical guarantees of recovering the target directions (Lee et al., 2024); we provide a more detailed discussion in Section 4.2. Note that a priori we do not require a growth constraint on the coefficients of the polynomials in $\mathcal{F}$. The optimal function $h$ in Theorem 1 automatically chooses a polynomial with suitably bounded coefficients in order to avoid incurring a large robust risk.

The following result establishes the sample and computational complexity for competing against polynomial predictors when having access to $\alpha$-DFL oracle.

**Theorem 6.** *Suppose Assumptions 1,2,4 hold. For a tolerance $\epsilon > 0$ define $\tilde{\epsilon} := \epsilon \wedge (\epsilon^2/\mathrm{AR}^*)$, and for the adversary budget $\varepsilon$ recall $\varepsilon_1 := 1 \vee \varepsilon$. Consider Algorithm 1 with $\alpha$-DFL oracle, $r_a = \tilde{\mathcal{O}}(1)$, $r_b = \tilde{\mathcal{O}}(\varepsilon_1)$. If the number of second phase samples $n_{\mathrm{FA}}$, neurons $N$, and $\alpha$-DFL error $\zeta$ satisfy*

$$n_{\mathrm{FA}} \geq \tilde{\Omega}\left(\frac{\varepsilon_1^{4(q+1)}}{\alpha^4\epsilon^2}\right), \quad N \geq \tilde{\Omega}\left(\frac{\varepsilon_1^{q+1}}{\alpha\zeta^{\frac{k-1}{2}}\sqrt{\tilde{\epsilon}}}\right), \quad \zeta \leq \tilde{\mathcal{O}}\left(\frac{\tilde{\epsilon}}{\varepsilon_1^{2(q+1)}}\right),$$

*we have $\mathrm{AR}(\hat{a}, W, b) \leq \mathrm{AR}^* + \epsilon$ with probability at least $1 - n_{\mathrm{FA}}^{-c}$ where $c > 0$ is an absolute constant.*

Consequently, when restricting $\mathcal{F}$ to the class of fixed degree polynomials, there is no curse of dimensionality for sample complexity, even in the latent dimension $k$. This is consistent with the standard learning setting, see e.g. Chen & Meka (2020). Further, similar to the general case above, it is possible to remove the $\zeta$ dependence from $N$ when having access to an SFL oracle, thus also achieving computational complexity as a fixed polynomial independent of the latent dimension.

**Theorem 7.** *In the setting of Theorem 6, consider using Algorithm 1 with an $(\alpha,\beta)$-SFL oracle. Then, the sufficient number of second phase samples, neurons, and oracle error are given as*

$$n_{\mathrm{FA}} \geq \tilde{\Omega}\left(\frac{\varepsilon_1^{4(q+1)}}{\alpha^4\beta^4\epsilon^2}\right), \quad N \geq \tilde{\Omega}\left(\frac{\varepsilon_1^{2(q+1)}}{\alpha\beta^2\tilde{\epsilon}}\right), \quad \zeta \leq \tilde{\mathcal{O}}\left(\frac{\beta^2\tilde{\epsilon}}{\varepsilon_1^{2(q+1)}}\right).$$

We remark that the guarantees provided in Theorem 7 are generally better than those in Theorem 6 for large $k$; yet, they are strictly worse for $k = 1$. That said, both Theorems 7 and 6 respectively achieve better sample complexity guarantees compared to their counterparts in the previous section, namely Theorems 4 and 5, simply by restricting the function class $\mathcal{F}$ to polynomials.

## 4.2 ORACLE IMPLEMENTATIONS FOR FEATURE LEARNING

The task of recovering the target directions $U$ is classical in statistics, and is known as sufficient dimension reduction (Li & Duan, 1989; Li, 1991), with many dedicated algorithms, see e.g. Kakade et al. (2011); Dudeja & Hsu (2018); Chen & Meka (2020); Yuan et al. (2023) to name a few. Here, we focus on algorithms based on neural networks and iterative gradient-based optimization.

While we only consider the case where $x$ is an isotropic Gaussian random vector, recovering the hidden direction has also been considered for non-isotropic Gaussians (Ba et al., 2023; Mousavi-Hosseini et al., 2023b) where the additional anisotropic structure in the inputs can provide further statistical benefits, or non-Gaussian spherically symmetric distributions (Zweig et al., 2023). Our results readily extend to these settings as well. First, we present the case of single-index polynomials.

**Proposition 8** (Lee et al. (2024)). *Suppose $x \sim \mathcal{N}(0, \mathbf{I}_d)$, $k = 1$, and $g$ is a polynomial of degree $p$ where $p$ is constant. Then, there exists an iterative first-order algorithm on two-layer neural networks (see Algorithm 2) that implements an $(\alpha,\beta)$-SFL oracle and an $\alpha$-DFL oracle, where $\alpha = \tilde{\Theta}(1)$ and $\beta = 1$. Furthermore, we have $n_{\mathrm{SFL}}(\zeta) = n_{\mathrm{DFL}}(\zeta) = \tilde{\mathcal{O}}(d/\zeta^2)$.*

Combined with Theorems 4-7, we obtain the following total sample complexity guarantee for robustly learning Gaussian single-index models.

**Corollary 9.** *Consider the data model of Proposition 8, and assume that the adversary budget is $\varepsilon = \mathcal{O}(1)$. Then, the total sample complexity of Algorithm 1 to achieve optimal adversarial risk $\mathrm{AR}^*$ with a tolerance $\epsilon$ using either $\alpha$-DFL or $(\alpha,\beta)$-SFL oracle in Proposition 8 is given as*

- *$n_{\mathrm{total}} = \tilde{\mathcal{O}}(d/\tilde{\epsilon}^2)$ when choosing $\mathcal{F}$ to be polynomials of fixed degree as in Assumption 4, and the polynomial activation according to Lee et al. (2024),*

- *$n_{\mathrm{total}} = \tilde{\mathcal{O}}(d/\tilde{\epsilon}^{\mathcal{O}(1)})$ when choosing $\mathcal{F}$ to be pseudo-Lipschitz functions as in Assumption 3,*

*where we recall $\tilde{\epsilon} := \epsilon \wedge (\epsilon^2/\mathrm{AR}^*)$.*

When considering Gaussian single-index models beyond polynomials, we must introduce the concepts of *information* and *generative exponent* to characterize the sample complexity of recovering the target direction. Let $\gamma = \mathcal{N}(0, 1)$, and for any $g : \mathbb{R} \to \mathbb{R}$ in $L^2(\gamma)$, let $g = \sum_{j \geq 0} \alpha_j \mathrm{He}_j$ denote its Hermite expansion, where $\mathrm{He}_j$ is the normalized Hermite polynomial of degree $j$. The information exponent of $g$ is defined as $s(g) := \min\{j > 0 : \alpha_j \neq 0\}$. The generative exponent on the other hand, is defined as the minimum information exponent attainable by any transformation of $g$, i.e. $s^*(g) := \min_{\mathcal{T}} s(\mathcal{T}(g))$, where the minimum is over all $\mathcal{T} \in L^2(g \# \gamma)$. Thus, $s^*(g) \leq s(g)$, and in particular, $s^* = 1$ for all polynomials. See Ben Arous et al. (2021); Damian et al. (2024) for details.

There exists an algorithm based on estimating partial traces that implements a 1-DFL (or a $(1, 1)$-SFL) oracle with $n_{\mathrm{DFL}}(\zeta) = \mathcal{O}(d^{s^*/2} + d/\zeta^2)$ (Damian et al., 2024). While it may be possible to achieve a similar sample complexity when training neural networks with a ReLU activation, the state of the art results for ReLU neural networks so far are only able to control the sample complexity with the information exponent $s$, e.g. Bietti et al. (2022) provides a gradient-based algorithm for optimizing a variant of a two-layer ReLU network that implements 1-DFL with $n_{\mathrm{DFL}} = \mathcal{O}(d^s \mathrm{poly}(\zeta^{-1}))$.

Recovering $U$ with $k > 1$ is more challenging, and the general picture is that the directions in $U$ are recovered hierarhically based on each direction's corresponding complexity, such as in Abbe et al. (2023). For simplicity, we look at a case that is sufficiently simple for all directions to be learned simultaneously, while emphasizing that in principle any guarantee for learning the subspace $U$ can be turned into an implementation of the oracles introduced in the previous section.

**Proposition 10** (Damian et al. (2022)). *Suppose $x \sim \mathcal{N}(0, \mathbf{I}_d)$, $g$ is a polynomial of degree $p$, and $p,k$ are constant. Further assume $\frac{\sigma_{\max}(\nabla^2 g)}{\sigma_{\min}(\nabla^2 g)} \geq \kappa$ for some $\kappa > 0$, where $\sigma_{\min}$ and $\sigma_{\max}$ denote the minimum and maximum singular values, respectively. Then, there exists a first-order algorithm on*

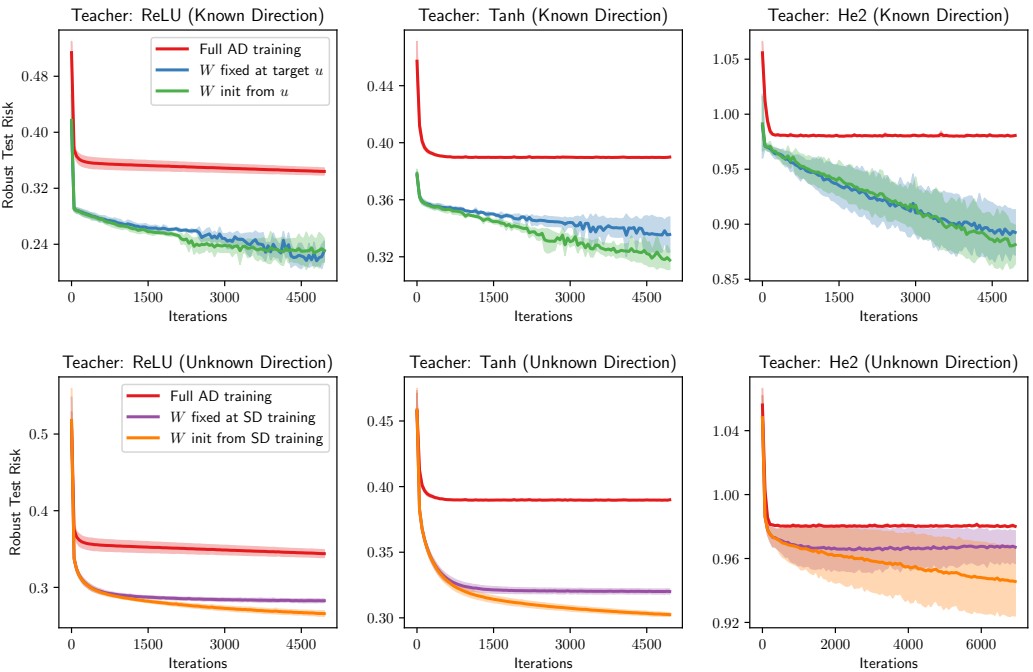

Figure 1: The adversarial test error of a two-layer ReLU network as a function of the number of adversarial training iterations, where each iteration is performed on a batch of independent 300 samples, except 500 samples for He2 with unknown direction to reduce variance. Full AD training performs adversarial training on all layers from random initialization. SD training is standard training, which provides a better initialization for $W$ before performing adversarial training. We use the adversary budget $\varepsilon = 1$ for all experiments, each of which are averaged over three runs.

*two-layer ReLU neural networks* (see Algorithm 3) *that implements an* $(\alpha,\beta)$-SFL *and an* $\alpha$-DFL *oracle, where* $\alpha = 1$ *and* $\beta \geq c_\kappa$ *for some constant* $c_\kappa > 0$ *depending only* $\kappa$*. Further, we have* $n_{\mathrm{SFL}}(\zeta) = n_{\mathrm{DFL}}(\zeta) = \tilde{\mathcal{O}}(d^2 + d/\zeta^2)$.

Combining the above proposition with Theorems 4-7, we obtain the following total sample complexity for robustly learning Gaussian multi-index models.

**Corollary 11.** *Under the data model of Proposition 10, assume that the adversary budget is* $\varepsilon = \mathcal{O}(1)$*. Then, the total sample complexity of Algorithm 1 to achieve optimal adversarial risk* $\mathrm{AR}^*$ *with a tolerance* $\epsilon$ *using either* $\alpha$-DFL *or* $(\alpha,\beta)$-SFL *oracle in Proposition 10 is given as*

- $n_{\mathrm{total}} = \tilde{\mathcal{O}}(d^2 + d/\tilde{\epsilon}^2)$ *when choosing* $\mathcal{F}$ *to be polynomials of fixed degree as in Assumption 4, and the polynomial activation according to Lee et al. (2024),*

- $n_{\mathrm{total}} = \tilde{\mathcal{O}}(d^2 + d/\tilde{\epsilon}^{\mathcal{O}(k)})$ *when choosing* $\mathcal{F}$ *to be pseudo-Lipschitz functions as in Assumption 3,*

*where we recall* $\tilde{\epsilon} := \epsilon \wedge (\epsilon^2/\mathrm{AR}^*)$.

**Remark.** We highlight that the gap in the total sample complexity of Corollary 9 and Corollary 11 is due to more efficient guarantees for recovering the hidden direction for single-index polynomials. It is an open question whether such efficient recovery is also possible for multi-index polynomials.

## 5 NUMERICAL EXPERIMENTS

As a proof of concept, we also provide small-scale numerical studies on Gaussian data to support intuitions derived from our theory. Additional experiments on real datasets are provided in Appendix E.

We consider a single-index setting, where the teacher non-linearity is given by either ReLU, tanh, or $\text{He2}(z) = (z^2 - 1)/\sqrt{2}$ which is the normalized second Hermite polynomial. The student network has $N = 100$ neurons, and the input is sampled from $\boldsymbol{x} \sim \mathcal{N}(0, \mathbf{I}_d)$ with $d = 100$. We implement adversarial training in the following manner. At each iteration, we sample a new batch of i.i.d. training examples. We estimate the adversarial perturbations on this batch by performing 5 steps of signed projected gradient ascent, with a stepsize of 0.1. We then perform a gradient descent step on the perturbed batch. To estimate the robust test risk, we fix a test set of 10,000 i.i.d. samples, and use 20 iterations to estimate the adversarial perturbation. Because of the online nature of the algorithm, the total number of samples used is the batch size times the number of iterations taken.

The first row of Figure 1 compares the performance of three different approaches. Full AD training refers to adversarially training all layers from random initialization, where first layer weights are initialized uniformly on the sphere $\mathbb{S}^{d-1}$, second layer weights are initialized i.i.d. from $\mathcal{N}(0, 1/N^2)$, and biases are initialized i.i.d. from $\mathcal{N}(0, 1)$. In the two other approaches, we initialize all first layer weights to the target direction $\boldsymbol{u}$. In one approach we fix this direction and do not train it, while in the other, we allow the training of first layer weights from this initialization. As can be seen from Figure 1, there is a considerable improvement in initializing from $\boldsymbol{u}$, which is consistent with our theory that this direction provides a Bayes optimal projection for robust learning.

In the practical setting where we do not have the knowledge of $\boldsymbol{u}$, we consider the following alternative. We first perform standard training on the network, i.e. assume $\varepsilon = 0$ (denoted in Figure 1 by SD training). We can then either fix the first layer weights to these directions, or further train them adversarially from this initialization. Note that for a fair comparison with the full AD method, we provide the same random bias and second layer weight initializations across all methods at the beginning of the adversarial training stage. Even though this approach is not perfect at estimating the unknown direction, it still provides a considerable benefit over adversarially training all layers from random initialization, as demonstrated in the second row of Figure 1.

## 6 CONCLUSION

In this paper, we initiated a theoretical study of the role of feature learning in adversarial robustness of neural networks. Under $\ell_2$-constrained perturbations, we proved that projecting onto the latent subspace of a multi-index model is sufficient for achieving Bayes optimal adversarial risk with respect to the squared loss, provided that the index directions are statistically independent from the rest of the directions in the input space. Remarkably, this subspace can be estimated through standard feature learning with neural networks, thus turning a high-dimensional robust learning problem into a low-dimensional one. As a result, under the assumption of having access to a feature learning oracle which returns an estimate of this subspace, and can be implemented e.g. by training the first-layer of a two-layer neural network, we proved that robust learning of multi-index models is possible with a number of (additional) samples and neurons independent from the ambient dimension.

We conclude by mentioning several open questions that arise from this work.

- Stronger notions of adversarial attacks such as $\ell_\infty$-norm constraints have been widely considered in empirical works. It remains open to understand optimal low-dimensional representations under such perturbations as well as their implications on sample complexity.

- While our work demonstrates that standard training is sufficient for the first layer, it is unclear what kind of representation is learned when all layers are trained adversarially. In particular, Figure 1 suggests that adversarial training of the first layer may be suboptimal in this setting, even when infinitely many samples are available during training.

- Since our main motivation was to show independence from input dimension, the dependence of our bounds on the final robust test risk suboptimality $\epsilon$ are potentially improvable by a more careful analysis. It is an interesting direction to obtain a sharper dependency and investigate the optimality of such dependence on the tolerance $\epsilon$.

Finally, it is worth emphasizing that our theorems can be easily adapted to other standard feature learning oracles. As such, based on the training procedure used and its complexity in feature learning, our results are amenable to further improvements in their total sample complexity.

ACKNOWLEDGMENTS

AJ was partially supported by the Sloan fellowship in mathematics, the NSF CAREER Award DMS-1844481, the NSF Award DMS-2311024, an Amazon Faculty Research Award, an Adobe Faculty Research Award and an iORB grant form USC Marshall School of Business. MAE was partially supported by the NSERC Grant [2019-06167], the CIFAR AI Chairs program, and the CIFAR Catalyst grant.

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

# A   GRADIENT-BASED NEURAL FEATURE LEARNING ALGORITHMS

In this section, we will provide examples of implementations of the feature learner oracles introduced in Section 4 using gradient-based training of two-layer neural networks. First, we look at the algorithm provided by Oko et al. (2024), which we restate here as Algorithm 2, for the case where $g$ is a polynomial of degree $p$. Consider the following two-layer neural network with zero bias

$$f(\boldsymbol{x}; \boldsymbol{a}, \boldsymbol{W}) = \sum_{j=1}^{N} a_j \sigma_j(\langle \boldsymbol{w}_j, \boldsymbol{x} \rangle).$$

Note that we allow the activation to vary based on neuron. Specifically, we let $\sigma_j = \sum_{l=1}^{q} \beta_{j,l} \mathrm{He}_l$, where $\mathrm{He}_j$ is the $j$th normalized Hermite polynomial, $\beta_{j,l} \overset{\text{i.i.d.}}{\sim} \mathrm{Unif}(\{\pm r_l\})$ for appropriately chosen $r_l$, and $q \geq C_p$, see Oko et al. (2024, Lemma 3) for details. Now, we consider the following algorithm.

---

**Algorithm 2** Gradient-Based Feature Learner for Single-Index Polynomials (Oko et al., 2024, Algorithm 1, Phase I).

---

**Input:** $T$, step size $(\eta^t)_{t=0}^{T-1}$, momentum parameters $(\zeta_j^t)$, $r_a$.

1: $\boldsymbol{w}_j^0 \overset{\text{i.i.d.}}{\sim} \mathrm{Unif}(\mathbb{S}^{d-1})$, $\quad a_j \overset{\text{i.i.d.}}{\sim} \mathrm{Unif}(\{\pm r_a/N\})$, $\quad \forall j \in [N]$.
2: $(\boldsymbol{x}^{(0)}, y^{(0)}) \sim \mathcal{P}$
3: **for** $t = 0, \dots, T-1$ **do**
4:    **if** $t > 0$ and $t$ is even **then**
5:       Draw $(\boldsymbol{x}^{(t/2)}, y^{(t/2)}) \sim \mathcal{P}$
6:       $\boldsymbol{w}_j^t \leftarrow \boldsymbol{w}_j^t - \zeta_j^t(\boldsymbol{w}_j^t - \boldsymbol{w}_j^{t-2})$, $\quad \forall j \in [N]$
7:       $\boldsymbol{w}_j^t \leftarrow \frac{\boldsymbol{w}_j^t}{\|\boldsymbol{w}_j^t\|}$ $\quad \forall j \in [N]$
8:    **end if**
9:    $\boldsymbol{w}_j^{t+1} \leftarrow \boldsymbol{w}_j^t - \eta_t \nabla_{\boldsymbol{w}_j}^S (f(\boldsymbol{x}^{(\lfloor t/2 \rfloor)}; \boldsymbol{a}, \boldsymbol{W}^t) - y^{(\lfloor t/2 \rfloor)})^2$
10: **end for**
11: **return** $(\boldsymbol{w}_0^T, \dots, \boldsymbol{w}_N^T)^\top$

---

Note that $\nabla^S f(\boldsymbol{w}) = (\boldsymbol{I} - \boldsymbol{w}\boldsymbol{w}^\top)\nabla f(\boldsymbol{w})$ denotes the spherical gradient. Essentially, Algorithm 2 takes two gradient steps on each new sample, and in the even iterations performs a certain interpolation. Proper choice of hyperparameters in the above algorithm leads to Proposition 8.

Next, we consider the algorithm of Damian et al. (2022), which we restate here as Algorithm 3, for the case where $g$ is a multi-index polynomial.

---

**Algorithm 3** Gradient-Based Feature Learner for Multi-Index Polynomials (Damian et al., 2022, Algorithm 1, Adapted)

---

**Input:** $\{\boldsymbol{x}^{(i)}, y^{(i)}\}_{i=1}^{n_{\mathrm{FL}}}$, $r_a$

1: $\boldsymbol{a}_j \overset{\text{i.i.d.}}{\sim} \mathrm{Unif}(\{\pm r_a\})$, $\boldsymbol{w}_j^0 \overset{\text{i.i.d.}}{\sim} \mathrm{Unif}(\mathbb{S}^{d-1})$, $\boldsymbol{a}_{N-j} = -\boldsymbol{a}_j$, $\boldsymbol{w}_{N-j} = \boldsymbol{w}_j^0$, $\quad \forall j \in [N/2]$.
2: $\alpha \leftarrow \frac{1}{n_{\mathrm{FL}}} \sum_{i=1}^{n_{\mathrm{FL}}} y^{(i)}$, $\quad \boldsymbol{\beta} \leftarrow \frac{1}{n_{\mathrm{FL}}} \sum_{i=1}^{n_{\mathrm{FL}}} y^{(i)} \boldsymbol{x}^{(i)}$
3: $y^{(i)} \leftarrow y^{(i)} - \alpha - \langle \boldsymbol{\beta}, \boldsymbol{x}^{(i)} \rangle$, $\quad \forall i \in [n_{\mathrm{FL}}]$.
4: $\boldsymbol{W} \leftarrow -\nabla_{\boldsymbol{W}} \frac{1}{n} \sum_{i=1}^{n_{\mathrm{FL}}} (f(\boldsymbol{x}^{(i)}; \boldsymbol{a}, \boldsymbol{W}^0) - y)^2$
5: $\boldsymbol{w}_i \leftarrow \frac{\boldsymbol{w}_i}{\|\boldsymbol{w}_i\|}$, $\quad \forall i \in [N]$
6: **return** $(\boldsymbol{w}_0, \dots, \boldsymbol{w}_N)^\top$

---

After performing a preprocessing on data, Algorithm 3 essentially performs one gradient descent step with weight decay, when the regularizer of the weight decay is the inverse of step size, thus cancelling out initialization and leaving only gradient as the estimate. Damian et al. (2022) prove that, with a sample complexity of $n_{\mathrm{FL}} = \tilde{\mathcal{O}}(d^2 + d/\zeta^2)$, the output of Algorithm 3 satisfies

$$\left\langle \boldsymbol{w}_i, \frac{\boldsymbol{U}^\top \boldsymbol{H} \boldsymbol{U} \boldsymbol{w}_i^0}{\|\boldsymbol{U}^\top \boldsymbol{H} \boldsymbol{U} \boldsymbol{w}_i^0\|} \right\rangle \geq 1 - \zeta, \quad \forall i \in [N],$$

witi high probability, where $\boldsymbol{H} = \mathbb{E}\big[\nabla^2 g(\boldsymbol{U}\boldsymbol{x})\big]$. Thus, for a full-rank $\boldsymbol{H}$, the output of Algorithm 3 satisfies the definition of a $(1, \beta)$SFL oracle for a constant $\beta > 0$ depending only on the conditioning of $H$ and the number of indices $k$.

## B ADDITIONAL NOTATIONS AND DETAILS OF SECTION 4

Throughout the appendix, we will assume the activation satisfies $\sigma(0) = 0$ for simplicity of presentation, without loss of generality. We will also assume that

$$|\sigma(z_1) - \sigma(z_2)| \le L_\sigma(|z_1|^{\bar{q}-1} + |z_2|^{\bar{q}-1} + 1)|z_1 - z_2|, \tag{B.1}$$

for all $z_1, z_2 \in \mathbb{R}$ and some absolute constant $L_\sigma$. In the case of ReLU, we have $\bar{q} = 1$ and $L_\sigma = 1$. For polynomial activations, $\bar{q}$ is the same as the degree of the polynomial. For a set of parameters $\psi$ (e.g. $\psi = q, k$), we will use $C_\psi$ to denote a generic constant whose value depends only on $\psi$ and may change from line to line.

### B.1 NECESSITY OF $\ell_2$ NORM AND ASSUMPTION 1 FOR THEOREM 1

In this section, we demonstrate that both restricting the attack norm and Assumption 1 are necessary for the statement of Theorem 1 to hold.

First, we focus on violating Assumption 1. Suppose $\boldsymbol{x} \sim \mathcal{N}(0, \boldsymbol{\Sigma})$. Suppose $k = 1$, $y = \langle \boldsymbol{u}_1, \boldsymbol{x} \rangle$, and let $\mathcal{F}$ be the class of linear predictors. Then, the adversarial risk associated to the predictor $\boldsymbol{x} \mapsto \langle \boldsymbol{w}, \boldsymbol{x} \rangle$ is given by

$$\mathrm{AR}(\langle \boldsymbol{w}, \cdot \rangle) = \left\| \boldsymbol{\Sigma}^{1/2}(\boldsymbol{w} - \boldsymbol{u}_1) \right\|^2 + \varepsilon^2 \|\boldsymbol{w}\|^2 + 2\sqrt{\frac{2}{\pi}} \left\| \boldsymbol{\Sigma}^{1/2}(\boldsymbol{w} - \boldsymbol{u}_1) \right\| \|\boldsymbol{w}\|.$$

From here, one can verify that the optimal weight $\boldsymbol{w}^*$ satisfies $\boldsymbol{w}^* = (\boldsymbol{\Sigma} + a\varepsilon\mathbf{I}_d)^{-1}\boldsymbol{\Sigma}\boldsymbol{u}_1$ for some $a > 0$. Note that $\boldsymbol{w}^*$ is only in the direction of $\boldsymbol{u}_1$ if $\boldsymbol{u}_1$ is an eigenvector of $\boldsymbol{\Sigma}$, which would imply $\langle \boldsymbol{u}_1, \boldsymbol{x} \rangle$ is statistically independent from $\langle \boldsymbol{v}, \boldsymbol{x} \rangle$ for every $\boldsymbol{v} \in \mathbb{R}^d$ orthogonal to $\boldsymbol{u}_1$.

Next, we replace the $\ell_2$ constraint for the adversary with an $\ell_\infty$ constraint, i.e. we define

$$\mathrm{AR}_\infty(f) = \mathbb{E}\left[ \max_{\|\boldsymbol{\delta}\|_\infty \le \varepsilon_\infty} (f(\boldsymbol{x} + \boldsymbol{\delta}) - y)^2 \right].$$

Suppose $\boldsymbol{x} \sim \mathcal{N}(0, \mathbf{I}_d)$, and let $y = \langle \boldsymbol{u}_1, \boldsymbol{x} \rangle$ and $\mathcal{F}$ be linear as above. Then, we have

$$\mathrm{AR}(\langle \boldsymbol{w}, \cdot \rangle) = \|\boldsymbol{w} - \boldsymbol{u}_1\|^2 + \varepsilon_\infty \|\boldsymbol{w}\|_1^2 + 2\sqrt{\frac{2}{\pi}} \|\boldsymbol{w} - \boldsymbol{u}_1\| \|\boldsymbol{w}\|_1.$$

Then, assuming all the coordinates of $\boldsymbol{u}_1$ are bounded away from zero and for sufficiently small $\varepsilon_\infty$, one can show $\boldsymbol{w}^* = \boldsymbol{u}_1 - c\varepsilon_\infty \mathrm{sign}(\boldsymbol{u}_1)$ for some $c > 0$, which will no longer necessary be in the direction of $\boldsymbol{u}_1$.

### B.2 COMPLETE VERSIONS OF THEOREMS IN SECTION 4

We first restate Theorem 4 with explicit exponents.

**Theorem 12.** *Suppose Assumptions 1,2, and 3 hold. For any $\epsilon > 0$, define $\tilde{\epsilon} := \epsilon \wedge (\epsilon^2/\mathrm{AR}^*)$, and recall $\varepsilon_1 := 1 \vee \varepsilon$. Consider Algorithm 1 with the $\alpha$-DFL oracle, $r_a = \tilde{\mathcal{O}}\big((\varepsilon_1/\sqrt{\tilde{\epsilon}})^{k+1+1/k}/\alpha\big)$, and $r_b = \tilde{\mathcal{O}}\big(\varepsilon_1(\varepsilon_1/\sqrt{\tilde{\epsilon}})^{1+1/k}\big)$. Then, if the number of second phase samples $n_{\mathrm{FA}}$, number of neurons $N$, and $\alpha$-DFL error $\zeta$ satisfy*

$$n_{\mathrm{FA}} \ge \tilde{\Omega}\Big(\frac{\varepsilon_1^4}{\alpha^4\epsilon^2}\big(\frac{\varepsilon_1^2}{\tilde{\epsilon}}\big)^{2k+4+4/k}\Big), \quad N \ge \tilde{\Omega}\Big(\frac{1}{\alpha\zeta^{(k-1)/2}}\big(\frac{\varepsilon_1}{\sqrt{\tilde{\epsilon}}}\big)^{k+3+2/k}\Big), \quad \zeta \le \tilde{\mathcal{O}}\Big(\big(\frac{\tilde{\epsilon}}{\varepsilon_1^2}\big)^{k+2+1/k}\Big),$$

*we have $\mathrm{AR}(\hat{\boldsymbol{a}}, \boldsymbol{W}, \boldsymbol{b}) \le \mathrm{AR}^* + \epsilon$ with probability at least $1 - n_{\mathrm{FA}}^{-c}$ where $c > 0$ is an absolute constant. The total sample complexity of Algorithm 1 is given by $n_{\mathrm{total}} = n_{\mathrm{FA}} + n_{\mathrm{DFL}}(\zeta)$.*

Similarly, we can restate Theorem 5 with explicit exponents.

**Theorem 13.** *Consider the same setting as Theorem 12, except that we use the $(\alpha,\beta)$-SFL oracle in Algorithm 1 with $r_a = \tilde{\mathcal{O}}\big((\varepsilon_1/\sqrt{\tilde{\epsilon}})^{k+1+1/k}/(\alpha\beta)\big)$. Then, if the number of second phase samples $n_{\mathrm{FA}}$, number of neurons $N$, and $\alpha$-DFL error $\zeta$ satisfy*

$$n_{\mathrm{FA}} \geq \tilde{\Omega}\Big(\frac{\varepsilon_1^4}{\alpha^4\beta^4\epsilon^2}\big(\frac{\varepsilon_1^2}{\tilde{\epsilon}}\big)^{2k+4+4/k}\Big), \quad N \geq \tilde{\Omega}\Big(\frac{1}{\alpha\beta^2}\big(\frac{\varepsilon_1^2}{\tilde{\epsilon}}\big)^{k+3+2/k}\Big), \quad \zeta \leq \tilde{\mathcal{O}}\Big(\beta^2\big(\frac{\tilde{\epsilon}}{\varepsilon_1^2}\big)^{k+2+1/k}\Big).$$

*The total sample complexity in this case is given by $n_{\mathrm{total}} = n_{\mathrm{FA}} + n_{\mathrm{SFL}}(\zeta)$.*

The proof of both theorems follows from combining the results of the following sections. Since both proofs are similar, we only present the proof of Theorem 12. The proof of Theorems 6 and 7 can be obtained in a similar manner.

**Proof.** [Proof of Theorem 12] The proof is based on decomposing the suboptimality into generalization and approximation terms, namely

$$\mathrm{AR}(\hat{a}, W, b) - \mathrm{AR}^* = \mathrm{AR}(\hat{a}, W, b) - \mathrm{AR}(a^*, W, b) + \mathrm{AR}(a^*, W, b) - \mathrm{AR}^*,$$

where $a^* := \min_{\|a\| \leq r_a/\sqrt{N}} \mathrm{AR}(a, W, b)$, thus we can see the first term above as generalization error, and the second term as approximation error.

From Proposition 22, we have $\mathrm{AR}(\hat{a}, W, b) - \mathrm{AR}(a^*, W, b) \leq \epsilon/2$ as soon as $n \geq \tilde{\Omega}(r_a^4(\varepsilon_1^4 + r_b^4)/\epsilon^2)$ (recall that $q = 1$ here, since we are considering the ReLU activation). For the approximation error, we can use Proposition 36, which guarantees there exists $a^*$ with $\|a^*\| \leq r_a/\sqrt{N}$ such that $\mathrm{AR}(a^*, W, b) - \mathrm{AR}^* \leq \epsilon/2$ with $r_a \leq \tilde{\mathcal{O}}((\varepsilon_1/\sqrt{\tilde{\epsilon}})^{k+1+1/k}/\alpha)$, as soon as

$$\zeta \leq \tilde{\mathcal{O}}\Big(\big(\frac{\tilde{\epsilon}}{\varepsilon_1^2}\big)^{k+2+1/k}\Big), \qquad \text{and} \qquad N \geq \tilde{\Omega}\Big(\frac{1}{\zeta^{(k-1)/2}\alpha}\big(\frac{\varepsilon_1}{\sqrt{\tilde{\epsilon}}}\big)^{k+3+2/k}\Big),$$

provided that we choose $r_b = \tilde{\Theta}(\varepsilon_1(\varepsilon_1/\sqrt{\tilde{\epsilon}})^{1+1/k})$. Plugging the value of $r_a$ and $r_b$ in the bound for $n$ completes the proof. $\qquad\square$

## C GENERALIZATION ANALYSIS

We will first focus on proving a generalization bound for bounded and Lipschitz losses, and then extend the results to cover the squared loss. In this section, we will typically use $n$ to refer to $n_{\mathrm{FA}}$, the number of Phase 2 samples.

### C.1 GENERALIZATION BOUNDS FOR BOUNDED LIPSCHITZ LOSSES

Let us focus on a general $C_\ell$ Lipschitz loss $\ell(f(\cdot; a, W, b) - y)$ for now. Later, we will argue how to extend the results of this section to the squared error loss. Our uniform convergence argument depends on the covering number of the family of adversarial loss functions. Let $\Theta \subseteq \mathbb{R}^N$ be the set of second layer weights, to be determined later. This family is given by

$$\mathcal{L}(W, b) = \{(x, y) \mapsto \max_{\|\delta\| \leq \varepsilon} \ell(f(x + \delta; a, W, b) - y) : a \in \Theta\}.$$

For brevity, we will also use $\mathcal{L}$ to denote $\mathcal{L}(W, b)$, but we highlight that $W$ and $b$ are fixed at this stage. We define the following metric over this family

$$\forall \tilde{l}, \tilde{l}' \in \mathcal{L}(W, b), \quad d_{\mathcal{L}}(\tilde{l}, \tilde{l}')^2 := \frac{1}{n}\sum_{i=1}^n (\tilde{\ell}(x^{(i)}, y^{(i)}) - \tilde{\ell}'(x^{(i)}, y^{(i)}))^2.$$

We say $\mathcal{S} \subseteq \mathcal{L}$ is an $\epsilon$-cover of $\mathcal{L}$ if for every $\tilde{l} \in \mathcal{L}$, there exists $\tilde{l}' \in \mathcal{S}$ such that $d_{\mathcal{L}}(\tilde{l}, \tilde{l}') \leq \epsilon$. The $\epsilon$-covering number of $\mathcal{L}$ is the least cardinality among all $\epsilon$-covers of $\mathcal{L}$, which we denote by $\mathcal{C}(\mathcal{L}, d_{\mathcal{L}}, \epsilon)$. Note that since $\mathcal{L}$ is parmeterized by $a$, constructing such a covering reduces to constructing a finite set over $\Theta$.

Therefore, we define the following metric over $\Theta$,

$$\forall a, a' \in \Theta, \quad d_\Theta(a, a')^2 := \frac{1}{n}\sum_{i=1}^n \max_{\|\delta^{(i)}\| \leq \epsilon} \big(f(x^{(i)} + \delta^{(i)}; a, W, b) - f(x^{(i)} + \delta^{(i)}; a', W, b)\big)^2.$$

We can similarly define the $\epsilon$-covering number of $\Theta$ with respect to the metric $d_\Theta$ as $\mathcal{C}(\Theta, d_\Theta, \epsilon)$. The following lemma relates the covering numbers of $\mathcal{L}$ and $\Theta$.

**Lemma 14.** *We have $\mathcal{C}(\mathcal{L}, d_\mathcal{L}, \epsilon) \leq \mathcal{C}(\Theta, d_\Theta, \epsilon/C_\ell)$ for all $\epsilon > 0$.*

**Proof.** We will use the following fact in the proof. For any $F_1, F_2 : S \to \mathbb{R}$, we have

$$\left| \max_{\boldsymbol{\delta}_1 \in S} F_1(\boldsymbol{\delta}_1) - \max_{\boldsymbol{\delta}_2 \in S} F_2(\boldsymbol{\delta}_2) \right| \leq \max_{\boldsymbol{\delta} \in S} |F_1(\boldsymbol{\delta}) - F_2(\boldsymbol{\delta})|. \tag{C.1}$$

This is true because

$$\max_{\boldsymbol{\delta}_1 \in S} F_1(\boldsymbol{\delta}_1) - \max_{\boldsymbol{\delta}_2 \in S} F_2(\boldsymbol{\delta}_2) \leq \max_{\boldsymbol{\delta}_1 \in S} \left\{ F_1(\boldsymbol{\delta}_1) - F_2(\boldsymbol{\delta}_1) \right\},$$

and the other direction holds by symmetry. This trick is used to relate the adversarial loss to its non-adversarial counterpart, e.g. in Xiao et al. (2024, Lemma 5).

Now, we will show that an $\epsilon/C_\ell$ cover for $\Theta$ implies an $\epsilon$ cover for $\mathcal{L}$. We will supress dependence on the fixed $\boldsymbol{W}$ and $\boldsymbol{b}$ in the notation. Let $\mathcal{S}_\Theta$ be an $\epsilon/C_\ell$ cover of $\Theta$ with respect to the $d_\Theta$ metric. Then, we define $\mathcal{S}$ via

$$\mathcal{S} = \{(\boldsymbol{x}, y) \mapsto \max_{\|\boldsymbol{\delta}\| \leq \varepsilon} \ell(f(\boldsymbol{x} + \boldsymbol{\delta}; \boldsymbol{a}) - y) : \boldsymbol{a} \in \mathcal{S}_\Theta\}.$$

To show $\mathcal{S}$ is an $\epsilon$ cover of $\mathcal{L}$, consider an arbitrary $\tilde{\ell}(\boldsymbol{x}, y) = \max_{\|\boldsymbol{\delta}\| \leq \varepsilon} \ell(f(\boldsymbol{x} + \boldsymbol{\delta}; \boldsymbol{a}) - y)$. Suppose $\boldsymbol{a}'$ is the closest element to $\boldsymbol{a}$ in $\mathcal{S}_\Theta$, and let $\tilde{\ell}'(\boldsymbol{x}, y) = \max_{\|\boldsymbol{\delta}\| \leq \varepsilon} \ell(f(\boldsymbol{x} + \boldsymbol{\delta}; \boldsymbol{a}') - y)$. Then,

$$\begin{aligned}
d_\mathcal{L}(\tilde{\ell}, \tilde{\ell}')^2 &= \frac{1}{n} \sum_{i=1}^n \left( \max_{\|\boldsymbol{\delta}_1^{(i)}\| \leq \varepsilon} \ell(f(\boldsymbol{x} + \boldsymbol{\delta}_1^{(i)}; \boldsymbol{a}) - y^{(i)}) - \max_{\|\boldsymbol{\delta}_2^{(i)}\| \leq \varepsilon} \ell(f(\boldsymbol{x} + \boldsymbol{\delta}_2^{(i)}; \boldsymbol{a}') - y^{(i)}) \right)^2 \\
&\leq \frac{1}{n} \sum_{i=1}^n \max_{\|\boldsymbol{\delta}^{(i)}\| \leq \varepsilon} \left( \ell(f(\boldsymbol{x} + \boldsymbol{\delta}^{(i)}; \boldsymbol{a}) - y^{(i)}) - \ell(f(\boldsymbol{x} + \boldsymbol{\delta}^{(i)}; \boldsymbol{a}') - y^{(i)}) \right)^2 \\
&\leq \frac{C_\ell^2}{n} \sum_{i=1}^n \max_{\|\boldsymbol{\delta}^{(i)}\| \leq \varepsilon} \left( f(\boldsymbol{x} + \boldsymbol{\delta}^{(i)}; \boldsymbol{a}) - f(\boldsymbol{x} + \boldsymbol{\delta}^{(i)}; \boldsymbol{a}') \right)^2 \\
&\leq C_\ell^2 d_\Theta(\boldsymbol{a}, \boldsymbol{a}')^2 \leq \epsilon^2,
\end{aligned}$$

where we used (C.1) for the first inequality. $\qquad\square$

To construct an $\epsilon$-cover of $\Theta$, we depend on the Maurey sparsification lemma (Pisier, 1981), which has been used in the literature for providing covering numbers for linear classes (Zhang, 2002) and neural networks via matrix covering, see e.g. Bartlett et al. (2017).

**Lemma 15** (Maurey Sparsification Lemma, (Zhang, 2002, Lemma 1)). *Let $\mathcal{H}$ be a Hilbert space with norm $\|\cdot\|$, let $\boldsymbol{u} \in \mathcal{H}$ be represented by $\boldsymbol{u} = \sum_{j=1}^m \alpha_j \boldsymbol{v}_j$, where $\alpha_j \geq 0$ and $\|\boldsymbol{v}_j\| \leq b$ for all $j \in [m]$, and $\alpha = \sum_{j=1}^m \alpha_j \leq 1$. Then, for every $k \geq 1$, there exist non-negative integers $k_1, \ldots, k_m$, such that $\sum_{j=1}^m k_j \leq k$ and*

$$\left\| \boldsymbol{u} - \frac{1}{k} \sum_{j=1}^m k_j \boldsymbol{v}_j \right\|^2 \leq \frac{\alpha b^2 - \|\boldsymbol{u}\|^2}{k}.$$

Then, we have the following upper bound on the the covering number of $\Theta$.

**Lemma 16.** *Suppose $\sigma$ satisfies (B.1), $\Theta = \{\|\boldsymbol{a}\|_1 \leq r_a\}$, and additionally $\|\boldsymbol{w}_i\| \leq r_w$ and $|b_i| \leq r_b$ for all $1 \leq i \leq N$. Then we have*

$$\log \mathcal{C}(\Theta, d_\Theta, \epsilon) \leq \frac{C_{\bar{q}} L_\sigma^2 r_a^2 \log N \left\{ T_{\boldsymbol{W}, \boldsymbol{X}}^{(\bar{q})} + r_w^{2\bar{q}} \varepsilon^{2\bar{q}} + r_b^{2\bar{q}} + T_{\boldsymbol{W}, \boldsymbol{X}}^{(2)} + r_w^2 \varepsilon^2 + r_b^2 \right\}}{\epsilon^2},$$

*where $T_{\boldsymbol{W}, \boldsymbol{X}}^{(\bar{q})} := \max_{1 \leq j \leq N} \frac{1}{n} \sum_{i=1}^n \langle \boldsymbol{w}_j, \boldsymbol{x}_i \rangle^{2\bar{q}}$.*

**Proof.** Given some positive integer $k > 0$, let $\mathcal{S}_\Theta$ be given by the following

$$\mathcal{S}_\Theta = \left\{ \frac{r_a}{k}(k_1 - k_1', k_2 - k_2', \ldots, k_N - k_N')^\top : \forall i, \ k_i, k_i' \geq 0, \quad \sum_{i=1}^N k_i + \sum_{i=1}^N k_i' = k \right\}.$$

Let $\boldsymbol{X}, \boldsymbol{\Delta} \in \mathbb{R}^{n \times d}$ be the matrices with $(\boldsymbol{x}_i)$ and $(\boldsymbol{\delta}_i)$ as rows respectively. Let $\boldsymbol{A} = \sigma((\boldsymbol{X} + \boldsymbol{\Delta})\boldsymbol{W}^\top + \mathbf{1}_n \boldsymbol{b}^\top) \in \mathbb{R}^{n \times N}$. Then,

$$\frac{1}{n}\sum_{i=1}^n \left(f(\boldsymbol{x}^{(i)}+\boldsymbol{\delta}^{(i)}; \boldsymbol{a}, \boldsymbol{W}, \boldsymbol{b}) - f(\boldsymbol{x}^{(i)}+\boldsymbol{\delta}^{(i)}; \boldsymbol{a}', \boldsymbol{W}, \boldsymbol{b})\right)^2 = \frac{1}{n}\|\boldsymbol{A}(\boldsymbol{a} - \boldsymbol{a}')\|^2 = \frac{1}{n}\left\|\sum_{i=1}^N \boldsymbol{A}_i(a_i - a_i')\right\|^2,$$

where $\boldsymbol{A}_i = \sigma((\boldsymbol{X} + \boldsymbol{\Delta})\boldsymbol{w}_i + \mathbf{1}_n b_i)$ is the $i$th column of $\boldsymbol{A}$. We are going to choose $\boldsymbol{a}'$ from $\mathcal{S}_\Theta$. To that end, define

$$\tilde{\boldsymbol{A}}_i = \text{sign}(a_i)\boldsymbol{A}_i.$$

By Maurey's sparsification lemma ([Xiao et al., 2024](#), Lemma 13), there exist $\tilde{k}_i \geq 0$ with $\sum_{i=1}^n \tilde{k}_i = k$ such that

$$\left\|\sum_{i=1}^N |a_i|\tilde{\boldsymbol{A}}_i - \frac{r_a}{k}\sum_{i=1}^N \tilde{k}_i\tilde{\boldsymbol{A}}_i\right\|^2 \leq \frac{r_a^2 b^2}{k},$$

where $\|\boldsymbol{A}_i\| \leq b$ for all $i$. We will then choose

$$k_i = \begin{cases} \tilde{k}_i, & \text{sign}(a_i) \geq 0, \\ 0, & \text{sign}(a_i) < 0 \end{cases}, \qquad k_i' = \begin{cases} 0, & \text{sign}(a_i) \geq 0, \\ \tilde{k}_i, & \text{sign}(a_i) < 0 \end{cases}.$$

Therefore, we have $\sum_{i=1}^N \tilde{k}_i = k$. Finally, with the constructed $(k_i)$ and $(k_i')$, let

$$\boldsymbol{a}' = \frac{r_a}{k}(k_1 - k_1', \ldots, k_N - k_N')^\top,$$

and also note that $\sum_{i=1}^N |a_i|\tilde{\boldsymbol{A}}_i = \sum_{i=1}^N a_i\boldsymbol{A}_i$. Consequently, given $\boldsymbol{a}$, we have constructed $\boldsymbol{a}' \in \mathcal{S}_\Theta$ such that

$$\frac{1}{n}\left\|\sum_{i=1}^N \boldsymbol{A}_i(a_i - a_i')\right\|^2 \leq \frac{r_a^2 b^2}{nk}.$$

Next, we provide a bound on $b$. By the assumptions on $\sigma$, we have

$$\|\boldsymbol{A}_i\|^2 \lesssim C_{\bar{q}}L_\sigma^2\left(\|\boldsymbol{X}\boldsymbol{w}_i\|_{2\bar{q}}^{2\bar{q}} + \|\boldsymbol{\Delta}\boldsymbol{w}_i\|_{2\bar{q}}^{2\bar{q}} + nb_i^{2\bar{q}} + \|\boldsymbol{X}\boldsymbol{w}_i\|^2 + \|\boldsymbol{\Delta}\|^2 + nb_i^2\right)$$

$$\lesssim nC_{\bar{q}}L_\sigma^2\left(T_{\boldsymbol{W},\boldsymbol{X}}^{(\bar{q})} + r_w^{2\bar{q}}\varepsilon^{2\bar{q}} + r_b^{2\bar{q}} + T_{\boldsymbol{W},\boldsymbol{X}}^{(2)} + r_w^2\varepsilon^2 + r_b^2\right).$$

Consequently, we can choose

$$k = \left\lceil \frac{C_{\bar{q}}L_\sigma^2 r_a^2\left(T_{\boldsymbol{W},\boldsymbol{X}}^{(\bar{q})} + r_w^{2\bar{q}}\varepsilon^{2\bar{q}} + r_b^{2\bar{q}} + T_{\boldsymbol{W},\boldsymbol{X}}^{(2)} + r_w^2\varepsilon^2 + r_b^2\right)}{\epsilon^2} \right\rceil.$$

Finally, we need to count $|\mathcal{S}_\Theta|$. Note that

$$|\mathcal{S}_\Theta| = \binom{2N + k - 1}{k} \leq \left(\frac{e(2N + k - 1)}{k}\right)^k \leq (3eN)^k,$$

which concludes the proof. $\qquad\square$

We can now turn the above covering number into Rademacher complexity via a chaining argument, as follows.

**Lemma 17.** *Let $\mathfrak{R}(\mathcal{L}(\boldsymbol{W}, \boldsymbol{b}))$ denote the Rademacher complexity of the class of adversarial loss functions $\mathcal{L}(\boldsymbol{W}, \boldsymbol{b})$, defined via*

$$\mathfrak{R}(\mathcal{L}(\boldsymbol{W}, \boldsymbol{b})) := \mathbb{E}\left[\sup_{\boldsymbol{a} \in \Theta}\left|\frac{1}{n}\sum_{i=1}^{n} \xi_i \max_{\|\boldsymbol{\delta}^{(i)}\| \leq \varepsilon} \ell(f(\boldsymbol{x}^{(i)} + \boldsymbol{\delta}^{(i)}; \boldsymbol{a}, \boldsymbol{W}, \boldsymbol{b}), y^{(i)})\right|\right],$$

*where $\xi_i$ are i.i.d. Rademacher random variables and $\Theta = \{\boldsymbol{a} : \|\boldsymbol{a}\|_1 \leq r_a\}$. For simplicity, assume $C_\ell, r_a \gtrsim 1$. Then we have*

$$\mathfrak{R}(\mathcal{L}(\boldsymbol{W}, \boldsymbol{b})) \lesssim \frac{C_\ell C_{\bar{q}} L_\sigma r_a \log n \log N\left(\mathbb{E}\left[\sqrt{T_{\boldsymbol{W},\boldsymbol{X}}^{(\bar{q})}}\right] + r_w^{\bar{q}}\varepsilon^{\bar{q}} + r_b^{\bar{q}} + \mathbb{E}\left[\sqrt{T_{\boldsymbol{W},\boldsymbol{X}}^{(2)}}\right] + r_w\varepsilon + r_b\right)}{\sqrt{n}}.$$

**Proof.** Let $\mathfrak{R}_n(\mathcal{L}(\boldsymbol{W}, \boldsymbol{b}))$ denote the empirical Rademacher complexity by

$$\mathfrak{R}_n(\mathcal{L}(\boldsymbol{W}, \boldsymbol{b})) := \mathbb{E}_{\boldsymbol{\xi}}\left[\sup_{\boldsymbol{a} \in \Theta}\left|\frac{1}{n}\sum_{i=1}^{n} \xi_i \max_{\|\boldsymbol{\delta}^{(i)}\| \leq \varepsilon} \ell(f(\boldsymbol{x}^{(i)} + \boldsymbol{\delta}^{(i)}; \boldsymbol{a}, \boldsymbol{W}, \boldsymbol{b}), y^{(i)})\right|\right],$$

where the expectation is only taken w.r.t. the randomness of $\xi$ and is conditional on the training set. For simplicity, define

$$B := C_{\bar{q}} L_\sigma\left(\sqrt{T_{\boldsymbol{W},\boldsymbol{X}}^{(\bar{q})}} + r_w^{\bar{q}}\varepsilon^{\bar{q}} + r_b^{\bar{q}} + \sqrt{T_{\boldsymbol{W},\boldsymbol{X}}^{(2)}} + r_w\varepsilon^2 + r_b\right).$$

Then, by a standard chaining argument, we have for all $\alpha > 0$,

$$\mathfrak{R}_n(\mathcal{L}(\boldsymbol{W}, \boldsymbol{b})) \lesssim \alpha + \int_{\epsilon=\alpha}^{\infty} \sqrt{\frac{\log \mathcal{C}(\mathcal{L}, d_{\mathcal{L}}, \epsilon)}{n}} \mathrm{d}\epsilon$$

$$\lesssim \alpha + \frac{C_\ell r_a B \log N}{\sqrt{n}} \log\left(\frac{1}{\alpha}\right).$$

By choosing $\alpha = 1/\sqrt{n}$, we obtain

$$\mathfrak{R}_n(\mathcal{L}(\boldsymbol{W}, \boldsymbol{b})) \lesssim \frac{C_\ell r_a B \log n \log N}{\sqrt{n}}.$$

Taking expectations with respect to the input distribution completes the proof. $\square$

Note that it remains to provide an upper bound for $T_{\boldsymbol{W},\boldsymbol{x}}^{(\bar{q})}$ introduced in Lemma 16. This is achieved by the following lemma.

**Lemma 18.** *Suppose $\|\boldsymbol{w}_i\| \leq r_w$. Then, for all $\bar{q} > 0$ and $N > e$, we have*

$$\mathbb{E}\left[\max_{1 \leq j \leq N} \frac{1}{n}\sum_{i=1}^{n}\left\langle \boldsymbol{w}_j, \boldsymbol{x}^{(i)}\right\rangle^{2\bar{q}}\right] \leq C_{\bar{q}} r_w^{2\bar{q}}(\log N)^{\bar{q}},$$

*where $C_{\bar{q}}$ is a constant depending only on $\bar{q}$.*

**Proof.** For conciseness, let $Z_j := \frac{1}{n}\sum_{i=1}^{n}\left\langle \boldsymbol{w}_j, \boldsymbol{x}^{(i)}\right\rangle^{2\bar{q}}$. By non-negativity of $Z_j$ and Jensen's inequality, for all $t \geq 1$ we have

$$\mathbb{E}\left[\max_{1 \leq j \leq N} Z_j\right] \leq \mathbb{E}\left[\max_{1 \leq j \leq N} Z_j^t\right]^{1/t} \leq \left(\sum_{j=1}^{N} \mathbb{E}[Z_j^t]\right)^{1/t} \leq N^{1/t}\left(\max_{1 \leq j \leq N} \mathbb{E}[Z_j^t]\right)^{1/t}.$$

Further, by Jensens's inequality

$$\mathbb{E}[Z_j^t] = \mathbb{E}\left[\left(\frac{1}{n}\sum_{i=1}^{n}\left\langle \boldsymbol{w}_j, \boldsymbol{x}^{(i)}\right\rangle^{2\bar{q}}\right)^t\right]$$

$$\leq \mathbb{E}\left[\frac{1}{n}\sum_{i=1}^{n}\left\langle \boldsymbol{w}_j, \boldsymbol{x}^{(i)}\right\rangle^{2\bar{q}t}\right]$$

$$\leq (Cr_w)^{2\bar{q}t}(2\bar{q}t)^{\bar{q}t},$$

where $C > 0$ is a absolute constant, and we used the moment bound of subGaussian random variables along with the fact that $\langle \boldsymbol{w}_j, \boldsymbol{x} \rangle$ is a centered subGaussian random variable with subGaussian norm $\mathcal{O}(r_w)$. As a result,

$$\mathbb{E}\left[\max_{1 \leq j \leq N} Z_j\right] \leq C_{\bar{q}} r_w^{2\bar{q}} N^{1/t} t^{\bar{q}} \lesssim C_{\bar{q}} r_w^{2\bar{q}} (\log N)^{\bar{q}},$$

where the last inequality follows by choosing $t = \log N$. $\qquad\qquad\square$

As a consequence, if the loss is also bounded, we get the following high-probability concentration bound.

**Corollary 19.** *Suppose $|\tilde{\ell}| \leq B_\ell$ for all $\tilde{\ell} \in \mathcal{L}(\boldsymbol{W}, \boldsymbol{b})$. Then, with probability at least $1 - \delta$ we have*

$$\left| \sup_{\tilde{\ell} \in \mathcal{L}(\boldsymbol{W}, \boldsymbol{b})} \mathbb{E}\left[\tilde{\ell}(\boldsymbol{x}, y)\right] - \frac{1}{n} \sum_{i=1}^{n} \tilde{\ell}(\boldsymbol{x}^{(i)}, y^{(i)}) \right| \lesssim \frac{C_\ell r_a R \log n \log N + B_\ell \sqrt{\log(1/\delta)}}{\sqrt{n}},$$

*where*

$$R := C_{\bar{q}} L_\sigma (r_w^{\bar{q}} (\log^{q/2} N + \varepsilon^{\bar{q}}) + r_b^{\bar{q}} + r_w (\log^{1/2} N + \varepsilon) + r_b).$$

## C.2 Applying the Generalization Bound to Squared Loss

To apply the generalization argument above to the squared loss, we bound it with a threshold $\tau$, and define the loss family

$$\mathcal{L}_\tau(\boldsymbol{W}, \boldsymbol{b}) := \left\{ (\boldsymbol{x}, y) \mapsto \left\{ \max_{\|\boldsymbol{\delta}\| \leq \varepsilon} (f(\boldsymbol{x} + \boldsymbol{\delta}; \boldsymbol{a}, \boldsymbol{W}, \boldsymbol{b}) - y)^2 \wedge \tau : \boldsymbol{a} \in \Theta \right\}.$$

We similarly define $\mathrm{AR}_\tau$ and $\widehat{\mathrm{AR}}_\tau$. Recall that our goal is to show

$$\mathrm{AR}(\hat{\boldsymbol{a}}, \boldsymbol{W}, \boldsymbol{b}) \leq \widehat{\mathrm{AR}}(\hat{\boldsymbol{a}}, \boldsymbol{W}, \boldsymbol{b}) + \epsilon_1(n, N, d).$$

We readily have $\widehat{\mathrm{AR}}_\tau(\hat{\boldsymbol{a}}, \boldsymbol{W}, \boldsymbol{b}) \leq \widehat{\mathrm{AR}}(\hat{\boldsymbol{a}}, \boldsymbol{W}, \boldsymbol{b})$. Further, Corollary 19 yields

$$\left| \mathrm{AR}_\tau(\hat{\boldsymbol{a}}, \boldsymbol{W}, \boldsymbol{b}) - \widehat{\mathrm{AR}}_\tau(\hat{\boldsymbol{a}}, \boldsymbol{W}, \boldsymbol{b}) \right| \lesssim \frac{\sqrt{\tau} r_a R \log n \log N}{\sqrt{n}} + \tau \sqrt{\frac{\log(1/\delta)}{n}},$$

with probability at least $1 - \delta$. Thus, the remaining step is to bound $\mathrm{AR}(\hat{\boldsymbol{a}}, \boldsymbol{W}, \boldsymbol{b})$ and $\widehat{\mathrm{AR}}(\hat{\boldsymbol{a}}, \boldsymbol{W}, \boldsymbol{b})$ with their clipped versions. To do so, we first provide the following tail probability estimate.

**Lemma 20.** *Suppose $(z_j)_{j=1}^N$ are non-negative random variables with subGaussian norm $r$. Then, for any $\bar{q} > 0$ and $\tau \geq C_{\bar{q}} r^{\bar{q}}$ where $C_{\bar{q}}$ is a constant depending only on $\bar{q}$, we have*

$$\mathbb{P}\left( \frac{1}{N} \sum_{j=1}^{N} z_j^{\bar{q}} \geq \tau \right) \leq \exp\left( -\frac{c \tau^{2/\bar{q}}}{r^2} \right),$$

*where $c > 0$ is an absolute constant.*

**Proof.** For any $t \geq 1$, we have the following Markov bound,

$$\mathbb{P}\left( \frac{1}{N} \sum_{j=1}^{N} z_j^{\bar{q}} \geq \tau \right) = \mathbb{P}\left( \left( \frac{1}{N} \sum_{j=1}^{N} z_j^{\bar{q}} \right)^t \geq \tau^t \right) \leq \frac{\mathbb{E}\left[ \left( \frac{1}{N} \sum_{j=1}^{N} z_j^{\bar{q}} \right)^t \right]}{\tau^t} \leq \frac{\mathbb{E}\left[ \frac{1}{N} \sum_{j=1}^{N} z_j^{\bar{q}t} \right]}{\tau^t},$$

where the last inequality follows from Jensen's inequality. Further, by subGaussianity of $z_j$, we have $\mathbb{E}\left[ z_j^{\bar{q}t} \right] \leq (Cr^2 \bar{q} t)^{\bar{q}t/2}$, where $C > 0$ is an absolute constant. As a result,

$$\mathbb{P}\left( \frac{1}{N} \sum_{j=1}^{N} z_j^{\bar{q}} \geq \tau \right) \leq \frac{(Cr^2 \bar{q} t)^{\bar{q}t/2}}{\tau^t}.$$

The above bound is minimized at $t = \frac{\tau^{2/\bar{q}}}{Cr^2\bar{q}e}$. Note that $t \geq 1$ requires $\tau \geq C_{\bar{q}}r^{\bar{q}}$. Plugging this choice of $t$ in the above bound yields

$$\mathbb{P}\left(\frac{1}{N}\sum_{j=1}^{N}z_j^{\bar{q}} \geq \tau\right) \leq \exp\left(-\frac{\tau^{2/\bar{q}}}{2Cr^2e}\right),$$

which completes the proof. $\qquad\square$

**Lemma 21.** *Suppose Assumption 2 holds. Let $\Theta = \{\boldsymbol{a} : \|\boldsymbol{a}\| \leq r_a/\sqrt{N}\}$, $\|\boldsymbol{w}_i\| \leq r_w$, and $|b_i| \leq r_b$ for all $i \in [N]$. Assume $\sigma$ satisfes (B.1). Define $\varepsilon_1 := 1 \vee \varepsilon$, and let*

$$\varkappa := C_{\bar{q}}r_a^2 L_\sigma^2(r_w^{2\bar{q}}\varepsilon_1^{2\bar{q}} + r_b^{2\bar{q}} + r_w^2\varepsilon_1^2 + r_b^2) + C_p,$$

*where $C_{\bar{q}}$ and $C_p$ are constants depending only on $\bar{q}$ and $p$ respectively. Then, for all*

$$\tau \geq C\left\{\varkappa \vee L_\sigma^2 r_w^{2\bar{q}}\log^{\bar{q}}\frac{n}{\delta} \vee \log^p\frac{n}{\delta}\right\},$$

*we have*

$$\left|\mathrm{AR}(\boldsymbol{a},\boldsymbol{W},\boldsymbol{b}) - \widehat{\mathrm{AR}}(\boldsymbol{a},\boldsymbol{W},\boldsymbol{b})\right| \leq \left|\mathrm{AR}_\tau(\boldsymbol{a},\boldsymbol{W},\boldsymbol{b}) - \widehat{\mathrm{AR}}_\tau(\boldsymbol{a},\boldsymbol{W},\boldsymbol{b})\right|$$
$$+ C\varkappa\left(\exp\left(-\Omega\left(\frac{\tau^{1/\bar{q}}}{L_\sigma^{2/\bar{q}}r_w^2}\right)\right) + \exp(-\Omega(\tau^{1/p}))\right),$$

*with probability at least $1 - \delta$ uniformly over all $\boldsymbol{a} \in \Theta$.*

**Proof.** Since $\boldsymbol{W}$ and $\boldsymbol{b}$ are fixed, we use the shorthand notation $f(\boldsymbol{x};\boldsymbol{a}) = f(\boldsymbol{x};\boldsymbol{a},\boldsymbol{W},\boldsymbol{b})$.

In the first section of the proof, we will upper and lower bound $\mathrm{AR}(\boldsymbol{a},\boldsymbol{W},\boldsymbol{b})$ with $\mathrm{AR}_\tau(\boldsymbol{a},\boldsymbol{W},\boldsymbol{b})$. Note that the lower bound is trivial as $\mathrm{AR}_\tau(\boldsymbol{a},\boldsymbol{W},\boldsymbol{b}) \leq \mathrm{AR}(\boldsymbol{a},\boldsymbol{W},\boldsymbol{b})$, thus we move on to the upper bound. Let

$$\tilde{\ell}(\boldsymbol{x},y) = \max_{\|\boldsymbol{\delta}\|\leq\varepsilon}(f(\boldsymbol{x}+\boldsymbol{\delta};\boldsymbol{a}) - y)^2.$$

Then,

$$\mathrm{AR}(\boldsymbol{a},\boldsymbol{W},\boldsymbol{b}) = \mathbb{E}\left[\tilde{\ell}(\boldsymbol{x},y)\mathbb{I}\left[\tilde{\ell}(\boldsymbol{x},y) \leq \tau\right]\right] + \mathbb{E}\left[\tilde{\ell}(\boldsymbol{x},y)\mathbb{I}\left[\tilde{\ell}(\boldsymbol{x},y) > \tau\right]\right]$$
$$\leq \mathrm{AR}_\tau(\boldsymbol{a},\boldsymbol{W},\boldsymbol{b}) + \mathbb{E}\left[\tilde{\ell}(\boldsymbol{x},y)^2\right]^{1/2}\mathbb{P}\left(\tilde{\ell}(\boldsymbol{x},y) \geq \tau\right)^{1/2}.$$

Further, we have the following upper bound for the adversarial loss,

$$\tilde{\ell}(\boldsymbol{x},y) = \max_{\|\boldsymbol{\delta}\|\leq\varepsilon}(f(\boldsymbol{x}+\boldsymbol{\delta};\boldsymbol{a}) - y)^2$$
$$\lesssim \max_{\|\boldsymbol{\delta}\|\leq\varepsilon}f(\boldsymbol{x}+\boldsymbol{\delta};\boldsymbol{a})^2 + y^2$$
$$\lesssim \max_{\|\boldsymbol{\delta}\|\leq\varepsilon}\|\boldsymbol{a}\|^2\|\sigma(\boldsymbol{W}(\boldsymbol{x}+\boldsymbol{\delta}) + \boldsymbol{b})\|^2 + y^2$$
$$\lesssim r_a^2 C_{\bar{q}} L_\sigma^2\left(\frac{1}{N}\sum_{j=1}^{N}\langle\boldsymbol{w}_j,\boldsymbol{x}\rangle^{2\bar{q}} + r_w^{2\bar{q}}\varepsilon^{2\bar{q}} + r_b^{2\bar{q}} + \frac{1}{N}\sum_{j=1}^{N}\langle\boldsymbol{w}_j,\boldsymbol{x}\rangle^2 + r_w^2\varepsilon^2 + r_b^2\right) + y^2$$

Moreover, by Jensen's inequality,

$$\mathbb{E}\left[\left(\frac{1}{N}\sum_{j=1}^{N}\langle\boldsymbol{w}_j,\boldsymbol{x}\rangle^{2\bar{q}}\right)^2\right] \leq \mathbb{E}\left[\frac{1}{N}\sum_{j=1}^{N}\langle\boldsymbol{w}_j,\boldsymbol{x}\rangle^{4\bar{q}}\right]$$
$$\leq (Cr_w)^{4\bar{q}}(4\bar{q})^{2\bar{q}} \leq C_{\bar{q}}r_w^{4\bar{q}}$$

for all $\bar{q} > 0$, where $C$ is an absolute constant and we used the subGaussianity of $\langle \boldsymbol{w}_j, \boldsymbol{x} \rangle$ to bound its moment. As a result,

$$\mathbb{E}\Big[\tilde{\ell}(\boldsymbol{x}, y)^2\Big]^{1/2} \lesssim r_a^2 C_{\bar{q}} L_\sigma^2 (r_w^{2\bar{q}}(1 + \varepsilon^{2\bar{q}}) + r_b^{2\bar{q}} + r_w^2(1 + \varepsilon^2) + r_b^2) + \mathbb{E}\big[y^4\big]^{1/2}.$$

By assumption 2, we have $\mathbb{E}\big[y^4\big]^{1/2} \le C_p$.

To estimate the tail probability of $\tilde{\ell}(\boldsymbol{x}, y)$. Using the assumption on $\tau$ and the upper bound on $\tilde{\ell}(\boldsymbol{x}, y)$ developed above, via a union bound we have

$$\mathbb{P}\Big(\tilde{\ell}(\boldsymbol{x}, y) \ge \tau\Big) \le \mathbb{P}\left(\frac{L_\sigma^2}{N} \sum_{j=1}^N \langle \boldsymbol{w}_j, \boldsymbol{x} \rangle^{2\bar{q}} + \frac{L_\sigma^2}{N} \sum_{j=1}^N \langle \boldsymbol{w}_j, \boldsymbol{x} \rangle^2 + y^2 \ge \frac{\tau}{2}\right)$$

$$\le \mathbb{P}\left(\frac{L_\sigma^2}{N} \sum_{j=1}^N \langle \boldsymbol{w}_j, \boldsymbol{x} \rangle^{2\bar{q}} \ge \frac{\tau}{6}\right) + \mathbb{P}\left(\frac{L_\sigma^2}{N} \sum_{j=1}^N \langle \boldsymbol{w}_j, \boldsymbol{x} \rangle^2 \ge \frac{\tau}{6}\right) + \mathbb{P}\left(y^2 \ge \frac{\tau}{6}\right)$$

$$\le 2 \exp\left(\frac{-c\tau^{1/\bar{q}}}{L_\sigma^{2/\bar{q}} r_w^2}\right) + \mathbb{P}\big(y^2 \ge \tau\big),$$

where we used Lemma 20, the fact that $|\langle \boldsymbol{w}_j, \boldsymbol{x} \rangle|$ is subGaussian with norm $\mathcal{O}(r_w)$, and that $\bar{q} \ge 1$. Furthermore, using the moment estimate on $y$ in Assumption 2 along with the technique developed in Lemma 20, we have

$$\mathbb{P}\Big(y^2 \ge \frac{\tau}{6}\Big) \le \exp\Big(-c\tau^{1/p}\Big),$$

for $\tau \ge C_p$, where $c > 0$ is an absolute constant.

As a result, we obtain

$$\mathrm{AR}(\boldsymbol{a}, \boldsymbol{W}, \boldsymbol{b}) - \mathrm{AR}_\tau(\boldsymbol{a}, \boldsymbol{W}, \boldsymbol{b}) \lesssim \varkappa \left(\exp\Big(-\frac{c\tau^{1/\bar{q}}}{L_\sigma^{2/\bar{q}} r_w^2}\Big) + \exp(-c\tau^{1/p})\right),$$

for all $\boldsymbol{a} \in \Theta$.

In the next part of the proof, we will show that with probability at least $1-\delta$, we have $\widehat{\mathrm{AR}}(\boldsymbol{a}, \boldsymbol{W}, \boldsymbol{b}) = \widehat{\mathrm{AR}}_\tau(\boldsymbol{a}, \boldsymbol{W}, \boldsymbol{b})$ uniformly over all $\boldsymbol{a}$. Note that this is equivalent to asking $\tilde{\ell}(\boldsymbol{x}^{(i)}, y^{(i)}) \le \tau$ for all $1 \le i \le n$. For any fixed $i$, using the upper bound on $\tilde{\ell}(\boldsymbol{x}^{(i)}, y^{(i)})$, we have

$$\mathbb{P}\Big(\tilde{\ell}(\boldsymbol{x}^{(i)}, y^{(i)}) \ge \tau\Big) \le \mathbb{P}\left(\frac{L_\sigma^2}{N} \sum_{j=1}^N \langle \boldsymbol{w}_j, \boldsymbol{x} \rangle^{2\bar{q}} + \frac{L_\sigma^2}{N} \sum_{j=1}^N \langle \boldsymbol{w}_j, \boldsymbol{x} \rangle^2 + y^2 \ge \frac{\tau}{2}\right)$$

$$\lesssim \exp\left(\frac{-c\tau^{1/\bar{q}}}{L_\sigma^{2/\bar{q}} r_w^2}\right) + \exp\Big(-c\tau^{1/p}\Big).$$

Consequently, by a union bound we have

$$\mathbb{P}\left(\max_{1 \le i \le n} \tilde{\ell}(\boldsymbol{x}^{(i)}, y^{(i)}) \ge \tau\right) \le n\left(\exp\Big(\frac{-c\tau^{1/\bar{q}}}{L_\sigma^{2/\bar{q}} r_w^2}\Big) + \exp\Big(-c\tau^{1/p}\Big)\right).$$

Choosing

$$\tau \ge C\Big\{L_\sigma^2 r_w^{2\bar{q}} \log^{\bar{q}} \frac{n}{\delta} \vee \log^p \frac{n}{\delta}\Big\}$$

with a sufficiently large constant $C$ ensures the above probability is at most $\delta$, finishing the proof.

$\square$

We are now ready to present the main result of this section.

**Proposition 22.** *Suppose Assumption 2 holds and $\sigma$ satisfies (B.1), $\Theta = \{a : \|a\| \leq r_a/\sqrt{N}\}$, $\|w_i\| \leq 1$, and $|b_i| \leq r_b$ for all $1 \leq i \leq N$. Let*

$$\varkappa := C_{\bar{q}}r_a^2 L_\sigma^2(1 + \varepsilon^{2\bar{q}} + r_b^{2\bar{q}}) + C_p,$$

*where $C_{\bar{q}}$ and $C_p$ are constants depending only on $\bar{q}$ and $p$ respectively. Then we have*

$$\mathrm{AR}(\hat{a}, \boldsymbol{W}, \boldsymbol{b}) - \min_{\boldsymbol{a} \in \Theta} \mathrm{AR}(\boldsymbol{a}, \boldsymbol{W}, \boldsymbol{b}) \leq \tilde{\mathcal{O}}\left(\frac{\varkappa}{\sqrt{n}}\right),$$

*with probability at least $1 - \mathcal{O}(n^{-c})$ for some constant $c > 0$.*

**Proof.** We can summarize the generalization bound of Corollary 19 as

$$\left|\mathrm{AR}_\tau(\hat{a}, \boldsymbol{W}, \boldsymbol{b}) - \widehat{\mathrm{AR}}_\tau(\hat{a}, \boldsymbol{W}, \boldsymbol{b})\right| \lesssim \sqrt{\frac{\tau\varkappa}{n}} + \tau\sqrt{\frac{\log(1/\delta)}{n}},$$

where

$$\varkappa := C_{\bar{q}}r_a^2 L_\sigma^2(1 + \varepsilon^{2\bar{q}} + r_b^{2\bar{q}}) + C_p,$$

is obtained from Lemma 21 by letting $r_w = 1$. Thanks to Lemma 21, we arrive at

$$\mathrm{AR}(\hat{a}, \boldsymbol{W}, \boldsymbol{b}) - \widehat{\mathrm{AR}}(\hat{a}, \boldsymbol{W}, \boldsymbol{b}) \leq \tilde{\mathcal{O}}\left(\sqrt{\frac{\tau\varkappa}{n}} + \tau\sqrt{\frac{\log(1/\delta)}{n}} + \varkappa e^{-\Omega\left(\frac{\tau^{1/\bar{q}}}{L_\sigma^{2/\bar{q}}}\right)} + \varkappa e^{-\Omega\left(\tau^{1/p}\right)}\right).$$

Note that $\varkappa \gtrsim L_\sigma^2$. Choosing $\tau = C\varkappa\log^{p\vee\bar{q}}(\varkappa n/\delta)$ with a sufficiently large absolute constant $C > 0$ satisfies the assumption of Lemma 21. By letting $\delta = n^{-c}$ for some constant $c > 0$, we obtain

$$\mathrm{AR}(\hat{a}, \boldsymbol{W}, \boldsymbol{b}) - \widehat{\mathrm{AR}}(\hat{a}, \boldsymbol{W}, \boldsymbol{b}) \leq \tilde{\mathcal{O}}\left(\frac{\varkappa}{\sqrt{n}}\right),$$

which holds with probability at least $1 - n^{-c}$ over the randomness of the training set.

Recall $\boldsymbol{a}^* = \arg\min_{\boldsymbol{a} \in \Theta} \mathrm{AR}(\boldsymbol{a}, \boldsymbol{W}, \boldsymbol{b})$. Similarly, Lemma 21 guarantees

$$\widehat{\mathrm{AR}}(\boldsymbol{a}^*, \boldsymbol{W}, \boldsymbol{b}) - \mathrm{AR}(\boldsymbol{a}^*, \boldsymbol{W}, \boldsymbol{b}) \leq \tilde{\mathcal{O}}\left(\frac{\varkappa}{\sqrt{n}}\right),$$

on the same event as above. Finally, we have $\widehat{\mathrm{AR}}(\hat{a}, \boldsymbol{W}, \boldsymbol{b}) \leq \widehat{\mathrm{AR}}(\boldsymbol{a}^*, \boldsymbol{W}, \boldsymbol{b})$ by definition of $\hat{a}$, which concludes the proof of the proposition. $\qquad\square$

## D  APPROXIMATION ANALYSIS

Let $\Pi_{\boldsymbol{U}}\boldsymbol{w} = \frac{\boldsymbol{U}^\top \boldsymbol{U}\boldsymbol{w}}{\|\boldsymbol{U}\boldsymbol{w}\|}$ denote the projection of $\boldsymbol{w} \in \mathbb{S}^{d-1}$ onto $\mathrm{span}(\boldsymbol{u}_1, \ldots, \boldsymbol{u}_k) \cap \mathbb{S}^{d-1}$ (if $\|\boldsymbol{U}\boldsymbol{w}\| = 0$ we can simply let $\Pi_{\boldsymbol{U}}\boldsymbol{w} = \boldsymbol{u}_1$). Suppose $\langle \boldsymbol{w}, \boldsymbol{u}\rangle \geq 1 - \zeta$ for some $\zeta \in (0, 1)$ and $\boldsymbol{u} \in \mathrm{span}(\boldsymbol{u}_1, \ldots, \boldsymbol{u}_k)$ with $\|\boldsymbol{u}\| = 1$. Then, we have the following properties for this projection:

- $\langle \Pi_{\boldsymbol{U}}\boldsymbol{w}, \boldsymbol{u}\rangle \geq 1 - \zeta$,
- $\|\boldsymbol{w} - \Pi_{\boldsymbol{U}}\boldsymbol{w}\| \leq \sqrt{2\zeta}$.

Let $h : \mathbb{R}^k \to \mathbb{R}$ be the function constructed in the proof of Theorem 1. Then,

$$\mathrm{AR}^* = \mathbb{E}\left[\max_{\|\boldsymbol{\delta}\| \leq \varepsilon}(h(\boldsymbol{U}(\boldsymbol{x} + \boldsymbol{\delta})) - y)^2\right].$$

Let us denote $f(\boldsymbol{x}) = f(\boldsymbol{x}; \boldsymbol{a}^*, \boldsymbol{W}, \boldsymbol{b})$ for conciseness. Then,

$$\mathrm{AR}(\boldsymbol{a}^*, \boldsymbol{W}, \boldsymbol{b}) - \mathrm{AR}^* = \mathbb{E}\left[\max_{\|\boldsymbol{\delta}\| \leq \varepsilon}(f(\boldsymbol{x} + \boldsymbol{\delta}) - y)^2 - \max_{\|\boldsymbol{\delta}\| \leq \varepsilon}(h(\boldsymbol{U}(\boldsymbol{x} + \boldsymbol{\delta})) - y)^2\right]$$

$$\leq \mathbb{E}\left[\max_{\|\boldsymbol{\delta}\| \leq \varepsilon}\left\{(f(\boldsymbol{x} + \boldsymbol{\delta}) - y)^2 - (h(\boldsymbol{U}(\boldsymbol{x} + \boldsymbol{\delta})) - y)^2\right\}\right]$$

$$= \mathbb{E}\left[\max_{\|\boldsymbol{\delta}\| \leq \varepsilon}\left\{(f(\boldsymbol{x} + \boldsymbol{\delta}) - h(\boldsymbol{U}(\boldsymbol{x} + \boldsymbol{\delta})))\underbrace{(f(\boldsymbol{x} + \boldsymbol{\delta}) + h(\boldsymbol{U}(\boldsymbol{x} + \boldsymbol{\delta})) - 2y)}_{=:\mathcal{Z}}\right\}\right]$$

Let $\Pi_U W = (\Pi_U w_1, \ldots, \Pi_U w_N)^\top$. Then, we have the decompositions

$$f(x + \delta; a^*, W, b) = f(x + \delta; a^*, W, b) - f(x + \delta; a^*, \Pi_U W, b) + f(x + \delta; a^*, \Pi_U W, b),$$

and

$$\mathcal{Z} = f(x + \delta; a^*, W, b) - f(x + \delta; a^*, \Pi_U W, b) + f(x + \delta; a^*, \Pi_U W, b) - h(U(x + \delta)) \\ + 2h(U(x + \delta)) - 2y.$$

Plugging this decomposition into the above and using the Cauchy-Schwartz inequality yields

$$\mathrm{AR}(a^*, W, b) - \mathrm{AR}^* \leq (\sqrt{\mathcal{E}_1} + \sqrt{\mathcal{E}_2})^2 + \sqrt{\mathcal{E}_3(\mathcal{E}_1 + \mathcal{E}_2)}, \tag{D.1}$$

where

$$\mathcal{E}_1 := \mathbb{E}\left[\max_{\|\delta\| \leq \varepsilon} (f(x + \delta; a^*, \Pi_U W, b) - h(U(x + \delta)))^2\right], \tag{D.2}$$

$$\mathcal{E}_2 := \mathbb{E}\left[\max_{\|\delta\| \leq \varepsilon} (f(x + \delta; a^*, \Pi_U W, b) - f(x + \delta; a^*, W, b))^2\right], \tag{D.3}$$

$$\mathcal{E}_3 := 4\,\mathbb{E}\left[\max_{\|\delta\| \leq \varepsilon} (h(U(x + \delta)) - y)^2\right] = 4\mathrm{AR}^*. \tag{D.4}$$

Under Definition 3, we have a set of good neurons $S$ to work with. To continue, we introduce a similar subset of good neurons under Definition 2.

**Definition 23.** *Suppose the weights $W = (w_1, \ldots, w_N)^\top$ are obtained from the $\alpha$-DFL oracle of Definition 2. Fix a maximal $2\sqrt{2\zeta}$-packing of $\mathbb{S}^{k-1}$ with respect to the Euclidean norm, denoted by $(\bar{v}_i)_{i=1}^M$. Define $v_j := \frac{U w_j}{\|U w_j\|}$ for all $j \in [N]$, and*

$$S_i := \{j \in [N] : \|v_j - \bar{v}_i\| \leq \sqrt{2\zeta}\},$$

*for all $i \in [M]$. Note that $(S_i)$ are mutually exclusive. Define $S := \bigcup_{i=1}^M S_i$. By upper and lower bounds on the surface area of the spherical cap (see e.g. Wang et al. (2024a, Lemma F.11)), there are constants $c_k, C_k > 0$ such that $c_k(1/\zeta)^{(k-1)/2} \leq M \leq C_k(1/\zeta)^{(k-1)/2}$. Therefore, using Definition 2, we have $|S|/N \geq \Omega(\alpha)$.*

Note that when considering the $(\alpha,\beta)$-SFL oracle, we leave $S$ unchanged from Definition 3. In either case, for every $j \notin S$, we will choose $a_j^* = 0$. Then, we then have the following upper bound on $\mathcal{E}_2$.

**Lemma 24.** *Suppose $a_j^* = 0$ for $j \notin S$ and $\|a^*\| \leq \tilde{r}_a/\sqrt{|S|}$. Then,*

$$\mathbb{E}\left[\max_{\|\delta\| \leq \varepsilon} (f(x + \delta; a^*, \Pi_U W, b) - f(x + \delta; a^*, W, b))^2\right] \lesssim L_\sigma^2 C_{\bar{q}} \tilde{r}_a^2 (1 + r_b^{2(\bar{q}-1)} + \varepsilon^{2(\bar{q}-1)})(1 + \varepsilon^2)\zeta,$$

*where $C_{\bar{q}}$ is a constant only depending on $\bar{q}$.*

**Proof.** To be concise, we define $\tilde{x}_\delta := x + \delta$ and hide dependence on $a^*$ and $b$ in the following notation. By pseudo-Lipschitzness of $\sigma$,

$$f(\tilde{x}_\delta; \Pi_U W) - f(\tilde{x}_\delta; W) = \sum_{j \in S} a_j^* (\sigma(\langle \Pi_U w_j, \tilde{x}_\delta \rangle + b_j) - \sigma(\langle w_j, \tilde{x}_\delta \rangle + b_j))$$

$$\leq L_\sigma \sum_{j \in S} |a_j^*| (|\langle \Pi_U w_j, \tilde{x}_\delta \rangle + b_j|^{\bar{q}-1} + |\langle w_j, \tilde{x}_\delta \rangle + b_j|^{\bar{q}-1} + 1)|\langle \Pi_U w_j - w_j, \tilde{x}_\delta \rangle|.$$

Let

$$\mathcal{A}_j := |\langle \Pi_U w_j, \tilde{x}_\delta \rangle + b_j|^{\bar{q}-1} + |\langle w_j, \tilde{x}_\delta \rangle + b_j|^{\bar{q}-1} + 1,$$

and

$$\mathcal{B}_j := |\langle \Pi_U w_j - w_j, \tilde{x}_\delta \rangle|.$$

Then by the Cauchy-Schwartz inequality,

$$\mathcal{E}_2 \le L_\sigma^2 \, \mathbb{E}\left[\max_{\|\boldsymbol{\delta}\| \le \varepsilon} \Big(\sum_{j \in S} |a_j^*| \mathcal{A}_j \mathcal{B}_j\Big)^2\right] \le \frac{L_\sigma^2 \tilde{r}_a^2}{|S|} \, \mathbb{E}\left[\max_{\|\boldsymbol{\delta}\| \le \varepsilon} \sum_{j \in S} \mathcal{A}_j^2 \mathcal{B}_j^2\right]$$

$$\le \frac{L_\sigma^2 \tilde{r}_a^2}{|S|} \sum_{j \in S} \mathbb{E}\left[\max_{\|\boldsymbol{\delta}\| \le \varepsilon} \mathcal{A}_j^2 \mathcal{B}_j^2\right]$$

$$\le \frac{L_\sigma^2 \tilde{r}_a^2}{|S|} \sum_{j \in S} \mathbb{E}\left[\max_{\|\boldsymbol{\delta}\| \le \varepsilon} \mathcal{A}_j^4\right]^{1/2} \mathbb{E}\left[\max_{\|\boldsymbol{\delta}\| \le \varepsilon} \mathcal{B}_j^4\right]^{1/2}.$$

Additionally, we have

$$\max_{\|\boldsymbol{\delta}\| \le \varepsilon} \mathcal{A}_j \le C_{\bar{q}}\Big(|\langle \Pi_{\boldsymbol{U}} \boldsymbol{w}_j, \boldsymbol{x}\rangle|^{\bar{q}-1} + |\langle \boldsymbol{w}_j, \boldsymbol{x}\rangle|^{\bar{q}-1} + \varepsilon^{\bar{q}-1} + r_b^{\bar{q}-1} + 1\Big),$$

and

$$\max_{\|\boldsymbol{\delta}\| \le \varepsilon} \mathcal{B}_j \le \varepsilon \|\Pi_{\boldsymbol{U}} \boldsymbol{w}_j - \boldsymbol{w}_j\| + |\langle \Pi_{\boldsymbol{U}} \boldsymbol{w}_j - \boldsymbol{w}_j, \boldsymbol{x}\rangle|.$$

Further, by Assumption 2, for all $\boldsymbol{v} \in \mathbb{R}^d$, $\langle \boldsymbol{v}, \boldsymbol{x}\rangle$ is a centered subGaussian random variable with subGaussian norm $\mathcal{O}(\|\boldsymbol{v}\|)$, therefore $\mathbb{E}\big[|\langle \boldsymbol{v}, \boldsymbol{x}\rangle|^{\bar{q}}\big] \le C_{\bar{q}} \|\boldsymbol{v}\|^{\bar{q}}$ for all $\bar{q} > 0$. In summary,

$$\mathbb{E}\left[\max_{\|\boldsymbol{\delta}\| \le \varepsilon} \mathcal{A}_j^4\right]^{1/2} \le C_{\bar{q}}(1 + r_b^{2(\bar{q}-1)} + \varepsilon_1^{2(\bar{q}-1)}), \quad \text{and} \quad \mathbb{E}\left[\max_{\|\boldsymbol{\delta}\| \le \varepsilon} \mathcal{B}_j^4\right]^{1/2} \lesssim (1 + \varepsilon^2)\zeta,$$

where we used the fact that $\|\Pi_{\boldsymbol{U}} \boldsymbol{w}_j - \boldsymbol{w}_j\|^2 \le 2\zeta$ for all $j \in S$. This completes the proof. □

While the term $\mathcal{E}_1$ defined in (D.2) is an expectation over the entire distribution of $\boldsymbol{x}$, most approximation bounds support only a compact subset of $\mathbb{R}^d$. The following lemma shows that approximation on compact sets is sufficient to bound $\mathcal{E}_1$.

**Lemma 25.** *Suppose $a_j^* = 0$ for $j \notin S$ and $\|\boldsymbol{a}^*\| \le \tilde{r}_a/\sqrt{|S|}$. Further, suppose $r_z \ge 1 \vee 2\varepsilon$. Let*

$$\epsilon_{\mathrm{approx}} := \sup_{\|\boldsymbol{U}\boldsymbol{x}\| \le r_z} |f(\boldsymbol{x}; \boldsymbol{a}^*, \Pi_{\boldsymbol{U}} \boldsymbol{W}, \boldsymbol{b}) - h(\boldsymbol{U}\boldsymbol{x})|.$$

*Assume $h$ satisfies $|h(\boldsymbol{z})| \le L_h(1 + \|\boldsymbol{z}\|^p)$ for all $\boldsymbol{z} \in \mathbb{R}^k$ and some constant $p \ge 0$. Then,*

$$\mathcal{E}_1 \le \epsilon_{\mathrm{approx}}^2 + \Big(L_\sigma^2 C_{\bar{q}} \tilde{r}_a^2 (1 + \varepsilon^{2\bar{q}} + r_b^{2\bar{q}}) + L_h^2 C_{p,k}(1 + \varepsilon^{2p})\Big) e^{-\Omega(r_z^2)}.$$

**Proof.** For brevity, define

$$\Delta_{\boldsymbol{\delta}} := \big(f(\tilde{\boldsymbol{x}}_{\boldsymbol{\delta}}; \boldsymbol{a}^*, \Pi_{\boldsymbol{U}} \boldsymbol{W}, \boldsymbol{b}) - h(\boldsymbol{U}(\boldsymbol{x} + \boldsymbol{\delta})))^2$$

where $\tilde{\boldsymbol{x}}_{\boldsymbol{\delta}} := \boldsymbol{x} + \boldsymbol{\delta}$. Then,

$$\mathbb{E}\left[\max_{\|\boldsymbol{\delta}\| \le \varepsilon} \Delta_{\boldsymbol{\delta}}\right] \le \mathbb{E}\left[\max_{\|\boldsymbol{\delta}\| \le \varepsilon} \Delta_{\boldsymbol{\delta}} \mathbb{I}[\|\boldsymbol{U}\tilde{\boldsymbol{x}}_{\boldsymbol{\delta}}\| \le r_z]\right] + \mathbb{E}\left[\max_{\|\boldsymbol{\delta}\| \le \varepsilon} \Delta_{\boldsymbol{\delta}} \mathbb{I}[\|\boldsymbol{U}\tilde{\boldsymbol{x}}_{\boldsymbol{\delta}}\| > r_z]\right]$$

$$\le \epsilon_{\mathrm{approx}}^2 + \mathbb{E}\left[\max_{\|\boldsymbol{\delta}\| \le \varepsilon} \Delta_{\boldsymbol{\delta}}^2\right]^{1/2} \mathbb{E}\left[\max_{\|\boldsymbol{\delta}\| \le \varepsilon} \mathbb{I}[\|\boldsymbol{U}\tilde{\boldsymbol{x}}_{\boldsymbol{\delta}}\| > r_z]\right]^{1/2}$$

$$\le \epsilon_{\mathrm{approx}}^2 + \mathbb{E}\left[\max_{\|\boldsymbol{\delta}\| \le \varepsilon} \Delta_{\boldsymbol{\delta}}^2\right]^{1/2} \mathbb{P}(\|\boldsymbol{U}\boldsymbol{x}\| > r_z - \varepsilon)^{1/2}$$

$$\le \epsilon_{\mathrm{approx}}^2 + \mathbb{E}\left[\max_{\|\boldsymbol{\delta}\| \le \varepsilon} \Delta_{\boldsymbol{\delta}}^2\right]^{1/2} \mathbb{P}\Big(\|\boldsymbol{U}\boldsymbol{x}\| > \frac{r_z}{2}\Big)^{1/2}.$$

Furthermore, we have

$$\mathbb{E}\left[\max_{\|\boldsymbol{\delta}\| \le \varepsilon} \Delta_{\boldsymbol{\delta}}^2\right] \lesssim \mathbb{E}\left[\max_{\|\boldsymbol{\delta}\| \le \varepsilon} f(\tilde{\boldsymbol{x}}_{\boldsymbol{\delta}}; \boldsymbol{a}^*, \Pi_{\boldsymbol{U}} \boldsymbol{W}, \boldsymbol{b})^4\right] + \mathbb{E}\left[\max_{\|\boldsymbol{\delta}\| \le \varepsilon} h(\boldsymbol{U}(\boldsymbol{x} + \boldsymbol{\delta}))^4\right].$$

Recall the notation $\boldsymbol{v}_j := \frac{\boldsymbol{U}\boldsymbol{w}_j}{\|\boldsymbol{U}\boldsymbol{w}_j\|}$ and $\boldsymbol{z} := \boldsymbol{U}\boldsymbol{x}$. Then, by Cauchy-Schwartz and Jensen inequalities,

$$
\mathbb{E}\left[\max_{\|\boldsymbol{\delta}\|\leq\varepsilon} f(\tilde{\boldsymbol{x}}_{\boldsymbol{\delta}}; \boldsymbol{a}^*, \Pi_{\boldsymbol{U}}\boldsymbol{W}, \boldsymbol{b})^4\right] \leq \mathbb{E}\left[\max_{\|\boldsymbol{\delta}\|\leq\varepsilon} \|\boldsymbol{a}^*\|^4 \|\sigma(\Pi_{\boldsymbol{U}}\boldsymbol{W}(\boldsymbol{x}+\boldsymbol{\delta})+\boldsymbol{b})\|^4\right]
$$

$$
\leq \frac{\tilde{r}_a^4}{|S|} \mathbb{E}\left[\max_{\|\boldsymbol{\delta}\|\leq\varepsilon} \left(\sum_{j\in S} \sigma(\langle\boldsymbol{v}_j, \boldsymbol{z}+\boldsymbol{U}\boldsymbol{\delta}\rangle+b_j)^4\right)\right]
$$

$$
\leq \frac{\tilde{r}_a^4 L_\sigma^4 C_{\bar{q}}}{|S|} \mathbb{E}\left[\sum_{j\in S}\langle\boldsymbol{v}_j, \boldsymbol{z}\rangle^{4\bar{q}}+\varepsilon^{4\bar{q}}+r_b^{4\bar{q}}\right]
$$

$$
\leq C_{\bar{q}} L_\sigma^4 \tilde{r}_a^4(1+\varepsilon^{4\bar{q}}+r_b^{4\bar{q}}).
$$

Similarly we can prove

$$
\mathbb{E}\left[\max_{\|\boldsymbol{\delta}\|} h(\boldsymbol{U}(\boldsymbol{x}+\boldsymbol{\delta}))^4\right] \leq C_{p,k} L_h^4(1+\varepsilon^{4p}).
$$

In summary,

$$
\mathbb{E}\left[\max_{\|\boldsymbol{\delta}\|\leq\varepsilon} \Delta_{\boldsymbol{\delta}}^2\right]^{1/2} \lesssim C_{\bar{q}} L_\sigma^2 \tilde{r}_a^2(1+\varepsilon^{2\bar{q}}+r_b^{2\bar{q}}) + C_{p,k} L_h^2(1+\varepsilon^{2p}).
$$

Finally, the probability bound

$$
\mathbb{P}\left(\|\boldsymbol{U}\boldsymbol{x}\| \geq \frac{r_z}{2}\right) \leq e^{-\Omega(r_z^2)}
$$

follows from subGaussianity of $\boldsymbol{x}$ and the fact that $k = \mathcal{O}(1)$. $\qquad\square$

## D.1 Approximating Univariate Functions

In this section, we recall prior results on approximating univariate functions with random biases in the infinite-width regime under ReLU and polynomial activations.

**Lemma 26** (Damian et al. (2022, Lemma 9, Adapted)). *Let $\sigma$ be the ReLU activation, $a \sim \text{Unif}(\{-1, +1\})$, and $b \sim \text{Unif}(-r_b, r_b)$. Then, there exists $f : \{-1, +1\} \times [-r_b, r_b] \to \mathbb{R}$, such that for all $|z| \leq r_b$ we have*

$$
\mathbb{E}_{a,b}[2r_b f(b)\sigma(az+b)] = h(z).
$$

*Additionally, if $h$ is a polynomial of degree $s$, we have $\sup_{a,b}|f(a,b)| \leq r_b^{(s-2)\vee 0}$.*

**Proof.** From integration by parts, namely

$$
\mathbb{E}_{a,b}[2r_b(1-a)h''(b)\sigma(az+b)] = \int_z^{r_b} h''(b)(-z+b)\mathrm{d}b
$$

$$
= h'(r_b)(-z+r_b) - \int_z^{r_b} h'(b)\mathrm{d}b
$$

$$
= h'(r_b)(-z+r_b) + h(z) - h(r_b).
$$

Therefore, it remains to approximate the constant and linear parts. It is straightforward to verify that

$$
\mathbb{E}_{a,b}\left[\frac{6b}{r_b^2}\cdot\sigma(az+b)\right] = 1, \quad \mathbb{E}_{a,b}[2a\sigma(az+b)] = z.
$$

Thus, we let

$$
f(a,b) = (1-a)h''(b) + \frac{ah'(r_b)}{r_b} - \frac{3b(h'(r_b)r_b - h(r_b))}{r_b^3},
$$

which completes the proof. $\qquad\square$

Furthermore, we have the following result for infinite-width approximation with polynomial activations.

**Lemma 27** (Oko et al. (2024, Lemma 30, Adapted))**.** *Let $\sigma$ be a polynomial of degree $q$ and suppose $b \sim \text{Unif}(-r_b, r_b)$ and $h$ is a polynomial of degree $p$ such that $q \geq p$, and in particular satisfies $|h(z)| \leq L_h(1 + |z|^p)$. Suppose $r_b \geq q$. Then, there exists a function $f : [-r_b, r_b] \to \mathbb{R}$ such that*

$$\mathbb{E}_b[2r_b f(b)\sigma(z + b)] = h(z), \quad \forall z \in \mathbb{R}.$$

*Furthermore, we have $|f(z)| \leq C_{\sigma,h}$ for all $z$, where $C_{\sigma,h}$ only depends on the activation and $L_h$.*

**Proof.** In order for $\sigma$ to approximate arbitrary polynomials of degree at most $q$, it is sufficient to show that $\sigma$ can approximate at least one polynomial per degree, ranging from degree 0 to $q$. Defining the corresponding polynomial with degree $i$ as $g_i(z)$, then $h$ will be in the span of $\{g_i\}_{i=0}^q$. More specifically, suppose $h(z) = \sum_{j=0}^p \alpha_j z^j$, and $g_i(z) = \sum_{j=0}^i \gamma_{i,j} z^j$. Then there exist $\{\beta_i\}_{i=0}^q$ such that

$$\sum_{i=0}^p \beta_i g_i(z) = \sum_{j=0}^p \sum_{i=j}^p \gamma_{i,j}\beta_i z^j = \sum_{j=0}^p \alpha_j z^j.$$

Indeed, we can let $\beta_i = 0$ for all $i > p$. Additionally, note that $\gamma_{i,i} \neq 0$ for all $i \leq q$ by definition. Therefore, the solution to the above equation is given iteratively by $\beta_p = \alpha_p/\gamma_{p,p}$ and

$$\beta_{p-j} = \frac{\alpha_{p-j} - \sum_{i=0}^{j-1} \gamma_{p-i,p-j}\beta_{p-i}}{\gamma_{p-j,p-j}},$$

for $1 \leq j \leq p$. Importantly, $|\beta_i|$ for all $i$ can be bounded polynomially by $\{\alpha_j\}_j$, $\{\gamma_{i,j}\}_{i,j}$ and $\{\gamma_{i,i}^{-1}\}_i$. Further, $|\alpha_i|$ can be bounded polynomially by $L_h$ for all $i$. Thus, it remains to construct $\{g_i\}$.

Following Oko et al. (2024), we define

$$g_q(z) = \int_{-q}^0 \sigma(z + b)\mathrm{d}b.$$

It is straightforward to verify that $g_q$ has degree (exactly) $q$. We then iteratively define

$$g_{q-i}(z) = g_{q-(i-1)}(z+1) - g_{q-(i-1)}(z), \quad \forall 1 \leq i \leq q.$$

Using the definition above and by induction, one can verify $g_i$ has degree exactly $i$. Furthermore, expanding the definition above yields

$$g_{q-i}(z) = \sum_{j=0}^i c_{i,j} g_q(z+j) = \sum_{j=0}^i c_{i,j} \int_{-q}^0 \sigma(z+b+j)\mathrm{d}b,$$

where $c_{i,j} = (-1)^{i-j}\binom{i}{j}$, i.e. the coefficients that satisfy $(z-1)^i = \sum_{j=0}^i c_{i,j} z^j$. In particular, we can write

$$g_{q-i}(z) = \sum_{j=0}^i c_{i,j} \int_{-q+j}^j \sigma(z+b)\mathrm{d}b = \mathbb{E}_b\left[2r_b \sum_{j=0}^i \mathbb{I}[-q+j \leq b \leq j]\sigma(z+b)\right].$$

Therefore, we can define

$$f(b) := \sum_{i=0}^q \beta_{q-i} \sum_{j=0}^i c_{i,j}\mathbb{I}[-q+j \leq b \leq j],$$

which completes the proof. $\qquad\square$

### D.2 APPROXIMATING MULTIVARIATE POLYNOMIALS

We adapt the approximation result of this section from Damian et al. (2022), modifying the proof to be consistent with our assumption on the first layer weights.

First, we remark that for any fixed $v \in \mathbb{S}^{k-1}$ and any degree $0 \leq s \leq p$, we can approximate the function $z \mapsto \langle v, z \rangle^s$ with random biases as established by Lemma 26 for the ReLU activation and

Lemma 27 for the polynomial activation. Therefore, our main effort will be spent in approximating a polynomial $h(\boldsymbol{z})$ using monomials $\langle \boldsymbol{v}, \boldsymbol{z} \rangle^s$. Note that we can represent $h$ by

$$h(\boldsymbol{z}) = \sum_{s=0}^{p} \boldsymbol{T}^{(s)}[\boldsymbol{z}^{\otimes s}],$$

where $\boldsymbol{T}^{(s)}$ is a symmetric tensor of order $s$, and we use the notation

$$\boldsymbol{T}^{(s)}[\boldsymbol{z}^{\otimes s}] = \operatorname{vec}(\boldsymbol{T}^{(s)})^\top \operatorname{vec}(\boldsymbol{z}^{\otimes s}) = \sum_{i_1,\dots,i_s=1}^{k} \boldsymbol{T}^{(s)}_{i_1,\dots,i_s} \boldsymbol{z}_{i_1} \dots \boldsymbol{z}_{i_s}.$$

The approximation result relies on the following fact.

**Lemma 28.** *Let $\boldsymbol{v} \sim \tau_k$. Then, the matrix $\mathbb{E}_{\boldsymbol{v} \sim \tau_k}\big[\operatorname{vec}(\boldsymbol{v}^{\otimes s})\operatorname{vec}(\boldsymbol{v}^{\otimes s})^\top\big]$ is invertible.*

**Proof.** Let $\boldsymbol{T}$ be an arbitrary symmetric tensor of order $s$ with $\|\boldsymbol{T}\|_{\mathrm{F}} = 1$. We need to find a constant $c_{s,k} > 0$ such that

$$\operatorname{vec}(\boldsymbol{T})^\top \mathbb{E}_{\boldsymbol{v} \sim \tau_k}\big[\operatorname{vec}(\boldsymbol{v}^{\otimes s})\operatorname{vec}(\boldsymbol{v}^{\otimes s})\big]\operatorname{vec}(\boldsymbol{T}) \geq c_{s,k}.$$

Note that

$$\operatorname{vec}(\boldsymbol{T})^\top \mathbb{E}_{\boldsymbol{v} \sim \tau_k}\big[\operatorname{vec}(\boldsymbol{v}^{\otimes s})\operatorname{vec}(\boldsymbol{v}^{\otimes s})\big]\operatorname{vec}(\boldsymbol{T}) = \mathbb{E}_{\boldsymbol{v} \sim \tau_k}\big[\boldsymbol{T}[\boldsymbol{v}^{\otimes s}]^2\big] = \mathbb{E}_{\boldsymbol{w} \sim \mathcal{N}(0,\mathbf{I}_k)}\left[\frac{\boldsymbol{T}[\boldsymbol{w}^{\otimes s}]^2}{\|\boldsymbol{w}\|^{2s}}\right].$$

Furthermore, (Damian et al., 2022, Lemma 23) implies that

$$\mathbb{E}_{\boldsymbol{w} \sim \mathcal{N}(0,\mathbf{I}_k)}\big[\boldsymbol{T}[\boldsymbol{w}^{\otimes s}]^2\big] \geq c'_{s,k},$$

for some constant $c'_{s,k} > 0$. Therefore, for any $r > 0$, we have

$$\mathbb{E}_{\boldsymbol{w} \sim \mathcal{N}(0,\mathbf{I}_k)}\big[\boldsymbol{T}[\boldsymbol{w}^{\otimes s}]^2 \mathbb{I}[\|\boldsymbol{w}\| > r]\big] + \mathbb{E}_{\boldsymbol{w} \sim \mathcal{N}(0,\mathbf{I}_k)}\big[\boldsymbol{T}[\boldsymbol{w}^{\otimes s}]^2 \mathbb{I}[\|\boldsymbol{w}\| \leq r]\big] \geq c'_{s,k}.$$

Note that the first term on the LHS above can become arbitrarily small by choosing $r$ sufficiently large (depending on $s$ and $k$). Thus for sufficiently large $r$ we have

$$\mathbb{E}_{\boldsymbol{w} \sim \mathcal{N}(0,\mathbf{I}_k)}\big[\boldsymbol{T}[\boldsymbol{w}^{\otimes s}]^2 \mathbb{I}[\|\boldsymbol{w}\| \leq r]\big] \geq \frac{c'_{s,k}}{2}.$$

Finally, we have

$$\mathbb{E}_{\boldsymbol{w} \sim \mathcal{N}(0,\mathbf{I}_k)}\left[\frac{\boldsymbol{T}[\boldsymbol{w}^{\otimes s}]^2}{\|\boldsymbol{w}\|^{2s}}\right] \geq \frac{1}{r^{2s}} \mathbb{E}_{\boldsymbol{w} \sim \mathcal{N}(0,\mathbf{I}_k)}\big[\boldsymbol{T}[\boldsymbol{w}^{\otimes s}]^2 \mathbb{I}[\|\boldsymbol{w}\| \leq r]\big] \geq \frac{c'_{s,k}}{2r^{2s}}.$$

Therefore, taking $c_{s,k} = \frac{c'_{s,k}}{2r^{2s}}$ completes the proof. $\qquad\square$

The following lemma establishes how we can use monomials of the form $(\boldsymbol{v}^\top \boldsymbol{z})^s$ to approximate each term appearing in $h(\boldsymbol{z})$.

**Lemma 29** (Damian et al. (2022, Corollary 4, Adapted)). *There exists $f : \mathbb{S}^{k-1} \to \mathbb{R}$ such that for all $\boldsymbol{z} \in \mathbb{R}^k$ and non-negative integers $s \geq 0$,*

$$\int_{\mathbb{S}^{k-1}} f(\boldsymbol{v})\langle \boldsymbol{v}, \boldsymbol{z} \rangle^s \mathrm{d}\tau_k(\boldsymbol{v}) = \boldsymbol{T}^{(s)}[\boldsymbol{z}^{\otimes s}].$$

*Further, $|f(\boldsymbol{v})| \leq C_{k,s}\big\|\boldsymbol{T}^{(s)}\big\|_{\mathrm{F}}$ for all $\boldsymbol{v} \in \mathbb{S}^{k-1}$.*

**Proof.** Note that by definition, $\langle \boldsymbol{v}, \boldsymbol{z} \rangle^s = \operatorname{vec}(\boldsymbol{v}^{\otimes s})^\top \operatorname{vec}(\boldsymbol{z}^{\otimes s})$. Therefore,

$$\int f(\boldsymbol{v})\langle \boldsymbol{v}, \boldsymbol{z} \rangle^s \mathrm{d}\tau_k(\boldsymbol{v}) = \left(\int f(\boldsymbol{v})\operatorname{vec}(\boldsymbol{v}^{\otimes s})\mathrm{d}\tau_k(\boldsymbol{v})\right)^\top \operatorname{vec}(\boldsymbol{z}^{\otimes s}).$$

We need to match the first vector on the RHS above with $\operatorname{vec}(\boldsymbol{T}^{\otimes s})$, thus our choice of $f$ is

$$f(\boldsymbol{v}) = \operatorname{vec}(\boldsymbol{v}^{\otimes s})^\top \mathbb{E}_{\boldsymbol{v} \sim \tau_k}\big[\operatorname{vec}(\boldsymbol{v}^{\otimes s})\operatorname{vec}(\boldsymbol{v}^{\otimes s})^\top\big]^{-1}\operatorname{vec}(\boldsymbol{T}^{(s)}).$$

The proof is then completed via the lower bound of Lemma 28 which gaurantees the existence of some constant $c_{s,k} > 0$ such that $\lambda_{\min}\big(\mathbb{E}_{\boldsymbol{v} \sim \tau_k}\big[\operatorname{vec}(\boldsymbol{v}^{\otimes s})\operatorname{vec}(\boldsymbol{v}^{\otimes s})^\top\big]\big) \geq c_{s,k}$. $\qquad\square$

The above result along with the univariate approximations proved earlier immediately yields the following corollary.

**Corollary 30.** *Suppose $h$ is a polynomial of degree $p$ denoted by $h(z) = \sum_{s=0}^{p} \boldsymbol{T}^{(s)}[\boldsymbol{z}^{\otimes s}]$. Further assume the activation $\sigma$ is either ReLU or a polynomial of degree $q \geq p$. Then, there exists $\hat{h} : \mathbb{S}^{k-1} \times [-r_b, r_b] \to \mathbb{R}$ such that for every $\|\boldsymbol{z}\| \leq r_b$, we have*

$$\int_{\mathbb{S}^{k-1} \times [-r_b, r_b]} \hat{h}(\boldsymbol{v}, b)\sigma(\langle \boldsymbol{v}, \boldsymbol{z} \rangle + b)\mathrm{d}\tau_k(\boldsymbol{v})\mathrm{d}b = h(\boldsymbol{z}).$$

*Furthermore, $\left|\hat{h}(\boldsymbol{v}, b)\right| \leq C_{k,q} \max_{s \leq p}\left\|\boldsymbol{T}^{(s)}\right\|_{\mathrm{F}}$ for the polynomial activation and $\left|\hat{h}(\boldsymbol{v}, b)\right| \leq C_k r_b^{(p-2)\vee 0}\left\|\boldsymbol{T}^{(s)}\right\|_{\mathrm{F}}$ for the ReLU activation.*

**Proof.** First, we consider the case where we use polynomial activations. Let

$$\hat{h}(\boldsymbol{v}, b) = \sum_{s=0}^{p} f_{1,s}(\boldsymbol{v})f_{2,s}(b),$$

for $(f_{1,s})$ and $(f_{2,s})$ which we now determine. We choose $f_{2,s}$ according to and Lemma 27, then

$$\int_{b=-r_b}^{r_b} f_{2,s}(b)\sigma(\langle \boldsymbol{v}, \boldsymbol{z} \rangle + b)\mathrm{d}b = \langle \boldsymbol{v}, \boldsymbol{z} \rangle^s,$$

for all $\|\boldsymbol{z}\| \leq r_b/2$, and $|f_{2,s}(b)| \leq C_{s,q}$ for all $b$. Then, we choose $f_{1,s}$ according to Lemma 29, which yields

$$\int_{\mathbb{S}^{k-1} \times [-r_b, r_b]} f_{1,s}(\boldsymbol{v})f_{2,s}(b)\sigma(\langle \boldsymbol{v}, \boldsymbol{z} \rangle + b)\mathrm{d}\tau_k(\boldsymbol{v})\mathrm{d}b = \int f_{1,s}(\boldsymbol{v})\langle \boldsymbol{v}, \boldsymbol{z} \rangle^s \mathrm{d}\tau_k(\boldsymbol{v}) = \boldsymbol{T}^{(s)}[\boldsymbol{z}^{\otimes s}],$$

for all $\|\boldsymbol{z}\| \leq r_b/2$. Additionally $|f_{1,s}(\boldsymbol{v})| \leq C_{s,k}\left\|\boldsymbol{T}^{(s)}\right\|_{\mathrm{F}}$, which completes the proof of the polynomial activation case.

Now, consider the case where we use the ReLU activation. Let

$$\hat{h}(\boldsymbol{v}, b) = \sum_{s=0}^{p} g_s(\boldsymbol{v}, b).$$

where

$$g_s(\boldsymbol{v}, b) = \frac{1}{2}f_{1,s}(\boldsymbol{v})\tilde{f}_{2,s}(1, b) + \frac{1}{2}f_{1,s}(-\boldsymbol{v})\tilde{f}_{2,s}(-1, b)$$

with $f_{1,s}$ given above and $\tilde{f}_{2,s}$ introduced below. Since $\boldsymbol{v}$ and $-\boldsymbol{v}$ have the same distribution, we have

$$\int g_s(\boldsymbol{v}, b)\sigma(\langle \boldsymbol{v}, \boldsymbol{z} \rangle)\mathrm{d}b\,\mathrm{d}\tau_k(\boldsymbol{v}) = \int \frac{1}{2}\Big(f_{1,s}(\boldsymbol{v})\tilde{f}_{2,s}(1, b) + f_{1,s}(-\boldsymbol{v})\tilde{f}_{2,s}(-1, b)\Big)\sigma(\langle \boldsymbol{v}, \boldsymbol{z} \rangle \mathrm{d}b\,\mathrm{d}\tau_k(\boldsymbol{v})$$

$$= \int_{\mathbb{S}^{d-1}} f_{1,s}(\boldsymbol{v})\frac{1}{2}\left\{\int_{b=-r_b}^{r_b} \tilde{f}_{2,s}(1, b)\sigma(\langle \boldsymbol{v}, \boldsymbol{z} \rangle + b) + \tilde{f}_{2,s}(-1, b)\sigma(-\langle \boldsymbol{v}, \boldsymbol{z} \rangle + b)\mathrm{d}b\right\}\mathrm{d}\tau_k(\boldsymbol{v})$$

$$= \int f_{1,s}(\boldsymbol{v})\langle \boldsymbol{v}, \boldsymbol{z} \rangle^s \mathrm{d}\tau_k(\boldsymbol{v}) = \boldsymbol{T}^{(s)}[\boldsymbol{z}^{\otimes s}].$$

As a result, it suffices to choose $\tilde{f}_{2,s}$ according to Lemma 26, which completes the proof of the corollary.

$\square$

As a last step in this section, we verify that one can indeed control $\max_{s \leq p}\left\|\boldsymbol{T}^{(s)}\right\|_{\mathrm{F}}$ with an absolute constant when $h$ is the minimizer of the adversarial risk.

**Lemma 31.** *Suppose $\mathcal{F}$ is the class of degree $p$ polynomials on $\mathbb{R}^d$. Let $\mathcal{H} = \{\boldsymbol{z} \mapsto \mathbb{E}[f(\boldsymbol{x}) \,|\, \boldsymbol{U}\boldsymbol{x} = \boldsymbol{z}] : f \in \mathcal{F}\}$, and define*

$$h = \arg\min_{h' \in \mathcal{H}} \mathbb{E}\left[\max_{\|\boldsymbol{\delta}\| \leq \varepsilon} (h'(\boldsymbol{U}(\boldsymbol{x} + \boldsymbol{\delta})) - y)^2\right].$$

*Denote the decomposition of $h$ by $h(\boldsymbol{z}) = \sum_{s=0}^{p} \boldsymbol{T}^{(s)}[\boldsymbol{z}^{\otimes s}]$. Then, $\left\|\boldsymbol{T}^{(s)}\right\|_{\mathrm{F}} \leq C_{k,y}$, where $C_{k,y}$ is a constant depending only on $k$ and the target second moment $\mathbb{E}[y^2]$ (thus an absolute constant in our setting). As a consequence, we have $|h(\boldsymbol{z})| \leq L_h(1 + \|\boldsymbol{z}\|^p)$ for all $\boldsymbol{z} \in \mathbb{R}^k$, where $L_h > 0$ is an absolute constant.*

**Proof.** By comparing with the zero function, we have

$$\mathbb{E}\big[(h(\boldsymbol{U}\boldsymbol{x}) - y)^2\big] \leq \mathbb{E}\Big[\max_{\|\boldsymbol{\delta}\|\leq\varepsilon}(h(\boldsymbol{U}(\boldsymbol{x} + \boldsymbol{\delta})) - y)^2\Big] \leq \mathbb{E}\big[y^2\big].$$

Furthermore, by the Cauchy-Schwartz inequality,

$$\mathbb{E}\big[(h(\boldsymbol{U}\boldsymbol{x}) - y)^2\big] \geq \mathbb{E}\big[h(\boldsymbol{U}\boldsymbol{x})^2\big] + \mathbb{E}\big[y^2\big] - 2\,\mathbb{E}\big[h(\boldsymbol{U}\boldsymbol{x})^2\big]^{1/2}\,\mathbb{E}\big[y^2\big]^{1/2}.$$

Combining the two inequalities above, we obtain $\mathbb{E}\big[h(\boldsymbol{U}\boldsymbol{x})^2\big] \leq 4\,\mathbb{E}\big[y^2\big]$. Let $\boldsymbol{z} := \boldsymbol{U}\boldsymbol{x}$, and let $\mu_{\boldsymbol{z}}$ be the marginal distribution of $\boldsymbol{z}$. Then

$$\mathbb{E}\big[h(\boldsymbol{z})^2\big] = \int h(\boldsymbol{z})^2 \frac{\mathrm{d}\mu_{\boldsymbol{z}}}{\mathrm{d}\mathcal{N}(0, C_k\mathbf{I}_k)}(\boldsymbol{z})\mathrm{d}\mathcal{N}(0, C_k\mathbf{I}_k)(\boldsymbol{z}).$$

Further, by subGaussianity of $\boldsymbol{x}$ and subsequent subGaussianity of $\boldsymbol{z}$, we have $\frac{\mathrm{d}\mu_{\boldsymbol{z}}}{\mathrm{d}\mathcal{N}(0, C_k)}(\boldsymbol{z}) \leq C'_k < \infty$ for all $\boldsymbol{z}$, when $C_k, C'_k$ are sufficiently large constants depending only on $k$. Therefore,

$$\mathbb{E}_{\boldsymbol{z}\sim\mathcal{N}(0, C_k\mathbf{I}_k)}\big[h(\boldsymbol{z})^2\big] \leq 4C'_k\,\mathbb{E}\big[y^2\big].$$

The proof is completed by using the Hermite decomposition of $h$. $\qquad\square$

### D.3 Approximating Multivariate Pseudo-Lipschitz Functions

We now turn to the more general problem of approximating pseudo-Lipschitz functions. Specifically, when $\mathcal{F}$ satisfies Assumption 3, functions of the form $h(\boldsymbol{z}) = \mathbb{E}[f(\boldsymbol{x})\,|\,\boldsymbol{U}\boldsymbol{x} = \boldsymbol{z}]$ will be $L$-pseudo-Lipschitz. The following lemma investigates approximating such functions with infinite-width two-layer neural networks.

**Lemma 32.** *Suppose $h : \mathbb{R}^k \to \mathbb{R}$ is $L$-Lipschitz on $\|\boldsymbol{z}\| \leq r_z$ and $\sigma$ is the ReLU activation. Then, for every $\Delta \geq C_k$, there exists $\hat{h} : \mathbb{S}^{k-1} \times [-r_b, r_b] \to \mathbb{R}$ such that*

$$\left|h(\boldsymbol{z}) - \int_{\mathbb{S}^{k-1}\times[-r_b,r_b]} \hat{h}(\boldsymbol{v}, b)\sigma(\langle\boldsymbol{v}, \boldsymbol{z}\rangle + b)\mathrm{d}\tau_k(\boldsymbol{v})\mathrm{d}b\right| \leq C_k L r_z\left\{\left(\frac{\Delta}{Lr_z}\right)^{\frac{-2}{k+1}}\log\frac{\Delta}{Lr_z} + \left(\frac{\Delta}{Lr_z}\right)^{\frac{2k}{k+1}}\left(\frac{r_z}{r_b}\right)^k\right\},$$

*for all $\|\boldsymbol{z}\| \leq r_z$. Furthermore, we have $\left|\hat{h}(\boldsymbol{v}, b)\right| \leq C_k L(\Delta/Lr_z)^{2k/(k+1)}/r_z$ for all $\boldsymbol{v}$ and $b$, and*

$$\int_{\mathbb{S}^{k-1}\times[-r_b,r_b]} \hat{h}(\boldsymbol{v}, b)^2\mathrm{d}\tau_k(\boldsymbol{v})\mathrm{d}b \leq \frac{C_k\Delta^2}{r_z^3}.$$

**Proof.** Let $\tilde{\boldsymbol{z}} := (\boldsymbol{z}^\top, r_z)^\top \in \mathbb{R}^{k+1}$. By (Bach, 2017, Proposition 6), we know that for all $\Delta \geq C_k$, there exists $p : \mathbb{S}^k \to \mathbb{R}$, such that $\|p\|_{L^2(\tau_{k+1})} \leq \Delta$ and

$$\left|h(\boldsymbol{z}) - \int_{\mathbb{S}^k} p(\tilde{\boldsymbol{v}})\sigma\Big(\frac{\langle\tilde{\boldsymbol{v}}, \tilde{\boldsymbol{z}}\rangle}{r_z}\Big)\mathrm{d}\tau_{k+1}(\tilde{\boldsymbol{v}})\right| \leq C_k L r_z\Big(\frac{\Delta}{Lr_z}\Big)^{\frac{-2}{k+1}}\log\frac{\Delta}{Lr_z},$$

for all $\|\boldsymbol{z}\| \leq r_z$. Furthermore, the proof of (Mousavi-Hosseini et al., 2024, Proposition 19) demonstrated that

$$|p(\tilde{\boldsymbol{v}})| \leq C_k L r_z\Big(\frac{\Delta}{Lr_z}\Big)^{\frac{2k}{k+1}}, \quad \forall\tilde{\boldsymbol{v}} \in \mathbb{S}^k.$$

Let $\tilde{\boldsymbol{v}} = (\tilde{\boldsymbol{v}}_{1:k}^\top, \tilde{v}_{k+1})^\top$ be the decomposition of $\tilde{\boldsymbol{v}}$ into its first $k$ and last coordinate. Then, we will use the fact that for $\tilde{\boldsymbol{v}} \sim \mathrm{Unif}(\mathbb{S}^k)$ when conditioned on $\tilde{v}_{k+1}$, by symmetry $\frac{\boldsymbol{v}_{1:k}}{\|\boldsymbol{v}_{1:k}\|}$ is uniformly distributed on $\mathbb{S}^{k-1}$. In other words, let $\boldsymbol{v} \sim \mathrm{Unif}(\mathbb{S}^{k-1})$ and $\tilde{b} \sim \rho_{k+1}$ independently, where we choose $\rho_{k+1}$ such that $\frac{\tilde{b}}{\sqrt{1+\tilde{b}^2}}$ has the same marginal distribution as $\tilde{v}_{k+1}$. Since the marginal distribution of $\tilde{v}_{k+1}$ is given by $\mathrm{d}\mathbb{P}(\tilde{v}_{k+1}) \propto (1 - \tilde{v}_{k+1}^2)^{(k-2)/2}\mathrm{d}\tilde{v}_{k+1}$, we have $\rho_{k+1}(\tilde{b}) = Z_k(1 + \tilde{b}^2)^{-(k+1)/2}$,

where $Z_k$ is the normalizing constant. Then, $\tilde{v} = \mathrm{T}(v, \tilde{b})$ is distributed uniformly on $\mathbb{S}^k$, where $\mathrm{T} : \mathbb{S}^{k-1} \times \mathbb{R} \to \mathbb{S}^k$ is given by $\mathrm{T}(v, \tilde{b}) = \frac{1}{\sqrt{1+\tilde{b}^2}}(v^\top, \tilde{b})$. As a result,

$$\int p(\tilde{v}) \sigma\Big(\frac{\langle \tilde{v}, \tilde{z} \rangle}{r_z}\Big) \mathrm{d}\tau_{k+1}(\tilde{v}) = \int p(\mathrm{T}(v, \tilde{b})) \sigma\Big(\frac{\langle v, z \rangle + \tilde{b} r_z}{r_z \sqrt{1+\tilde{b}^2}}\Big) \mathrm{d}\tau_k(v) \mathrm{d}\rho_{k+1}(\tilde{b})$$

$$= Z_k \int_{\mathbb{S}^{k-1} \times \mathbb{R}} \frac{p(\mathrm{T}(v, \tilde{b}))}{r_z \sqrt{1+\tilde{b}^2}} \cdot \frac{1}{(1+\tilde{b}^2)^{(k+1)/2}} \sigma(\langle v, z \rangle + \tilde{b} r_z) \mathrm{d}\tau_k(v) \mathrm{d}\tilde{b}$$

$$= Z_k \int_{\mathbb{S}^{k-1} \times \mathbb{R}} \frac{r_z^k p(\mathrm{T}(v, b/r_z))}{(r_z^2 + b^2)^{(k+2)/2}} \sigma(\langle v, z \rangle + b) \mathrm{d}\tau_k(v) \mathrm{d}b.$$

Therefore, our choice of $\hat{h}$ will be

$$\hat{h}(v, b) = Z_k \frac{r_z^k p(\mathrm{T}(v, b/r_z))}{(r_z^2 + b^2)^{(k+2)/2}}.$$

Next, we bound the following error term due to cutoff of bias,

$$\mathcal{E} := Z_k \left| \int_{\mathbb{S}^{k-1} \times (\mathbb{R} \setminus [-r_b, r_b])} \frac{r_z^k p(\mathrm{T}(v, b/r_z))}{(r_z^2 + b^2)^{(k+2)/2}} \sigma(\langle v, z \rangle + b) \mathrm{d}\tau_k(v) \mathrm{d}b \right|.$$

We have

$$\mathcal{E} \lesssim C_k L r_z \Big(\frac{\Delta}{L r_z}\Big)^{\frac{2k}{k+1}} \int_{|b| > r_b} \frac{r_z^k (r_z + |b|)}{(r_z^2 + b^2)^{(k+2)/2}} \mathrm{d}b$$

$$\lesssim C_k L r_z \Big(\frac{\Delta}{L r_z}\Big)^{\frac{2k}{k+1}} \int_{|b| > r_b} \frac{r_z^k}{(r_z^2 + b^2)^{(k+1)/2}} \mathrm{d}b$$

$$\lesssim C_k \Delta^{\frac{2k}{k+1}} \int_{|b| > r_b} \frac{r_z^k}{b^{k+1}} \mathrm{d}b$$

$$\lesssim C_k L r_z \Big(\frac{\Delta}{r_z}\Big)^{\frac{2k}{k+1}} \Big(\frac{r_z}{r_b}\Big)^k.$$

Finally, we prove the guarantees provided for $\hat{h}$. The uniform bound on $\left|\hat{h}(v, b)\right|$ follows directly by plugging in the uniform bound on $p$. For the $L^2$ bound on $\hat{h}$, we have

$$\int_{\mathbb{S}^{k-1} \times [-r_b, r_b]} \hat{h}(v, b)^2 \mathrm{d}\tau_k(v) \mathrm{d}b \leq \int_{\mathbb{S}^{k-1} \times \mathbb{R}} \hat{h}(v, b)^2 \mathrm{d}\tau_k(v) \mathrm{d}b$$

$$= \int \frac{Z_k^2 r_z^{2k} p(\mathrm{T}(v, b/r_z))^2}{(r_z^2 + b^2)^{k+2}} \mathrm{d}\tau_k(v) \mathrm{d}b$$

$$= \int \frac{Z_k^2 p(\mathrm{T}(v, \tilde{b}))^2}{r_z^3 (1 + \tilde{b}^2)^{k+2}} \mathrm{d}\tau_k(v) \mathrm{d}\tilde{b}$$

$$= \frac{Z_k}{r_z^3} \int \frac{p(\mathrm{T}(v, \tilde{b}))^2}{(1 + \tilde{b}^2)^{(k+3)/2}} \mathrm{d}\tau_k(v) \mathrm{d}\rho_{k+1}(\tilde{b})$$

$$= \frac{Z_k}{r_z^3} \int (1 - \tilde{v}_{k+1}^2)^{(k+3)/2} p(\tilde{v})^2 \mathrm{d}\tau_{k+1}(\tilde{v})$$

$$\leq \frac{Z_k \|p\|_{L^2(\tau_{k+1})}^2}{r_z^3} \leq \frac{Z_k \Delta^2}{r_z^3},$$

completing the proof. $\qquad\qquad\square$

### D.4 DISCRETIZING INFINITE-WIDTH APPROXIMATIONS

In this section, we provide finite-width guarantees corresponding to the infinite-width approximations proved earlier. Define the following integral operator

$$\mathcal{T}\hat{h}(\boldsymbol{z}) = \int_{\mathbb{S}^{k-1}\times[-r_b, r_b]} \hat{h}(\boldsymbol{v}, b)\sigma(\langle \boldsymbol{v}, \boldsymbol{z}\rangle + b)\mathrm{d}\tau_k(\boldsymbol{v})\mathrm{d}b. \tag{D.5}$$

The type of discretization error depends on whether we are using the $\alpha$-DFL or the $(\alpha,\beta)$-SFL oracle. We first cover the case of $\alpha$-DFL oracles.

**Proposition 33** (Approximation by Riemann Sum). *Suppose $\sigma$ satisfies* (B.1). *Let* $(\boldsymbol{w}_1, \ldots, \boldsymbol{w}_N)$ *be the first layer weights obtained from the $\alpha$-DFL oracle (Definition 2), and define* $\boldsymbol{v}_i = \frac{\boldsymbol{U}\boldsymbol{w}_i}{\|\boldsymbol{U}\boldsymbol{w}_i\|}$ *for* $i \in [N]$. *Suppose* $(b_j)_{j\in[N]} \overset{\text{i.i.d.}}{\sim} \mathrm{Unif}(-r_b, r_b)$, *and let* $\|\hat{h}\|_\infty := \sup_{\boldsymbol{v},b}\left|\hat{h}(\boldsymbol{v}, b)\right|$. *Then, there exists* $\boldsymbol{a}^*$ *such that* $a_j^* = 0$ *for* $j \notin S$ *and* $\left|a_j^*\right| \leq C_k\|\hat{h}\|_\infty r_b \log(\alpha N/(\zeta\delta))/(\alpha N)$ *for* $j \in S$ *(where $S$ is given by Definition 23), and*

$$\left|\sum_{j\in S} a_j^*\sigma(\langle \boldsymbol{v}_j, \boldsymbol{z}\rangle + b_j) - \mathcal{T}\hat{h}(\boldsymbol{z})\right| \leq C_{\bar{q}}\|\hat{h}\|_\infty L_\sigma r_b^{\bar{q}}\left(r_z\sqrt{\zeta} + \frac{r_b\log(N/\delta)}{\zeta^{(k-1)/2}\alpha N}\right), \tag{D.6}$$

*for all $\boldsymbol{z} \in \mathbb{R}^k$ where $\|\boldsymbol{z}\| \leq r_z \leq r_b$, with probability at least $1 - \delta$ over the randomness of biases.*

**Proof.** The proof is a multivariate version of the argument given in (Oko et al., 2024, Lemma 29). Let $\{\bar{\boldsymbol{v}}_i\}_{i=1}^M$ be the maximal $2\sqrt{2\zeta}$-packing of $\mathbb{S}^{k-1}$ from Definition 23, which is also a $2\sqrt{2\zeta}$-covering of $\mathbb{S}^{k-1}$. Recall from Definition 23 that $M \leq C_k(\frac{1}{\zeta})^{(k-1)/2}$.

For every $i \in [M]$, define
$$S_i := \{j \in [N], \|\boldsymbol{v}_j - \bar{\boldsymbol{v}}_i\| \leq \sqrt{2\zeta}\}.$$
Note that by definition of packing and Definition 2, each $\boldsymbol{v}_j$ can only belong to exactly one of $S_i$ when $j \in S$, meaning that $(S_i)$ are disjoint and $\bigcup_{i\in[M]} S_i = S$. In particular, $|S_i|/N \geq \zeta^{(k-1)/2}\alpha$, and $|S_i|/N \leq 1/M \leq \zeta^{(k-1)/2}/c_k$.

We want each group of biases $(b_j)_{j\in S_i}$ to cover the interval $[-r_b, r_b]$. We divide this interval into $2A$ subintervals of the form $[-r_b(1 + \frac{l}{A}), r_b(1 + \frac{l+1}{A}))$ for $0 \leq l \leq 2A - 1$. When $b_j \overset{\text{i.i.d.}}{\sim} \mathrm{Unif}(-r_b, r_b)$, by a union bound, the probability that there exists some subinterval and some $S_i$ such that the subinterval contains no element of $\{b_j : j \in S_i\}$ is at most $2A\sum_{i=1}^M (1 - \frac{1}{2A})^{|S_i|}$. Thus, taking $A \leq \lfloor\frac{|S_i|}{2\log(|S_i|M/\delta)}\rfloor$ for all $i \in [M]$ guarantees that all subintervals have at least one bias from every $S_i$ inside them with probability at least $1 - \delta$.

Next, we define $\Pi_1 : \mathbb{S}^{k-1} \to \mathbb{S}^{k-1}$ as the projection onto the packing, i.e. $\Pi_1(\boldsymbol{v}) = \arg\min_{\{\bar{\boldsymbol{v}}_i : i\in[M]\}}\|\boldsymbol{v} - \bar{\boldsymbol{v}}_i\|$. Further, we define $\Pi_2 : [M] \times [-r_b, r_b] \to [-r_b, r_b]$ by $\Pi_2(i, b) = \arg\min_{\{b_j : j\in S_i\}}|b - b_j|$. Tie braking can be performed by choosing any of the answers. By definition, we have $\|\boldsymbol{v} - \Pi_1(\boldsymbol{v})\| \leq 2\sqrt{2\zeta}$, and additionally $|b - \Pi_2(i, b)| \leq r_b/A$ for all $i \in [M]$ on the event described above.

We are now ready to construct $\boldsymbol{a}^*$. Specifically, let

$$a_j^* = \begin{cases} \int\int \hat{h}(\boldsymbol{v}, b)\mathbb{I}[i = \Pi_1(\boldsymbol{v}), b_j = \Pi_2(i, b)]\mathrm{d}\tau_k(\boldsymbol{v})\mathrm{d}b & \text{if } j \in S_i \text{ for some } i, \\ 0 & \text{if } j \notin S. \end{cases}$$

Note that by definition,

$$\sum_{i=1}^M \sum_{j\in S_i} \mathbb{I}[i = \Pi_1(\boldsymbol{v}), b_j = \Pi_2(i, b)] = 1,$$

For conciseness, we define $E(\boldsymbol{v}, i, j) = \mathbb{I}[i = \Pi_1(\boldsymbol{v}), b_j = \Pi_2(i, b)]$. When $j \in S_i$, on the event $E(\boldsymbol{v}, i, j)$ we have

$$\|\boldsymbol{v} - \boldsymbol{v}_j\| \leq \|\boldsymbol{v} - \bar{\boldsymbol{v}}_i\| + \|\bar{\boldsymbol{v}}_i - \boldsymbol{v}_j\| \leq 3\sqrt{2\zeta}.$$

Moreover, since $\mathbb{P}_{\boldsymbol{v} \sim \tau_k}[\|\boldsymbol{v} - \bar{\boldsymbol{v}}_i\| \leq 2\sqrt{2\zeta}] \leq C_k \zeta^{(k-1)/2}$, for $j \in S$ we have

$$|a_j^*| \leq \frac{C_k \|\hat{h}\|_\infty \zeta^{(k-1)/2} r_b}{A} \leq \frac{C_k \|\hat{h}\|_\infty r_b \log(N/\delta)}{\alpha N}.$$

As a result,

$$\left| \sum_{j \in S} a_j^* \sigma(\langle \boldsymbol{v}_j, \boldsymbol{z} \rangle + b_j) - \mathcal{T}\hat{h}(z) \right| = \left| \sum_{j \in S} a_j^* \sigma(\langle \boldsymbol{v}_j, \boldsymbol{z} \rangle + b_j) - \int \hat{h}(\boldsymbol{v}, b) \sigma(\langle \boldsymbol{v}, \boldsymbol{z} \rangle + b) \mathrm{d}\tau_k(\boldsymbol{v}) \mathrm{d}b \right|$$

$$= \left| \sum_{i=1}^{M} \sum_{j \in S_i} \int \hat{h}(\boldsymbol{v}, b) E(\boldsymbol{v}, i, j)(\sigma(\langle \boldsymbol{v}_j, \boldsymbol{z} \rangle + b_j) - \sigma(\langle \boldsymbol{v}, \boldsymbol{z} \rangle + b)) \mathrm{d}\tau_k(\boldsymbol{v}) \mathrm{d}b \right|$$

$$\lesssim C_{\bar{q}} \|\hat{h}\|_\infty L_\sigma r_b^{\bar{q}} (r_z \sqrt{\zeta} + \frac{r_b}{A}),$$

for all $\|\boldsymbol{z}\| \leq r_z$, where we used the fact that $\sigma(z)$ is $\mathcal{O}(L_\sigma r_b^{\bar{q}-1})$ Lipschitz when restricted to $|z| \leq r_b$. This concludes the proof. $\qquad\square$

Next, we provide a discretization guarantee when using $(\alpha,\beta)$-SFL oracles.

**Proposition 34.** *Consider the same setting as Proposition 33, except the first-layer weights $(\boldsymbol{w}_1, \ldots, \boldsymbol{w}_N)$ are obtained from the $(\alpha,\beta)$-SFL oracle (Definition 3). Then, there exists $\boldsymbol{a}^*$ such that $a_i^* = 0$ for $i \notin S$ and $|a_i^*| \leq \|\hat{h}\|_\infty r_b/(\beta\alpha N)$ for $i \in S$, and*

$$\left| \sum_{j \in S} a_j^* \sigma(\langle \boldsymbol{v}_j, \boldsymbol{z} \rangle + b_j) - \mathcal{T}\hat{h}(\boldsymbol{z}) \right| \leq \frac{C_{\bar{q}} L_\sigma \|\hat{h}\|_\infty r_b^{\bar{q}+1}}{\beta} \sqrt{\frac{\log(\alpha N/\delta)}{\alpha N}},$$

*for all $\boldsymbol{z} \in \mathbb{R}^k$ with $\|\boldsymbol{z}\| \leq r_b$, with probability at least $1 - \delta$ over the randomness of $(\boldsymbol{v}_i, b_i)_{i \in [N]}$. Moreover, suppose $\mathbb{E}_{\boldsymbol{v}, b \sim \tau_k \otimes \mathrm{Unif}(-r_b, r_b)} \left[ \hat{h}(\boldsymbol{v}, b)^2 \right] \leq M_2(\hat{h})^2$. Then, assuming $\frac{\mathrm{d}\mu}{\mathrm{d}\tau_k} \leq \beta'$, we have*

$$\|\boldsymbol{a}^*\|^2 \lesssim \frac{r_b^2 \beta' M_2(\hat{h})^2}{\alpha \beta^2 N}, \qquad \text{provided that,} \qquad N \gtrsim \frac{\|\hat{h}\|_\infty^4 \log(1/\delta)}{\alpha \beta'^2 M_2(\hat{h})^4},$$

*which also holds with probability at least $1 - \delta$.*

**Proof.** By definition,

$$\mathcal{T}\hat{h}(\boldsymbol{z}) = \int_{\mathbb{S}^{k-1} \times [-r_b, r_b]} \hat{h}(\boldsymbol{v}, b) \sigma(\langle \boldsymbol{v}, \boldsymbol{z} \rangle + b) \mathrm{d}\tau_k(\boldsymbol{v}) \mathrm{d}b$$

$$= \int_{\mathbb{S}^{k-1} \times [-r_b, r_b]} \hat{h}(\boldsymbol{v}, b) \frac{\mathrm{d}\tau_k}{\mathrm{d}\mu}(\boldsymbol{v}) \sigma(\langle \boldsymbol{v}, \boldsymbol{z} \rangle + b) \mathrm{d}\mu(\boldsymbol{v}) \mathrm{d}b$$

$$= \mathbb{E}_{\boldsymbol{v}, b \sim \mu \otimes \mathrm{Unif}(-r_b, r_b)} \left[ 2r_b \hat{h}(\boldsymbol{v}, b) \frac{\mathrm{d}\tau_k}{\mathrm{d}\mu}(\boldsymbol{v}) \sigma(\langle \boldsymbol{v}, \boldsymbol{z} \rangle + b) \right].$$

Consider $(\boldsymbol{v}_i, b_i)_{i \in S} \overset{\text{i.i.d.}}{\sim} \mu \otimes \mathrm{Unif}(-r_b, r_b)$ from Definition 3. Let

$$a_i^* = \begin{cases} \frac{2r_b \hat{h}(\boldsymbol{v}_i, b_i)}{|S|} \frac{\mathrm{d}\tau_k}{\mathrm{d}\mu}(\boldsymbol{v}_i) & \text{if } i \in S, \\ 0 & \text{if } i \notin S. \end{cases}$$

Consequently

$$|a_i^*| \leq \frac{2r_b \|\hat{h}\|_\infty}{\beta|S|},$$

for all $i \in S$. Given $\boldsymbol{z}$, define the random variable

$$\hat{\mathcal{T}}\hat{h}(\boldsymbol{z}) = \sum_{i \in S} a_i^* \sigma(\langle \boldsymbol{v}_i, \boldsymbol{z} \rangle + b_i).$$

Our next step is to bound the difference between $\hat{\mathcal{T}}\hat{h}(\boldsymbol{z})$ and $\mathcal{T}\hat{h}(\boldsymbol{z})$ uniformly over all $\|\boldsymbol{z}\| \leq r_b$. Let $(\hat{\boldsymbol{z}}_j)_{j=1}^M$ be a $\Delta$-covering of $\{\boldsymbol{z} : \|\boldsymbol{z}\| \leq r_b\}$, therefore $M \leq (3r_b/\Delta)^k$. Note that for any fixed $\boldsymbol{z}$ with $\|\boldsymbol{z}\| \leq r_b$, we have $\left|\hat{\mathcal{T}}\hat{h}(\boldsymbol{z})\right| \lesssim \|\hat{h}\|_\infty L_\sigma r_b^{\bar{q}+1}/\beta$. Thus, by Hoeffding's lemma,

$$\left|\hat{\mathcal{T}}\hat{h}(\boldsymbol{z}) - \mathcal{T}\hat{h}(\boldsymbol{z})\right| \lesssim \frac{\|\hat{h}\|_\infty L_\sigma r_b^{\bar{q}+1}}{\beta}\sqrt{\frac{\log(1/\delta)}{|S|}},$$

with probability at least $1 - \delta$ for a fixed $\boldsymbol{z}$. By a union bound,

$$\max_{j \in [M]}\left|\hat{\mathcal{T}}\hat{h}(\hat{\boldsymbol{z}}_j) - \mathcal{T}\hat{h}(\hat{\boldsymbol{z}}_j)\right| \lesssim \frac{\|\hat{h}\|_\infty L_\sigma r_b^{\bar{q}+1}}{\beta}\sqrt{\frac{\log(M/\delta)}{|S|}},$$

with probability at least $1 - \delta$. For any $\boldsymbol{z}$ with $\|\boldsymbol{z}\| \leq r_b$, let $\hat{\boldsymbol{z}}$ denote the projection of $\boldsymbol{z}$ onto the covering $(\hat{\boldsymbol{z}}_j)_{j=1}^M$. Then,

$$\sup_{\|\boldsymbol{z}\| \leq r_b/2}\left|\hat{\mathcal{T}}\hat{h}(\boldsymbol{z}) - \mathcal{T}\hat{h}(\boldsymbol{z})\right| \leq \max_{j \in [M]}\left|\hat{\mathcal{T}}\hat{h}(\hat{\boldsymbol{z}}_j) - \mathcal{T}\hat{h}(\hat{\boldsymbol{z}}_j)\right| + \left|\mathcal{T}\hat{h}(\hat{\boldsymbol{z}}) - \mathcal{T}\hat{h}(\boldsymbol{z})\right| + \left|\hat{\mathcal{T}}\hat{h}(\hat{\boldsymbol{z}}) - \mathcal{T}\hat{h}(\boldsymbol{z})\right|$$

$$\lesssim \frac{\|\hat{h}\|_\infty L_\sigma r_b^{\bar{q}+1}}{\beta}\sqrt{\frac{\log(M/\delta)}{|S|}} + \frac{\|\hat{h}\|_\infty L_\sigma r_b^{\bar{q}}\Delta}{\beta}.$$

$$\lesssim \frac{\|\hat{h}\|_\infty L_\sigma r_b^{\bar{q}+1}}{\beta}\sqrt{\frac{\log(r_b/(\Delta\delta))}{|S|}} + \frac{\|\hat{h}\|_\infty L_\sigma r_b^{\bar{q}}\Delta}{\beta}.$$

Choosing $\Delta = r_b/\sqrt{|S|}$ implies

$$\sup_{\|\boldsymbol{z}\| \leq r_b/2}\left|\hat{\mathcal{T}}\hat{h}(\boldsymbol{z}) - \mathcal{T}\hat{h}(\boldsymbol{z})\right| \lesssim \frac{\|\hat{h}\|_\infty L_\sigma r_b^{\bar{q}+1}}{\beta}\sqrt{\frac{\log(|S|/\delta)}{|S|}}$$

with probability at least $1 - \delta$ over the randomness of $(\boldsymbol{v}_i, b_i)_{i \in [N]}$.

The last step is to bound $\|\boldsymbol{a}^*\|^2$. Note that,

$$\|\boldsymbol{a}^*\|^2 \leq \frac{4r_b^2}{\beta^2|S|}\sum_{i \in S}\frac{\hat{h}(\boldsymbol{v}_i, b_i)^2}{|S|}.$$

Further, by the Hoeffding inequality,

$$\sum_{i \in S}\frac{\hat{h}(\boldsymbol{v}_i, b_i)^2}{|S|} - \mathbb{E}_{\boldsymbol{v}, b \sim \mu \otimes \text{Unif}(-r_b, r_b)}\left[\hat{h}(\boldsymbol{v}, b)^2\right] \lesssim \|\hat{h}\|_\infty^2\sqrt{\frac{\log(1/\delta)}{|S|}}$$

with probability at least $1 - \delta$. Moreover,

$$\mathbb{E}_{\boldsymbol{v}, b \sim \mu \otimes \text{Unif}(-r_b, r_b)}\left[\hat{h}(\boldsymbol{v}, b)^2\right] = \mathbb{E}_{\boldsymbol{v}, b \sim \tau_k \otimes \text{Unif}(-r_b, r_b)}\left[\hat{h}(\boldsymbol{v}, b)^2\frac{d\mu}{d\tau_k}(\boldsymbol{v})\right]$$

$$\leq \beta' M_2(\hat{h})^2.$$

Thus, when $|S| \geq \frac{\|\hat{h}\|_\infty^4 \log(1/\delta)}{\beta'^2 M_2(\hat{h})^4}$, we have $\|\boldsymbol{a}^*\|^2 \lesssim r_b^2 \beta' M_2(\hat{h})^2/(\beta^2|S|)$ with probability at least $1 - \delta$, which completes the proof. $\square$

## D.5    COMBINING ALL STEPS

We can finally bound our original objective of this section, i.e. $\text{AR}(\boldsymbol{a}^*, \boldsymbol{W}, \boldsymbol{b}) - \text{AR}^*$. Let us begin with the case where $\mathcal{F}$ is the class of polynomials of degree $p$.

**Proposition 35.** *Suppose $\mathcal{F}$ and $\sigma$ satisfy Assumption 4 and $(b_i)_{i \in [N]} \overset{\text{i.i.d.}}{\sim} \text{Unif}(-r_b, r_b)$. Recall that $\varepsilon_1 := 1 \vee \varepsilon$, and $\tilde{\epsilon} := \epsilon \wedge \frac{\epsilon^2}{\text{AR}^*}$ for any $\epsilon \in (0, 1)$. Using the simplification $k, q, p, L_\sigma \lesssim 1$ and recalling $\varepsilon_1 := 1 \vee \varepsilon$, there exists a choice of $r_b = \tilde{\Theta}(\varepsilon_1)$ such that:*

- *If $\boldsymbol{W} = (\boldsymbol{w}_1, \ldots, \boldsymbol{w}_N)^\top$ are given by the $\alpha$-DFL oracle, there exists $\boldsymbol{a}^*$ such that $|a_i^*| \leq \tilde{\mathcal{O}}(\varepsilon_1/(\alpha N))$ for all $i \in [N]$, and $\mathrm{AR}(\boldsymbol{a}^*, \boldsymbol{W}, \boldsymbol{b}) - \mathrm{AR}^* \leq \epsilon$ as soon as*

$$\zeta \leq \tilde{\mathcal{O}}\Big(\frac{\tilde{\epsilon}}{\varepsilon_1^{2(q+1)}}\Big) \qquad and \qquad N \geq \tilde{\Omega}\Big(\frac{\varepsilon_1^{q+1}}{\alpha\zeta^{(k-1)/2}\sqrt{\tilde{\epsilon}}}\Big).$$

- *If $\boldsymbol{W} = (\boldsymbol{w}_1, \ldots, \boldsymbol{w}_N)^\top$ are given by the $(\alpha,\beta)$-SFL oracle, there exists $\boldsymbol{a}^*$ such that $|a_i^*| \leq \tilde{\mathcal{O}}(\varepsilon_1/(\beta\alpha N))$ for all $i \in [N]$, and $\mathrm{AR}(\boldsymbol{a}^*, \boldsymbol{W}, \boldsymbol{b}) - \mathrm{AR}^* \leq \epsilon$ as soon as*

$$\zeta \leq \tilde{\mathcal{O}}\Big(\frac{\beta^2\tilde{\epsilon}}{\varepsilon_1^{2(q+1)}}\Big) \qquad and \qquad N \geq \tilde{\Omega}\Big(\frac{\varepsilon_1^{2(q+1)}}{\alpha\beta^2\tilde{\epsilon}}\Big).$$

*Both cases above hold with probability at least $1 - n^{-c}$ for some absolute constant $c > 0$ over the choice of random biases $(b_i)_{i\in[N]}$ (and random weights $(\boldsymbol{w}_i)$ in the case of SFL).*

**Proof.** Recall from (D.1) that

$$\mathrm{AR}(\boldsymbol{a}^*, \boldsymbol{W}, \boldsymbol{b}) - \mathrm{AR}^* \lesssim \mathcal{E}_1 + \mathcal{E}_2 + \sqrt{\mathcal{E}_3(\mathcal{E}_1 + \mathcal{E}_2)}.$$

By definition, $\mathcal{E}_3 \lesssim \mathrm{AR}^*$. By Lemma 24, we have

$$\mathcal{E}_2 \lesssim L_\sigma^2 \tilde{r}_a^2 (1 + r_b^{2(\bar{q}-1)} + \varepsilon^{2(\bar{q}-1)})(1 + \varepsilon^2)\zeta.$$

Further, thanks to Lemma 31 we have $|h(\boldsymbol{z})| \lesssim 1 + \|\boldsymbol{z}\|^p$. Therefore, by Lemma 25 with $r_z = r_b$, we have

$$\mathcal{E}_1 \lesssim \epsilon_{\mathrm{approx}}^2 + \big(L_\sigma^2 \tilde{r}_a^2(1 + \varepsilon^{2\bar{q}} + r_b^{2\bar{q}}) + 1 + \varepsilon^{2p}\big)e^{-\Omega(r_b^2)}.$$

Let us now consider the case of $\alpha$-DFL. Define $\bar{p} = (p-2) \vee 0$ if the ReLU activation is used and $\bar{p} = 0$ if the polynomial activation is used. Notice that by the definition in Assumption 4, we have $\bar{q} + \bar{p} = q$. By Proposition 33, we know there exists $\boldsymbol{a}^*$ with $|a_i^*| \leq \tilde{\mathcal{O}}(r_b^{1+\bar{p}}/(\alpha N))$ (we used the fact that $\max_{s\leq p}\|\boldsymbol{T}^{(s)}\|_\mathrm{F} \lesssim 1$ from Lemma 31) such that

$$\epsilon_{\mathrm{approx}} \leq \tilde{\mathcal{O}}\Big(r_b^{q+1}\big(\sqrt{\zeta} + \frac{1}{\zeta^{(k-1)/2}\alpha N}\big)\Big),$$

provided that $r_b \gtrsim \varepsilon_1$ where we recall $\varepsilon_1 = 1 \vee \varepsilon$, and the above statement holds with probability at least $1 - \delta$ for any polynomially decaying $\delta$, e.g. $\delta = n^{-c}$ for some absolute constant $c > 0$. Therefore, we have $\tilde{r}_a \leq \tilde{\mathcal{O}}(r_b^{1+\bar{p}})$. Further, it suffices to choose $r_b$ large enough such that $r_b \gtrsim \varepsilon_1 \vee \sqrt{\log(NL_\sigma^2\tilde{r}_a^2 r_b^{2\bar{q}} + \varepsilon_1^{2p})} = \tilde{\Theta}(\varepsilon_1)$ to have

$$\mathcal{E}_1 \leq \tilde{\mathcal{O}}\Big(r_b^{2(q+1)}\big(\zeta + \frac{1}{\zeta^{k-1}\alpha^2 N^2}\big)\Big).$$

Plugging in the values of $\tilde{r}_a$ and $r_b$, we obtain,

$$\mathcal{E}_2 \leq \tilde{\mathcal{O}}(\varepsilon_1^{2(q+1)}\zeta), \qquad and \qquad \mathcal{E}_1 \leq \tilde{\mathcal{O}}\Big(\varepsilon_1^{2(q+1)}\zeta + \frac{\varepsilon_1^{2(q+1)}}{\zeta^{k-1}\alpha^2 N^2}\Big).$$

Hence, choosing

$$\zeta \leq \tilde{\mathcal{O}}\Big(\frac{\tilde{\epsilon}}{\varepsilon_1^{2(q+1)}}\Big), \qquad and \qquad N \geq \tilde{\Omega}\Big(\frac{\varepsilon_1^{q+1}}{\alpha\zeta^{(k-1)/2}\sqrt{\tilde{\epsilon}}}\Big)$$

which concludes the proof of the $\alpha$-DFL case.

In the case of $(\alpha,\beta)$-SFL, we instead invoke Proposition 34, thus obtain $|a_i^*| \lesssim r_b^{1+\bar{p}}/(\beta\alpha N)$, and

$$\epsilon_{\mathrm{approx}} \leq \tilde{\mathcal{O}}\Big(\frac{L_\sigma r_b^{q+1}}{\beta\sqrt{\alpha N}}\Big),$$

which holds with probability at least $1 - \delta$ for any polynomially decaying $\delta$ such as $\delta = n^{-c}$ for some absolute constant $c > 0$. Consequently, with the same choice of $r_b = \tilde{\Theta}(\varepsilon_1)$ as before, we have

$$\mathcal{E}_2 \leq \tilde{\mathcal{O}}\Big(\frac{\varepsilon_1^{2(q+1)}\zeta}{\beta^2}\Big), \qquad \text{and} \qquad \mathcal{E}_1 \leq \tilde{\mathcal{O}}\Big(\frac{\varepsilon_1^{2(q+1)}}{\beta^2 \alpha N}\Big),$$

which completes the proof. $\qquad\square$

We can also combine approximation bounds for the more general class of pseudo-Lipschitz $\mathcal{F}$.

**Proposition 36.** *Suppose $\mathcal{F}$ and $\sigma$ satisfy Assumption 3 and $(b_i)_{i \in [N]} \overset{i.i.d.}{\sim} \mathrm{Unif}(-r_b, r_b)$. Recall that $\varepsilon_1 := 1 \vee \varepsilon$, and $\tilde{\epsilon} := \epsilon \wedge \frac{\epsilon^2}{\mathrm{AR}^*}$ for any $\epsilon \in (0, 1)$. Using the simplification $k, p, L \lesssim 1$, there exists a choice of $r_b = \tilde{\Theta}\big(\varepsilon_1(\varepsilon_1/\sqrt{\tilde{\epsilon}})^{1+1/k}\big)$ such that:*

- *If $\mathbf{W} = (\mathbf{w}_1, \ldots, \mathbf{w}_N)^\top$ is given by the $\alpha$-DFL oracle, there exists $\mathbf{a}^*$ such that $|a_i^*| \leq \tilde{\mathcal{O}}((\varepsilon_1/\sqrt{\tilde{\epsilon}})^{k+1+1/k}/(\alpha N))$ for all $i \in [N]$, and $\mathrm{AR}(\mathbf{a}^*, \mathbf{W}, \mathbf{b}) - \mathrm{AR}^* \leq \epsilon$ as soon as*

$$\zeta \leq \tilde{\mathcal{O}}\Big(\big(\frac{\tilde{\epsilon}}{\varepsilon_1^2}\big)^{k+2+1/k}\Big), \qquad \text{and} \qquad N \geq \tilde{\Omega}\Big(\frac{1}{\zeta^{(k-1)/2}\alpha}\big(\frac{\varepsilon_1}{\sqrt{\tilde{\epsilon}}}\big)^{k+3+2/k}\Big).$$

- *If $\mathbf{W} = (\mathbf{w}_1, \ldots, \mathbf{w}_N)^\top$ is given by the $(\alpha, \beta)$-SFL oracle, there exists $\mathbf{a}^*$ such that $|a_i^*| \leq \tilde{\mathcal{O}}\big((\varepsilon_1/\sqrt{\tilde{\epsilon}})^{k+1+1/k}/(\alpha\beta N)\big)$, and $\mathrm{AR}(\mathbf{a}^*, \mathbf{W}, \mathbf{b}) - \mathrm{AR}^* \leq \epsilon$ as soon as*

$$\zeta \leq \tilde{\mathcal{O}}\Big(\beta^2\big(\frac{\tilde{\epsilon}}{\varepsilon_1^2}\big)^{k+2+1/k}\Big), \qquad \text{and} \qquad N \geq \tilde{\Omega}\Big(\frac{1}{\alpha\beta^2}\big(\frac{\varepsilon_1^2}{\tilde{\epsilon}}\big)^{k+3+2/k}\Big).$$

*Both cases above hold with probability at least $1 - n^{-c}$ for some absolute constant $c > 0$ over the choice of random biases $(b_i)_{i \in [N]}$ (and random weights $(\mathbf{w}_i)$ in the case of SFL).*

**Proof.** Our starting point is once again the decomposition

$$\mathrm{AR}(\mathbf{a}^*, \mathbf{W}, \mathbf{b}) - \mathrm{AR}^* \leq \mathcal{E}_1 + \mathcal{E}_2 + \sqrt{\mathcal{E}_3(\mathcal{E}_1 + \mathcal{E}_2)}.$$

Given Assumption 3, it is straightforward to verify that $|h(\mathbf{z}_1) - h(\mathbf{z}_2)| \lesssim (\varepsilon_1^{1-p}\|\mathbf{z}_1\|^{p-1} + \varepsilon_1^{1-p}\|\mathbf{z}_2\|^{p-1} + 1)\|\mathbf{z}_1 - \mathbf{z}_2\|$ for $\mathbf{z}_1, \mathbf{z}_2 \in \mathbb{R}^k$. As a consequence, we have $|h(\mathbf{z})| \lesssim 1 + \|\mathbf{z}\|^p$ for all $\mathbf{z} \in \mathbb{R}^k$. Therefore, by Lemma 25 with a choice of $r_z = \tilde{\Theta}(\varepsilon_1)$, we have $\mathcal{E}_1 \lesssim \epsilon_{\mathrm{approx}}^2$. In the rest of the proof we will fix $r_z = \tilde{\Theta}(\varepsilon_1)$.

We begin by considering the case of $\alpha$-DFL. Unlike the proof of Proposition 35 where $\mathcal{T}\hat{h} = h$, in this case we have an additional error due to $\mathcal{T}\hat{h}$ only approximating $h$. From Lemma 32, we have

$$\|\hat{h}\|_\infty \leq \tilde{\mathcal{O}}\Big(\frac{1}{\varepsilon_1}\big(\frac{\Delta}{\varepsilon_1}\big)^{2k/(k+1)}\Big).$$

Thus,

$$
\begin{aligned}
\epsilon_{\mathrm{approx}} &\leq \sup_{\|\mathbf{z}\| \leq r_z} \Big|\sum_{j \in S} a_j^* \sigma(\langle \mathbf{v}_j, \mathbf{z}\rangle + b_j) - \mathcal{T}\hat{h}(\mathbf{z})\Big| + \Big|\mathcal{T}\hat{h}(\mathbf{z}) - h(\mathbf{z})\Big| \\
&\leq \tilde{\mathcal{O}}\Big(\frac{r_b}{\varepsilon_1}\big(\frac{\Delta}{\varepsilon_1}\big)^{\frac{2k}{k+1}}\big(\varepsilon_1\sqrt{\zeta} + \frac{r_b}{\zeta^{(k-1)/2}\alpha N}\big)\Big) + \tilde{\mathcal{O}}\Big(\varepsilon_1\big(\frac{\Delta}{\varepsilon_1}\big)^{-\frac{2}{k+1}} + \varepsilon_1\big(\frac{\Delta}{\varepsilon_1}\big)^{\frac{2k}{k+1}}\big(\frac{\varepsilon_1}{r_b}\big)^k\Big),
\end{aligned}
$$

where we bounded the first term via Proposition 33 with $\bar{q} = 1$, and the second term via Lemma 32. Additionally, we have

$$|a_j^*| \leq \frac{r_b}{\varepsilon_1 \alpha N}\big(\frac{\Delta}{\varepsilon_1}\big)^{\frac{2k}{k+1}},$$

for all $j \in [N]$. To obtain $\mathrm{AR}(\mathbf{a}^*, \mathbf{W}, \mathbf{b}) \leq \mathrm{AR}^* + \epsilon$, we must choose $\Delta = \tilde{\Theta}(\varepsilon_1(\varepsilon_1/\sqrt{\tilde{\epsilon}})^{(k+1)/2})$. Next, we choose $r_b = \tilde{\Theta}(\varepsilon_1(\varepsilon_1/\sqrt{\tilde{\epsilon}})^{(k+1)/k})$. This combination ensures $\Big|\mathcal{T}\hat{h}(\mathbf{z}) - h(\mathbf{z})\Big| \lesssim \sqrt{\tilde{\epsilon}}$. To make sure $\Big|\hat{\mathcal{T}}\hat{h}(\mathbf{z}) - \mathcal{T}\hat{h}(\mathbf{z})\Big| \lesssim \sqrt{\tilde{\epsilon}}$, we should let

$$\zeta \leq \tilde{\mathcal{O}}\Big(\big(\frac{\tilde{\epsilon}}{\varepsilon_1^2}\big)^{k+2+1/k}\Big), \qquad \text{and} \qquad N = \tilde{\Theta}\Big(\frac{1}{\zeta^{(k-1)/2}\alpha}\big(\frac{\varepsilon_1}{\sqrt{\tilde{\epsilon}}}\big)^{k+3+2/k}\Big).$$

The above guarantees that $\epsilon_{\mathrm{approx}} \lesssim \sqrt{\tilde{\epsilon}}$ and consequently $\mathcal{E}_1 + \sqrt{\mathcal{E}_3 \mathcal{E}_1} \lesssim \epsilon$. Note that the above choices imply $\left|a_j^*\right| \leq \tilde{r}_a/|S|$ for all $i \in S$ with $\tilde{r}_a = \tilde{\mathcal{O}}((\varepsilon_1/\sqrt{\tilde{\epsilon}})^{k+1+1/k})$. From Lemma 24 with $\bar{q} = 1$, we have $\mathcal{E}_2 \lesssim \tilde{r}_a^2 \varepsilon_1^2 \zeta$. Therefore, if we let

$$\zeta = \tilde{\Theta}\Big(\big(\frac{\tilde{\epsilon}}{\varepsilon_1^2}\big)^{k+2+1/k}\Big),$$

we have $\mathcal{E}_2 \lesssim \tilde{\epsilon}$ and consequently $\mathcal{E}_2 + \sqrt{\mathcal{E}_3 \mathcal{E}_1} \lesssim \epsilon$. This concludes the proof of the $\alpha$-DFL case.

Next, we consider the case of $(\alpha,\beta)$-SFL. Note that the error $\left|\mathcal{T}\hat{h}(z) - h(z)\right|$ remains unchanged. However, this time we invoke Proposition 34 for controlling $\left|\hat{\mathcal{T}}\hat{h}(z) - \mathcal{T}\hat{h}(z)\right|$. Therefore,

$$\epsilon_{\mathrm{approx}} \leq \sup_{\|z\| \leq r_z} \left|\sum_{j \in S} a_j^* \sigma(\langle v_j, z\rangle + b_j) - \mathcal{T}\hat{h}(z)\right| + \left|\mathcal{T}\hat{h}(z) - h(z)\right|$$

$$\leq \tilde{\mathcal{O}}\Big(\frac{r_b^2}{\beta\varepsilon_1}\big(\frac{\Delta}{\varepsilon_1}\big)^{\frac{2k}{k+1}}\sqrt{\frac{1}{\alpha N}}\Big) + \tilde{\mathcal{O}}\Big(\varepsilon_1\big(\frac{\Delta}{\varepsilon_1}\big)^{-\frac{2}{k+1}} + \varepsilon_1\big(\frac{\Delta}{\varepsilon_1}\big)^{\frac{2k}{k+1}}\big(\frac{\varepsilon_1}{r_b}\big)^k\Big).$$

Since the second term is unchanged, we have the same choices of $\Delta = \tilde{\Theta}\big(\varepsilon_1(\varepsilon_1/\sqrt{\tilde{\epsilon}})^{(k+1)/2}\big)$ and $r_b = \tilde{\Theta}\big(\varepsilon_1(\varepsilon_1/\sqrt{\tilde{\epsilon}})^{1+1/k}\big)$ as in the $\alpha$-DFL case. However, for the finite-width discretization, we should choose

$$N = \tilde{\Theta}\Big(\frac{1}{\alpha\beta^2}\big(\frac{\varepsilon_1^2}{\tilde{\epsilon}}\big)^{k+3+2/k}\Big). \tag{D.7}$$

Moreover, Proposition 34 implies $\left|a_j^*\right| \leq \tilde{r}_a/|S|$ with $\tilde{r}_a = \tilde{\mathcal{O}}\big((\varepsilon_1/\sqrt{\tilde{\epsilon}})^{k+1+1/k}/\beta\big)$. As a result, to get $\mathcal{E}_2 \lesssim \tilde{r}_a^2 \varepsilon_1^2 \zeta \leq \tilde{\epsilon}$ from Lemma 24 with $q = 1$, we let

$$\zeta = \tilde{\Theta}\Big(\beta^2\big(\frac{\tilde{\epsilon}}{\varepsilon_1^2}\big)^{k+2+1/k}\Big),$$

completing the proof.

$\square$

# E   ADDITIONAL EXPERIMENTS

In this section, we perform a simple experiment on the MNIST dataset (LeCun et al., 1998) to demonstrate that the intuitions from our theoretical results go beyond the setting of multi-index models, squared loss, and $\ell_2$ norm attacks. We choose a convolutational neural network as our predictor, given by two convolution layers with 32 and 64 channels respectively, a max pooling layer, and two fully connected layers. For each choice of $\epsilon$, we do two experiments:

1. *ADV training*: We train the network adversarially for 11 epochs, where we initialize the model using the default PyTorch initialization, then use the adversarial training algorithm of Madry et al. (2018).

2. *STD + ADV training*: We first train the model with standard SGD for 10 epochs from PyTorch initialization, then perform 10 epochs of adversarial training on top of this standard presentation.

For both approaches, we use the corss entropy loss, a batch size of 64, a learning rate of 0.01 for both PGD and SGD updates, and we use $\ell_\infty$ norm to constrain perturbations, where pixels are normalized between 0 and 1. Note that due to the additional cost of generating adversarial samples, taking one additional epoch on ADV ensures a fare comparison where ADV and STD + ADV will roughly have the same computational complexity.

As can be seen from Table 1, the STD + ADV training approach achieves a higher test accuracy compared to the ADV approach across all model architectures considered here. This is consistent with our intuition from the guarantees of Algorithm 1. The standard training phase can recover an

| Algorithm | $\varepsilon = 0.2$ | $\varepsilon = 0.3$ | $\varepsilon = 0.4$ |
|---|---|---|---|
| ADV | $91 \pm 1$ | $82 \pm 2$ | $69 \pm 2$ |
| STD + ADV | $\mathbf{93.80} \pm 0.09$ | $\mathbf{87.2} \pm 0.4$ | $\mathbf{75.3} \pm 0.6$ |

Table 1: Adversarial test accuracy comparison between ADV and STD + ADV across three different values for $\varepsilon$ on the MNIST dataset. The error shown for each accuracy is the standard deviation over three runs.

optimal low-dimensional representation on top of which adversarial training becomes easier. Note that since we work with multi-layer convolutional neural networks, there is no longer a single layer that captures the entirety of the low-dimensional projection, which is why we choose to retrain all parameters of the network adversarially. One interesting direction for future research is to understand in settings beyond two layers, which parameters need to be adversarially trained after the standard training phase. The code to reproduce the results of Figure 1 and Table 1 is provided at: https://github.com/mousavih/robust-feature-learning.

