# OpenReview forum: "Robust Feature Learning for Multi-Index Models in High Dimensions"
_ICLR.cc/2025/Conference — ICLR 2025 Poster_

### Official Review · Reviewer_6kxP · 2024-11-02

**Soundness:** 3
**Presentation:** 2
**Contribution:** 2
**Rating:** 8
**Confidence:** 3

**Summary:**

This paper studies the adversarial training of two-layer neural networks on learning multi-index target function (a function of a low-dimensional projection of the input) under squared loss. Under some assumptions, the authors first show that there exists an optimal low-dimensional projection (of inputs) for robustness against $l_2$ adversarial perturbations (Theorem 1). They then provide conditions for the first layer weights to satisfy such that the optimal low-dimensional projection matrix and the first layer matrix are partially aligned (Definition 2-3). Since there exist practical feature learning algorithms (examples given in Section 4.2) satisfying these conditions (definitions), they focus on the case where only the second layer is trained with adversarial loss while the first layer is pre-trained with a standard (not adversarial) feature learning algorithm. This simplification allows them to study the adversarial training of the second layer mainly. In this setting, the paper shows that adversarial training does not significantly affect the sample complexity of learning the target function. Specifically, the number of additional samples required for adversarial training (in comparison to standard training) is shown to be independent of the input dimension (Theorem 4-7). Thus, the authors conclude that adversarial learning in this setting is as easy as standard feature learning.

**Strengths:**

- **Interesting problem:** Understanding the effects of adversarial training of neural networks on the sample complexity may be significant for the deep learning literature. Furthermore, this paper combines two lines of work: 1) feature learning in two-layer neural networks and 2) adversarially robust learning in linear and random feature models.

- **Theoretical study under reasonable assumptions:** The authors study the problem under reasonable assumptions (in line with the prior work about such as Hassani & Javanmard, 2024).

- **Provides a valuable insight into the problem:** The paper shows that the adversarial training does not significantly (in the order of dimensions) affect the sample complexity in comparison to standard training.

- **Good enough presentation:** The paper answers most of the questions a reader may have, which is the main criterion when assessing the quality of the presentation. However, I think it is not an easy-to-follow paper. Instead, some parts of the paper require back-and-forth reading, which may be acceptable for such a theoretical paper.

**Weaknesses:**

- **No characterization of robust test risk:** Prior works such as (Hassani & Javanmard, 2024) provide an asymptotic characterization of robust test risk for random feature models, which allows them to show how factors such as dimensions or perturbation budget $\epsilon$ impact the robust test risk. However, this work just focuses on the sample complexities and ignores the characterization of robust test risk.

- **Significant gap to practice:** Although the assumptions are reasonable from a theoretical perspective, they differ significantly from practical adversarially robust learning.
   + **Data assumption:** Input is assumed to be subGaussian with constant subGaussian norm in Assumption 2. Furthermore, Assumption 1 imposes an implicit assumption on inputs and labels (target function), which needs further discussion (please see Question 1 about this below).
   + **The first layer is also trained with adversarial loss in practice:** While it seems like the adversarial training of the first layer hurts the robust test risk in Figure 1, this requires explanation (mentioned as future work in the conclusion). Thus, adversarial training of the first layer  is missing in this work.
   + **Limitation of $l_2$ perturbations:** The paper only studies adversarial training with $l_2$-bounded perturbations. Although $l_2$ perturbations for the adversarial training may be a good starting point considering the prior work on the theory of robust learning (Hassani & Javanmard, 2024), $l_{\infty}$ is primarily used in empirical studies as mentioned by the authors.
   + **Assumption on perturbation budget $\epsilon$:** The paper assumes constant adversary budget $\epsilon = \mathcal{O}(1)$ and justifies it by saying even this much perturbation can significantly perturb the output of the prediction function. However, as the authors noted, $\mathbb{E}[||\mathbf{x}||] = \mathcal{O}(\sqrt{d})$ and thus perturbation budget of $\epsilon = \mathcal{O}(\sqrt{d})$ is used in practice, which causes another discrepancy between the assumptions and practice.
   +  **Only polynomial and ReLU activation functions are considered.**

- **Limited numerical results and limited discussion of those results:**
   + Most of Section 5 (Numerical Experiments) describes the experimental setting, while only one or two sentences discuss the results. There should be a more insightful discussion of the numerical results and their relation to the theoretical results.
   + There are some missing details about Figure 1 (please see Question 2.a-b below).
   + Effects of some parameters of the setting should be illustrated numerically or at least discussed: 1) activation function (polynomial vs ReLU), 2) different values for adversary budget $\epsilon$, 3) dimensions and number of samples.

**Questions:**

1.  **The assumption on the data distribution:**
     + a) The authors mention that inputs can be Gaussian vectors with isotropic covariance, but what are other examples satisfying this assumption? For example, can we have anisotropic covariance (e.g., spiked covariance)? What if the covariance matrix and the projection matrix $U$ are aligned?
     + b) The authors mention (in Line 388) that "Our results readily extend to these settings as well," referring to non-isotropic Gaussian inputs and non-Gaussian symmetric input distributions. How so?

2. **Numerical results:**
    + a) What is the step size for the gradient descent step on the perturbed batch for Figure 1?
    + b) What do shaded error bands denote in Figure 1? If variance, how much variance is illustrated?
    + c) Why do we see oscillating robust test risk for the two cases other than the full adversarial (full AD) training, while the robust test risk for the full adversarial training seems to be smooth in Figure 1? Furhermore, the robust test risk for full AD seems to converge fast while those for the other two cases do not seem converge even at the end of the iterations. Why so?
    + d) The authors provide theoretical results on polynomial predictors in Section 4.1. A natural question is, "How does polynomial activation compare to ReLU activation in numerical results?".
    + e) Prior work (Hassani & Javanmard, 2024) showed that adversary budget $\epsilon$ can significantly impact the robust test risk in their setting. How does adversary budget $\epsilon$ affect the robust test risk here? Similarly, what is the impact of dimensions and the number of samples?

3. The authors mention $\zeta^{(k-1)/2}$ as "the normalizing factor" in Line 239. But, $\zeta$ on its own deserves an explanation since it appears in most of the theorems. What does $\zeta$ intuitively correspond to?

4. **Minor issues:**
    + Some non-standard notations are used without definition. For example, soft O notation $\tilde{\mathcal{O}}$ and its counterparts for $\Omega$ and $\Theta$ should be mentioned in the "Notation" paragraph starting in Line 122.
    + There seems like a notational abuse when using the aforementioned asymptotic notations. There is asymptotic notation inside another asymptotic notation (e.g., in Theorem 4): $N \geq \tilde{\Omega}(\cdot^{\mathcal{O}(k)})$. Indeed, the authors assume $k = \mathcal{O}(1)$ in Line 154, which means $k < C$ for some $C > 0$. So, these equations can be simplified, and the notational abuse can be fixed at the expense of introducing one additional constant. Furthermore, note that the asymptotic notations are conventionally used with equality sign $(=)$ instead of $\leq$ or $\geq$.
    + In Line 135, the authors define the family of prediction functions to be "non-parametric". However, they then use two-layer neural networks as a prediction function in Line 140 or d-variate polynomials as the family of prediction functions in Section 4.1, which are "parametric" families of functions. There seems to be a contradiction here. Or am I missing something?
    + Ranges of $\alpha$ and $\beta$ are missing from Definitions 2-3.
    + Complexity results provided in Theorem 4-7 reduce the readability of main text since they include many parameters related to the setting. I think that the author can provide more insightful presentation of these results showing the essence.

---

> ### Author Response · Authors · 2024-11-23
>
> We thank the reviewer for their careful assessment and valuable feedback, and address their comments and questions below.
>
> ---
> **Tight characterization of robust test risk**:
>
> While we agree with the reviewer that a tight characterization of the robust test risk would be highly valuable, we would like to highlight that *even in the standard setting* there is no tight characterization of the risk under feature learning, unlike its random feature counterpart. This is due to the inherent complexity of analyzing feature learning, and it is unlikely that the current techniques will be able to provide a sharp characterization of test risks in feature learning, especially in the distribution-free setting of our Theorems 4-7.
>
> ---
> **Data assumption: having anisotropic covariance and non-Gaussian data**:
>
> As we show in the general response above, it is possible for anisotropic covariance matrices to satisfy Assumption 1. Further, Assumption 1 is *necessary* for Theorem 1, and without it the optimal representation may no longer be $U$. Based on our discussion in the general response, it is an interesting direction to consider other algorithms that achieve optimal performance beyond Assumption 1.
>
> Please note that Theorem 4-7 only require Assumption 1 and *do not depend on Gaussianity or isotropic covariance*. In particular, since gradient-based feature learning algorithms are known for anisotropic covariance (e.g. [1,2]), and for non-Gaussian distributions (e.g. [3]) one can combine these with our results to obtain end to end robust learnability guarantees.
>
> ---
> **Limitation of $\ell_2$ perturbation**:
>
> As we discuss in the general response above, the $\ell_2$ attack norm is also *necessary* for Theorem 1, as the optimal representation may change with the $\ell_\infty$ attack. This is another interesting open direction, and our discussion in the general response above can provide a starting point for developing algorithms in that setting.
>
> ---
> **Assumption on the perturbation budget**:
>
> We would like to highlight that scaling $\varepsilon$ with $\sqrt{d}$ in this setting will be undesirable, since it will encourage the optimal predictor given by $x \mapsto h(\langle u, x\rangle)$ to have a fixed prediction over an input range which grows with high dimensions. This is because $\langle u, x\rangle$ is $\mathcal{O}(1)$ with high probability, yet $\langle u, \delta\rangle$ can be $\mathcal{O}(\sqrt{d})$ where $\delta$ is the perturbation. Therefore, the prediction of $h$ will collapse to a constant regardless of input in high dimensions, which is undesirable. We believe the natural scaling of $\varepsilon$ in our setting is constant with respect to dimension.
>
> ---
> **Only polynomial and ReLU activations are considered**:
>
> We remark that in principle our results should be applicable to other activation functions. However, please see the general response above for a discussion on the technical difficulties of considering other activations.

---

> > ### Author Response · Authors · 2024-11-23
> >
> > ---
> > **Limitations of the Numerical Experiments Section**:
> > We would like to point out that we have added a simple experiment on MNIST in Appendix E to demonstrate that the intuition from our theoretical results carry over to settings with neural networks with multiple layers, cross entropy loss, and $\ell_\infty$ norm for attack. Please note however that our experiments are for illustrative purposes, and are not the main focus of the submission. Therefore, our results and discussions in the numerical experiments section are not as extensive as our theoretical results and discussions. Below, we address the concerns regarding the numerical experiment of Section 5:
> > * **Further discussions needed**: We thank the reviewer for this suggestion, and will provide additional discussions on interpreting the results of the experiment in the final version of the manuscript.
> > * **Missing details**: We will further clarify the experimental setup in the revised version. Further, we will provide *links to the implementation in the de-anonymized final version of the manuscript.*
> >     * We choose a step size of $0.001$ on the perturbed batch, and a step size of $0.01$ for (signed) gradient ascent to find the perturbation.
> >     * The shaded error bars represent one standard deviation estimated over three runs.
> >     * Many aspects of the dynamics shown in the figures are not theoretically well-understood, and remain open for further investigation. For example, there are no guarantees on the convergence or behavior of adversarial training, therefore it is hard to argue when it will converge fast and what the typical variance of its convergence will be. Further studies outside the scope of our paper are required to fully capture all phenomena occurring in these training dynamics, which we believe are interesting future directions.
> >
> >     * We expect polynomial activations to perform poorly here, since they are not able to approximate non-polynomial functions such as ReLU or Tanh. We have focused on the ReLU activation for the student network in our experiment to be closer to practical neural networks.
> >
> >     * As the budget increases, the optimal predictor is pushed towards being constant. In that case, we believe there will not be a significant difference between the different approaches considered in Figure 1. Similarly, when $\varepsilon \to 0$, the performance of different approaches are close as we fall back to the standard regime. We expect that in high dimensions the sample complexity to recover the optimal direction and learn the optimal predictor increases, although the optimization landscape of low dimensions can be significantly different, and in certain cases high dimensions can help with convergence. We will further discuss these in the final version of the manuscript.
> >
> > ---
> > **The meaning of $\zeta$**:
> >
> > $\zeta$ controls the error tolerance of the oracle, smaller $\zeta$ requires a more accurate oracle that can cover the target subspace with higher precision, and as a consequence will require more samples. We will further clarify this in the final version.
> >
> > ---
> > **The non-parametric family of functions $\mathcal{F}**:
> > We remark that $\mathcal{F}$ is not the class from which we choose our predictors, it is the class that we would like our predictor to compete with, i.e. we measure the optimality of the trained neural network with respect to the performance of the best predictor in $\mathcal{F}$. When choosing $\mathcal{F}$ to be the class of pseudo-Lipschitz functions, $\mathcal{F}$ is indeed non-parametric. However, we agree with the reviewer that when considering the class of degree $p$ polynomials, they can be seen as a parametric family. Therefore, the non-parametric remark mostly applies to the case of pseudo-Lipschitz $\mathcal{F}$.
> >
> > ---
> > **Notational suggestions**:
> > We thank the reviewer for additional suggestions for notations, and will modify our manuscript for better readability.
> >
> > ---
> > References:
> >
> > [1] J. Ba et al. “Learning in the presence of low-dimensional structure: a spiked random matrix perspective.” NeurIPS 2023.
> >
> > [2] A. Mousavi-Hosseini et al. “Gradient-based feature learning under structured data.” NeurIPS 2023.
> >
> > [3] A. Zweig et al. “On single-index models beyond gaussian data.” NeurIPS 2023.

---

> > > ### Comment · Reviewer_6kxP · 2024-11-26
> > >
> > > I thank the authors for their time and effort in providing such a detailed response and additional experimental results. They addressed some of my concerns and explained why the others (e.g. $\ell_{\infty}$ attack and general activation functions) were challenging. However, some clarifications (e.g. missing details) promised by the authors have not yet been added to the revised paper. Furthermore, the citation for the MNIST dataset is missing while the additional experiments are on this dataset. Supposing that these will be added to the paper before the camera-ready version, I am raising my score to 8 (accept).

---

> > > > ### Author Response · Authors · 2024-11-27
> > > >
> > > > We thank the reviewer for providing additional feedback and raising their score. We have added some of the missing details to the newly revised version, and will take another pass on the manuscript before submitting the final version to ensure clarity and readability.

---

### Official Review · Reviewer_1AAV · 2024-11-03

**Soundness:** 4
**Presentation:** 4
**Contribution:** 3
**Rating:** 8
**Confidence:** 4

**Summary:**

This paper explores the robust feature learning properties of two-layer neural networks for learning multi-index models in high-dimensional settings, under $\ell_2$-bounded perturbations. In particular, the authors show that Bayes-optimal adversarial risk can be achieved by neural networks once the input data are projected onto the low-dimensional manifold defined by the target multi-index model (first-layer feature learning) and the second layer is trained by minimizing the empirical adversarial risk. They demonstrate that the resulting sample complexity is independent of the ambient dimension. Leveraging prior results, the paper provides guarantees for robustly learning multi-index models with gradient-based algorithms and includes numerical simulations to support its findings.

**Strengths:**

This paper
* presents results with clarity and precision, offering complete proofs, insightful comments, and remarks that make the content accessible and well-structured
* leverages previous results and introduces interesting, non-trivial findings in a rigorous framework (e.g. the possibility of performing standard feature learning to obtain optimal adversarial risk)
* validates its theoretical results with numerical simulations
* acknowledges its limitations and open problems.

**Weaknesses:**

I did not find any major weaknesses in this work. A minor issue would be the reliance of the results on the assumption of independence between the coordinates $x_\parallel$ and $x_\perp$ (Assumption 1), see also my question 1.

**Questions:**

1. Can you provide insights or intuitions about the validity of your results beyond Assumption 1? Have you numerically tested your algorithmic framework in settings that do not satisfy this assumption?

2. In lines 384-388, you claim that your results in this section can be readily extended to the setting of anisotropic Gaussian $x$, which seems to contrast with Assumption 1. Is it still possible to leverage Theorems 4-7 in this case?


3. I believe there is a typo in equation (4.1), where $y$ should instead be $y^{(i)}$.

---

> ### Author Response · Authors · 2024-11-23
>
> We thank the reviewer for their careful assessment and valuable feedback, and address their comments and questions below.
>
> ---
> **The validity of Theorem 1 beyond Assumption 1**:
>
> As we show in the general response above, Assumption 1 is *necessary* for Theorem 1, and without it the statement of the theorem may no longer hold. However, as also shown in the general response, in certain settings one can still identify closed form transformations on $U$ that lead to optimal presentations provided that they are able to estimate the covariance matrix. Developing algorithms that are optimal beyond Assumption 1 based on this observation is an interesting direction for future work.
>
> ---
> **Using Theorems 4-7 with Anisotropic Gaussians**:
>
> We demonstrate in the general response above that Gaussians with certain anisotropic covariance matrices can satisfy Assumption 1, in which case we can use Theorems 4-7.
>
> Further, we thank the reviewer for noticing the typo that we fixed in the revised version.
>
> ---
> We would be happy to clarify any concerns or answer any questions that may come up during the discussion period.

---

> > ### Comment · Reviewer_1AAV · 2024-11-26
> >
> > Thank you for your response

---

### Official Review · Reviewer_E3tK · 2024-11-03

**Soundness:** 3
**Presentation:** 2
**Contribution:** 3
**Rating:** 6
**Confidence:** 3

**Summary:**

The authors study the problem of $\ell_2$-constrained adversarial regression under a multi-index model setting. Specifically, in this setting, the conditional form of $y\vert x = g(\{\langle u_i, x\rangle\}_{i \in [k]})$ holds, where $u_i$ are the weights (or features) to be learned, $g:\mathbb R^k \to \mathbb R$ is the link function and $x,y$ are the covariates and target, respectively.

Their main result shows that under the assumption that the model relevant co ordinates $Ux$ (for $U \in \mathbb R^{k \times d}$) and model irrelevant co ordinates $\tilde Ux$ ( $\tilde U \in \mathbb R^{(k-d)\times d}$) are statistically independent, there is a learning method that attains a Bayes optimal adversarial risk.

This is achieved through access of oracles which can be realized through gradient based methods, with additional assumptions on the data distribution and for a certain class of prediction functions, such as variants of Lipschitz and polynomials.

Under the assumption that the predictor is a "pseudo"-Lipschitz, the sample complexity is not dependent on the data dimension, though it has an exponential dependence on $k$; for polynomial predictors, their sample complexity results do not depend on k.


They do not analyze the computational complexity of this problem, and provide empirical evidence by comparing training approaches including initializations based on a known $u$ and fully adversarial training.

**Strengths:**

The contribution is good because it is a new theoretical study for adversarial feature learning for learning multi index models, which has been gaining traction recently. Their main results highlight that robust learning is achievable without a significant increase (at least for some functions) in sample complexity. Moreover, the oracles assumed for learning the target subspace can be realized by training the first layer in a 2 layered neural network. Overall, this is a well structured, theoretically sound work with a good upshot and I recommend an accept.

**Weaknesses:**

I would like to reiterate a few points they already raised. Firstly, for the case of pseudo-Lipschitz functions, while the sample complexity is independent of $d$, there is an exponential dependence on $k$, which seems quite high.

I also think that many mathematical terms could benefit with more intuitions for the first time reader. For example, a minor point of clarification would be in Theorem 1, where it would be helpful to specify the random variables with respect to which the expectation is taken in the towering property, $\mathbb E_X[\cdot] = \mathbb E_Y[\mathbb E_X[\cdot \vert x,y]$.

It is also somewhat unclear to me what $\beta$ represents in the SFL oracle and also if a lower bound on $d\mu/d \tau_k$ connects to the Assumption 1. Moreover, I also think that Assumption 1 does not hold for natural scenarios where $x$ has a low rank structure where $\tilde Ux$ may not vary independently from $Ux$.

**Questions:**

Asked above.

---

> ### Author Response · Authors · 2024-11-23
>
> We thank the reviewer for their careful assessment and valuable feedback, and address their comments and questions below.
>
> ---
> **Exponential dependency on $k$**:
>
> We would like to highlight that without additional assumptions, the exponential dependency on $k$ is predicted by minimax lower bounds [1, Section 2.5], therefore unfortunately not improvable in general.
>
> ---
> **Improving Notation**:
>
> We thank the reviewer for their suggestion, and will improve the notation for an easier understanding in the final version of the manuscript.
>
> ---
> **The meaning of $\beta$ in the SFL definition**:
>
> A lower bound on $d\mu/d\tau_k$ is asking for the stochastic weights sampled from $\mu$ to be sufficiently present across all directions on the low-dimensional sphere $\mathbb{S}^{k-1}$. Note that when $d\mu/d\tau_k \geq \beta$, then if the uniform measure assigns some probability $\tau_k(A)$ to some $A \subseteq \mathbb{S}^{k-1}$, then $\mu(A) \geq \beta \tau_k(A)$, thus we are sufficiently covering $A$ when $\beta > 0$.
>
> ---
> **Validity of Assumption 1**:
>
> We refer to the general response for examples of anisotropic/low-rank covariance structure where Assumption 1 is verified. Please note that in the example provided in the general response, both $\Sigma_1$ and $\Sigma_2$ can be low-rank, leading to a low-rank overall covariance matrix $\Sigma$. However, we agree with the reviewer in that the independence between $Ux$ and $\tilde{U}x$ is still *necessary* for Theorem 1, as we show in the general response that without this assumption, the theorem may no longer hold.
>
> ---
> We would be happy to clarify any concerns or answer any questions that may come up during the discussion period.
>
> References:
>
> [1] A. Tsybakov. “Introduction to Nonparametric Estimation.” 2009.

---

> > ### Comment · Reviewer_E3tK · 2024-11-28
> >
> > Thanks for your comments. I retain my score.

---

### Official Review · Reviewer_UkaQ · 2024-11-04

**Soundness:** 3
**Presentation:** 2
**Contribution:** 3
**Rating:** 6
**Confidence:** 3

**Summary:**

This work considers robust learning in the multi-index model for two-layer ReLU networks. In particular, authors study the setting of worst-case $\ell_2$-perturbation of the input and the problem of regression with squared loss.

They propose the following two-stage method to find a set of parameters (weight of the first layer and readout vector):
First, algorithm finds a low-dimensional subspace, on which the target function depends. This is done using non-adversarial training, using prior feature learner methods.
Using this subspace, adversarial training is done on the readout vector.

**Strengths:**

The question of robust learning in the multi-index model is well-motivated by the authors. Authors clearly state the assumptions and limitations of their results.

As the main result, authors prove that in the multi-index model, sample complexity of adversarial training is the same as in standard training. They combine prior work on feature learning with a second stage adversarial training which requires careful technical analysis. Authors also provide experiments to support their claim.

**Weaknesses:**

Authors highlight themselves that the result is limited to $\ell_2$ norm of perturbations. Some discussion on the extension to $\ell_\infty$ norm is lacking, i.e., why current proof techniques cannot tackle this setting.

Furthermore Theorem 1, which is central for the results of the paper, relies on the squared loss assumption. For classification tasks, which is a standard setting for adversarial robustness, logistic loss would be a more natural choice.

Organization of the main text is hard to follow, theorems for SFL and DFL are more or less copy each other, but stated in different places. Since only one proof (out of four Theorems 4-7) is given in the Appendix, possibly a cleaner way to present the results is to combine Theorems 4&5 together, and show Theorems 6&7 as a corollary under additional assumption (with a separate proof).

Proof of Lemma 16 in the Appendix is hard to follow because of typos (line 894, should be $A_i (a_i - a'_i)$), and also $\tilde k_i$ is defined with $k_i$ and $k'_i$ in line 899, but both $k_i, k'_i$ are not defined at this point. I think what authors wanted to say is that $k_i, k'_i$ is defined with $\tilde k_i$, which is obtained through Mauery's lemma. It is confusing as it is written now.

Numerical experiments can also be improved, see Questions.

**Questions:**

Second row of Figure 1: why having $W$ fixed at SD training plateaus, while continuing to train $W$ improves the performance? Doesn't it contradict the theoretical results, since according to Algorithm 1, one should not update $W$ after the first stage?

Why for the numerical experiments Algorithms 2,3 are not used?

---

> ### Author Response · Authors · 2024-11-23
>
> We thank the reviewer for their careful assessment and valuable feedback, and address their comments and questions below.
>
> ---
> **Discussion on extension to $\ell_\infty$ norm**:
>
> We refer to the general response above about a discussion on extension to $\ell_\infty$ attack norm, and the necessity of the $\ell_2$ assumption for our theoretical result.
>
> ---
> **Going from squared to logistic loss**:
>
> We agree with the reviewer that our proof technique relies crucially on a decomposition of the squared loss, and considering other loss functions is an important open direction. We have added a simple experiment on the MNIST dataset in Appendix D which demonstrates that the intuitions from our theory carry over to the case of cross entropy loss.
>
> ---
> **Improving the structure of the paper**:
>
> We thank the reviewer for raising their concern about the structure of the paper, and we will improve the presentation in the final version of the manuscript.
>
> ---
> **Typos in the proof of Lemma 16**:
>
> We thank the reviewer for noticing the typos. We have fixed them in the revised version, and made the proof more easily readable.
>
> ---
> **Having W fixed at SD training vs. continuing with AD training**:
>
> The additional improvement from continuing with adversarial training in the first layer weights comes from the fact that given the finite number of samples in the standard training phase, it is only able to approximately recover the low-dimensional information in the target function. Further training allows exploiting further samples and more information about the low-dimensional projection.
>
> ---
> **Why Algorithms 2 and 3 are not used in the numerical experiments**:
>
> Our goal in the numerical experiments section was to show that one does not need to consider the contrived feature learning algorithms of prior works to take advantage of the observation in Theorem 1, and even simpler algorithms such as vanilla SGD that are used in practice can be sufficient. We will further clarify this in the final version.
>
> ---
> We would be happy to clarify any concerns or answer any questions that may come up during the discussion period.

---

> > ### Comment · Reviewer_UkaQ · 2024-11-25
> >
> > I thank the authors for addressing my comments. I will maintain my positive score. I hope that the authors will incorporate clearer theorem statements (combining pairs of similar statements into one) and extend experiment discussion (considering Algorithm 2 and 3) in the final version.

---

### Official Review · Reviewer_vLR3 · 2024-11-04

**Soundness:** 3
**Presentation:** 2
**Contribution:** 3
**Rating:** 8
**Confidence:** 3

**Summary:**

This paper demonstrates that it is possible to robustly train a multi-index model without impacting the statistical performance of the algorithm compared to standard learning methods. Assuming a low-dimensional structure in the probability distribution of the observations, the authors propose a procedure for robust learning in two-layer neural networks. This approach involves a standard learning step for the input layer aiming at recovering the relevant low-dimensional subspace, followed by a robust learning phase for the output layer in lower dimension.

In Theorems 4, 5, 6, and 7, the authors provide rigorous asymptotic bounds on the sample complexity (in the second step), number of neurons, and achievable error for the feature learning algorithm, ensuring the algorithm attains a small adversarial generalization error. These bounds are shown to be independent of the ambient dimension. Finally, using recent results on the number of observations necessary to learn the relevant low-dimensional subspace, the authors establish a scaling for the sample complexity for both single and multi-index models in Corollaries 9 and 11.

**Strengths:**

-	Although I do not have any particular knowledge in robust training, I found the paper accessible and well explained. The stakes are clearly presented in the introduction. Additionally, the setup and the assumptions are generally well motivated.
-	Despite not having read the proofs in detail, I appreciated that the proof techniques originate from diverse ideas and prior works including statistical learning, with generalization bounds provided in Appendix C1, geometrical analysis from the low-dimensional subspace assumption, approximation of functions using infinite-width neural networks, etc.
-	From the references provided in the introduction, it is clear that this paper offers new results and insights, distinguishing this work from previous research in the field.
-	The numerical experiment displayed in the main text gives a nice support to the theoretical claims of the paper.

**Weaknesses:**

-	Theorems 4, 5, 6 and 7 rely on specific activations (ReLU / polynomial). What is the technical challenge arising when considering more general activations?
-	Although it is interesting to include some proofs in the main text, the proof of Theorem 1 could be easily replaced with an outline of the proof ideas.
-	On the other side, it would be helpful for the reader to have a paragraph or two summarizing the main steps or ideas of the proofs of the main theorems.
-	One of the main claim of this work is that the bounds derived do not depend on the dimension. These bounds include the optimal adversarial risk (defined in equation 2.2), therefore, this quantity should not depend on the dimension. Whether this is a direct consequence of the definition or of Theorem 1, I don’t see it mentioned in the main text.
-	Several concepts and notations are not clearly defined, and it sometimes affect the understanding of the results. My main concern is the lack of a definition for the Landau notations in the main results. As a consequence, it is not clear whether or not the obtained bounds are non-asymptotic or asymptotic and if they are, with respect to which variable.

**Questions:**

-	The proposed method only involves an adversarial training of the second layer. This has the benefit of achieving a robust training without requiring more samples or neurons than standard training. In addition, Figure 1 suggests that the convergence is faster compared to a full adversarial training. However, in the ReLu case (left panel), it seems that the robust test risk in the case of the full AD training is still decreasing. Given this observation and in the same setting as the paper, is there a possibility that robustly training both layers could achieve the same guarantees (sample complexity, number of neurons required) as standard training?

-	In line 187, it is mentioned that the parameters $a, b$ are used to approximate $h$. This corresponds to roughly $N$ parameters to approximate the $k$-dimensional function $h$: can we think of the power $k$ in the bound for $N$ in Theorems 4 and 5 as a form of curse of dimensionality associated with this approximation task? If so, is there an intuitive reason why those exponents do not appear in Theorems 6 and 7 ?

-	I am not sure about the assumptions of Proposition 8: In [Lee et. Al., 2024] Theorem 2 requires that the activation is also a polynomial with a higher degree than the link function. Is this additional assumption required for Proposition 8? Likewise, in Corollaries 9 and 11, the mention “as in Assumption 4” may be confusing: do you also make the assumptions regarding the activation?

More precise questions and remarks:

-	In line 215, it is not clear why the expectation of the norm of $x$ is proportional to the square root of the dimension.
-	Likewise, in line 218, the argument for choosing $\varepsilon$ of order 1 can be made clearer (is this because the scalar dot between $u$ and $x$ is also of order 1?).
-	In and before Definition 2, it is not mentioned that the columns of the weight matrix should be of unit norm. Without this detail, it seems that the definition of the DFL do not make much sense.
-	In Definition 3, the last sentence is not very understandable and is not explained afterwards.
-	In Theorems 4 and 6, it would be nice for completeness to precise that the absolute constant c is positive.
-	Since the results of section 4.2 do not cover the case of general activation, the discussion spanning from line 405 to 423 could be shortened.


[Lee et. Al., 2024]: Jason D. Lee, Kazusato Oko, Taiji Suzuki, and Denny Wu. Neural network learns low-dimensional polynomials with sgd near the information-theoretic limit.

---

> ### Author Response · Authors · 2024-11-23
>
> We thank the reviewer for their careful assessment and valuable feedback, and address their comments and questions below.
>
> ---
> **Technical challenges in considering activations beyond ReLU and polynomial**:
>
> We refer to the general response above for a discussion on these challenges.
>
> ---
> **Restructuring the proof of Theorem 1 and including proof outline for other results**:
>
> We thank the reviewer for this suggestion, and will restructure the proof of Theorem 1 and add a high-level overview of the proof technique of other results in the final version.
>
> ---
> **The dependence of the optimal adversarial risk on dimension**:
>
> Assuming the zero predictor is in $\mathcal{F}$, one can notice that $\mathrm{AR}^* \leq \mathbb{E}[y^2]$ which is independent of dimension. Moreover, even in an unnatural setting where $\mathrm{AR}^*$ grows with dimension, our final bounds will be dimension-independent, since in that case $\tilde{\epsilon} = \epsilon$ in all theorem statements.
>
> ---
> **Lack of definition for Landau notation**:
>
> We thank the reviewer for expressing their concern about the notation, in the final version, we will provide further clarification in the notation section of the manuscript. All our bounds are non-asymptotic, and the Landau notation only hides absolute constants (or polylogairthmic terms when used with tilde). We also assume certain quantities such as $k$ are absolute constants in this paper, and we mention such assumptions when introducing those quantities.
>
> ---
> **Guarantees from robust training**:
>
> Currently there are no theoretical guarantees for the representation learned by adversarial training on the first layer in our setting. We believe investigating the sample and computational complexity arising from this adversarial training is an interesting open direction.
>
> ---
> **Curse of dimensionality in $k$**:
>
> The reviewer’s interpretation of the exponential dependence on $k$ as a curse of dimensionality for approximating functions in a $k$-dimensional subspace is indeed correct. The reason this dependence does not show up when considering polynomials is that the space of polynomials of bounded degree is a much smaller space. For example, the space of $k$-index polynomials of degree $p$ is a finite-dimensional space with dimension $k^p$ which only scales polynomially with $k$. The space of all (pseudo-Lipschitz) $k$-index functions is infinite-dimensional, and approximating it with a finite-dimensional space using an $\epsilon$-covering requires $(1/\epsilon)^\mathcal{O(k)}$ many parameters, exponential in $k$.
>
> ---
> **The choice of activation from Lee et al., 2024**:
>
> We thank the reviewer for pointing out this potential confusion. The algorithm in Proposition 8 makes use of a special polynomial activation with a sufficiently large degree. In the revised version, we have provided further clarification in Corollary 9 and 11 in terms of the assumption on activation.
>
> ---
> **Norm of $x$ and scale of $\varepsilon$**:
>
> When considering weights on $\mathbb{S}^{d-1}$, the natural scaling of $\Vert x \Vert$ is $\sqrt{d}$ since it leads to $\langle w, x\rangle$ being of order 1. This is indeed the same reason why $\varepsilon$ is assumed to be of order 1. We will further clarify this in the final version of the manuscript.
>
> ---
> **Further recommendations for clarity**:
>
> We thank the reviewer for their recommendations to improve clarity. We will incorporate them in the final version. In particular, Definitions 2 and 3 indeed assume the rows to be unit-norm.
>
>
> ---
> We would be happy to clarify any concerns or answer any questions that may come up during the discussion period.

---

> > ### Comment · Reviewer_vLR3 · 2024-11-25
> >
> > Dear authors, thank you for answering the questions. On my side, everything is understood and as long as the main text is made clearer I do not have any further remarks. I have raised my original rating.

---

### Official Review · Reviewer_1ZeF · 2024-11-04

**Soundness:** 3
**Presentation:** 3
**Contribution:** 3
**Rating:** 6
**Confidence:** 3

**Summary:**

This paper addresses the challenge of learning adversarially robust models using 2-layer neural networks where the targets are a function of a latent low-dimensional projection of the inputs. Throughout the paper, they consider L2-constrained adversarial perturbations. They first prove that the optimal features in terms of adversarial robustness are in fact the true latent features of the data model. Then building off of prior work in 2-layer network feature learning, they assume access to a feature-learning oracle and analyze algorithms which query the oracle for the first-layer features and then adversarially train over the second layer’s parameters. For different function classes, they provide bounds for the number of required samples for adversarial training, neurons, and feature learning oracle error in order to achieve optimal robustness up to epsilon error with high probability. Through brief experiments, they validate their theoretical results and show that standard training followed by training just the last layer using the adversarial objective performs much better than full adversarial training.

**Strengths:**

The paper’s first contribution is that the ground truth latent features are in fact the optimal features in terms of adversarial robustness. This is not largely surprising, but it plays an important role in informing the kinds of algorithms one should use to achieve an adversarially robust model. They then show how to characterize the sample complexity and model complexity for just adversarially training the second layer for a broad class of target functions when given access to a feature learning oracle. Importantly, these bounds are independent of the ambient dimension as the hardness only depends on the effective dimension of the problem. These results are an interesting extension of previous works in feature learning for 2-layer neural networks which do not consider adversarial robustness.

**Weaknesses:**

The major weakness is the lack of extensive experiments. I think that the first row of figure one is not a fair comparison as the paper graphs a model trained completely using adversarial training techniques to just training the last layer using adversarial techniques while initializing the first layer to the ground truth representation. In the second row,  all models are forced to learn representations which is a more realistic scenario. However, based on the high-level result that adversarial training is no harder than standard training and that the optimal adversarially robust features are just the ground truth features, it would be interesting to see how different combinations of standard vs. adversarial training on different network layers for real-world data affects performance.

Also, the statement of Theorem 1 is confusing. First of all it is not clear that the multi-index data model is still assumed. Also, the phrase “with equality when f^* in F” is confusing since f* in F doesn’t imply equality and that part is obvious, so stating it in this way is confusing for no reason. Also, the statement “Further, h is represented as E[]” is also confusing because it seems to mean that any h that satisfies the theorem must be represented in that way which is not true. The theorem should be restated as something like “Assume multi-index data model as explained above, Assumption 1, and (2.2) admits a min. Then there exists a function f* = h(Ux) where h = E[f] for some f in F where AR(f*) leq AR*.”

**Questions:**

Can you discuss how this result and analysis techniques connect to other recent results in how shallow networks learn representations, such as (Collins et. al., Provable Multi-task Representation Learning by 2-Layer ReLU Neural Networks, ICML 2024) and (Wang et al, Learning Hierarchical Polynomials with Three-Layer Neural Networks, ICLR 2024) ?

Is the first inequality of the proof of Theorem 1 actually an equality?

In the experiment section, how did you select the “direction u”? Is this a vector in R^d? I do not see where u is defined.

Can you discuss what the actionable insights are in terms of training robust models? It would be interesting to see more practical experiments of how standard feature learning followed by running adversarial training on the last layers of a network can result in a full model which is adversarially robust.

---

> ### Author Response · Authors · 2024-11-23
>
> We thank the reviewer for their careful assessment and valuable feedback, and address their comments and questions below.
>
> ---
> **Lack of extensive experiments**:
>
> We thank the reviewer for this suggestion. We have added a simple experiment on MNIST in **Appendix E** to demonstrate that the intuitions from our theoretical results carry over to settings with neural networks with multiple layers, cross entropy loss, and $\ell_\infty$ norm for attack. We must highlight however that this experiment is not meant to perform an extensive study. In particular, we retrain all layers from standard training, as in this case it is not entirely clear which layers are performing representation learning. We believe that in practice, retraining all layers will be useful nevertheless, since the algorithm will not be able to perfectly learn the optimal representation during standard training due to finite samples and compute. It will be an interesting direction for future work to perform an extensive empirical study over modern architectures to consider the effect of fixing certain layers from standard training and performing adversarial training on the rest.
>
> ---
> **Confusion in the statement of Theorem 1**:
>
> Thank you for your feedback. We will accordingly rewrite the statement of Theorem 1 in the final version to make it more readable.
>
> ---
> **Discussing relevant works**:
>
> Thank you for pointing out these relevant works. Collins et al., 2024 consider learning multiple tasks of the multi-index model form using the feature learning framework. Wang et al., 2024 consider learning a generalization of the single-index model where instead of having a linear feature, we have a degree $k$ polynomial of the input as the feature. We believe that intuitions from our work can extend to these settings as well, i.e. one can combine standard representation learning with robust fine-tuning. We have added a discussion of these works in the revised version of our manuscript.
>
> ---
> **Is the first inequality in the proof of Theorem 1 an equality?**:
>
> The inequality can be strict because we are moving the maximum outside of the expectation. Equality would only hold when $f$ is a minimizer of $\mathrm{AR}$ in $\mathcal{F}$, and additionally $h(U\cdot) \in \mathcal{F}$.
>
> ---
> **How did you select the direction of $u$ in the experiments?**:
>
> In this section, we look at single-index models of the form $x \mapsto g(\langle u, x\rangle)$. We randomly sample $u$ from the unit sphere in $\mathbb{S}^{d-1}$. Since we use the same seed in all experiments, this means $u$ is the same across experiments. We will make this more explicit in the final version.
>
> ---
> **Actionable insights from this work**:
>
> Indeed, while our focus is on the theoretical part, one actionable insight from our work is that standard training yields a better initialization than the usual random initialization for adversarial robustness. Furthermore, in certain settings, fine-tuning the final layers of a network might be enough to achieve adversarial robustness, although this still requires a full backpropagation over the network to generate adversarial examples. We will highlight these insights in the final version of our manuscript.
>
> ---
> We would be happy to clarify any concerns or answer any questions that may come up during the discussion period.

---

> > ### Comment · Reviewer_1ZeF · 2024-11-26
> >
> > Thank you for your response. I have no further questions.

---

### Official Review · Reviewer_weXL · 2024-11-11

**Soundness:** 3
**Presentation:** 2
**Contribution:** 3
**Rating:** 6
**Confidence:** 3

**Summary:**

The paper analyzses adversarially robust learning for two-layer neural networks trained on multi-index targets dependent on a low-dimensional subspace. The paper begins by proving that an optimal adversarially-robust predictor depends solely on the target subspace. Subsequently, the paper shows that existing guarantees for recovering the target subspace through SGD can be combined with covering number, rademacher complexity-based generalization and approximation arguments to obtain bounds on the adversarial risk. Crucially, the paper shows that the additional sample-complexity requirements for vanishing adversarial risk are independent of the ambient dimension.

**Strengths:**

- The problem of obtaining theoretical guarantees on adversarial risk and the dependence on the structure of the data is well-motivated of interest to the community.
- The direction of studying the interplay between feature learning and adversarial robustness appears to be novel.
- The paper is well written and describes the setup, literature well.
- The proofs are easy to follow.

**Weaknesses:**

My main criticism of the paper is that the extent of the theoretical novelty in the analysis and unexpectedness of the result is unclear in the paper's presentation. The paper would improve by emphasizing the central difficulties in combining the finite-dimensional guarantees with feature learning results. Specifically, it would be useful to know what parts of the analysis would change if one were directly studying the setup of random features on finite dimensions. It appears that the role of the low-dimensionality of the target subspace and feature learning is primarily in obtaining a second layer with dimension-independent norm ($r_a$ in Algorithm 1) that fits the target allowing for the Rademacher-complexity argument to go through. This aspect of the analysis is however, the same as generalization bounds in existing works on feature learning such as Damian et al. 2022.

**Questions:**

- What are the assumptions on the class $\mathcal{F}$ in Theorem 1?
- In the proof of Theorem 1, is $h(x)$ assumed to be in $\mathcal{F}$?
- Does definition 2 (DFL) implicitly assume constraints on the magnitudes of $w_i$? Otherwise, DFL is trivially satisfied by scaling up the weights. How is this scaling issue consistent with Theorem 4?

---

> ### Author Response · Authors · 2024-11-23
>
> We thank the reviewer for their careful assessment and valuable feedback, and address their comments and questions below.
>
> ---
> **Technical challenges and novelties**:
>
> *Unexpectedness of the result*: One reason for the unexpectedness of the result of Theorem 1 is that we simply do not *always* have $U$ as an optimal low-dimensional projection of the input, unless we take into account additional assumptions. Some examples where $U$ is not the optimal direction are given in the general response above, which are due to using $\ell_\infty$ norm or having certain anisotropic covariances. This is in contrast with the standard setting, where $U$ *always* provides an optimal projection *regardless of additional assumptions*. This highlights the importance of the assumptions and the proof of Theorem 1, and that one can not always take the statement of Theorem 1 for granted without taking into account the assumptions. We will add this discussion to the final version of the manuscript.
>
> *Technical challenges*: We agree with the reviewer that once the rather unexpected result of Theorem 1 is proved, it is expected that we can end up with dimension-independent bounds for sample and computational complexity. However, we would like to highlight some other aspects of our analysis here that differs from that of prior works:
> * We introduce DFL and SFL as a formalism for feature learning that unifies the previous approaches to establishing alignment with target directions and performing dimension reduction. This might be of interest for future works since one only needs to prove that the DFL or SFL assumption holds in some setting, and then they can readily use our approximation and generalization guarantees to end up with learnability results.
> * The Rademacher complexity analyses for two-layer neural networks are typically performed using Talagrand’s contraction lemma, see e.g. [1, Proposition 7]. This approach is not compatible with the adversarial loss, and we instead take the covering number approach with Maurey’s sparsification lemma as used in [2] for our Rademacher complexity analysis.
> * Our approximation analysis goes beyond that of the recent literature such as Damian et al., 2022 in the sense that we cover pseudo-Lipschitz functions with a universal approximation argument inspired from [1]. However, the universal approximation argument of [1] requires trained biases, whereas we prove our results under random biases, which helps keep the training of the second layer a convex problem.\
> We will further discuss these differences in the final version of our manuscript.
>
> ---
> **Assumptions on $\mathcal{F}$ in Theorem 1**:
>
> Theorem 1 does not require any particular assumptions on $\mathcal{F}$, beyond the ones that are required for the mathematical objects in the theorem to make sense such as measurability and finite expectation.
>
> ---
> **Is $h$ assumed to be in $\mathcal{F}$**:
>
> $h(U\cdot)$ does not need to be in $\mathcal{F}$. However, if $h(U\cdot)$ is additionally in $\mathcal{F}$, such as in the examples, then $\mathrm{AR}(h(U \cdot)) = \mathrm{AR}^*$.
>
> ---
> **Constraint on the norm of $w_i$ in DFL**:
>
> Yes, DFL and SFL both assume the weights $w_i$ to be on the unit sphere $\mathbb{S}^{d-1}$, which is also the setting in the rest of the paper, including Theorem 4. We thank the reviewer for pointing out that we had missed to mention this, and have fixed it in the revised version.
>
> ---
> We would be happy to clarify any concerns or answer any questions that may come up during the discussion period.
>
> References:
>
> [1] F. Bach. “Breaking the Curse of Dimensionality with Convex Neural Networks.” JMLR 2017.
>
> [2] T. Zhang. “Covering number bounds of certain regularized linear function classes.” JMLR 2002.

---

> > ### Comment · Reviewer_weXL · 2024-11-27
> >
> > I thank the authors for the clarifications. Including a discussion of the $\infty$-norm setting would certainly strengthen the paper. I've decided to maintain my score due to the technical novelty concerns.

---

### Author Response · Authors · 2024-11-23
**General Response**

We thank all reviewers for carefully reading our manuscript and providing a thoughtful evaluation. Here, we address some of the common questions raised in the reviews. The specific questions are answered separately for each reviewer.

---
**Changing the attack norm from $\ell_2$ to $\ell_\infty$**:

By considering a simple linear setting where $g(x) = \langle u, x\rangle$ and $\mathcal{F}$ is the class of linear predictors, we observe that under the $\ell_\infty$ attack norm, the optimal $w$ roughly follows $w \approx u - \epsilon \operatorname{sign}(u)$. This highlights the importance of our $\ell_2$ norm assumption, and shows that considering other attack norms is an interesting open problem, where other approaches might yield optimal results.

---
**Considering anisotropic covariance and going beyond Assumption 1**:

First, we would like to highlight that Assumption 1 can cover certain anisotropic cases. Without loss of generality, suppose $U$ refers to the first $k$ coordinates in $x$. Then, for Gaussian $x$, Assumption 1 requires the covariance matrix to be of the form $\Sigma = \begin{pmatrix} \Sigma_1 & 0\\\\ 0 & \Sigma_2\end{pmatrix}$, where $\Sigma_1 \in \mathbb{R}^{k \times k}$ and $\Sigma_2 \in \mathbb{R}^{(d-k) \times (d-k)}$ can be arbitrary PSD matrices, therefore covering anisotropic matrices.

Second, we remark that Assumption 1 is *necessary* for Theorem 1. Once again, we consider the linear setting where $g(x) = \langle u, x\rangle$ and $\mathcal{F}$ is the set of linear predictors. Suppose $x \sim \mathcal{N}(0,\Sigma)$. Then, the optimal weight is given by $w = ((1 + a\varepsilon) \Sigma + b\varepsilon I_d)^{-1} \Sigma u$ for some $a,b > 0$. This does not match the direction of $u$ unless $\Sigma$ satisfies the structure presented above.

We will add these discussions in the revised version to provide further intuitions for the readers.

---
**Technical challenges in considering more general activations compared to ReLU and polynomial activations**:

The main challenge is to prove quantitative approximation guarantees. While in principle any non-polynomial activation function is sufficient for universal approximation, and e.g. must be able to compete with the class of pseudo-Lipschitz predictors, quantitative guarantees for universal approximation are only well-known for few activations, such as homogeneous activations as in [1] which includes ReLU, or polynomial activations (which can learn polynomial functions).

---
References:

[1] F. Bach. “Breaking the Curse of Dimensionality with Convex Neural Networks.” JMLR 2017.

---

### Meta-Review · Area_Chair_VoTe · 2024-12-21

**Metareview:**

**(a) Summarize the scientific claims and findings of the paper based on your own reading and characterizations from the reviewers.**
The paper explores adversarially robust learning in multi-index models for two-layer neural networks. The authors demonstrate that under specific assumptions, the Bayes-optimal adversarial risk can be achieved with adversarial training focused on the second layer while leveraging standard training for the first layer. The main theoretical finding is that adversarial training does not significantly increase the sample complexity compared to standard training. The work also introduces robust feature learning formulations and supports its claims with numerical experiments, highlighting scenarios where adversarial training is as efficient as standard training.

**(b) What are the strengths of the paper?**
- Provides a rigorous theoretical foundation linking adversarial robustness to feature learning in multi-index models.
- Demonstrates that robust training can be accomplished without a significant sample complexity penalty.
- Offers a novel approach that combines standard feature learning with adversarially fine-tuning the second layer.
- Clear explanations of assumptions and limitations, accompanied by detailed proofs.
- Numerical experiments validate theoretical results, adding practical relevance.

**(c) What are the weaknesses of the paper? What might be missing in the submission?**
- Limited empirical results and lack of exploration of practical scenarios beyond theoretical assumptions.
- Focuses exclusively on \( \ell_2 \)-norm adversarial perturbations without extending to other norms like \( \ell_\infty \), which are more commonly used in practice.
- Relies on assumptions such as statistical independence of features (Assumption 1), which may not always hold in realistic settings.
- Limited discussion on computational complexity or real-world applicability of the proposed methods.

**(d) Provide the most important reasons for your decision to accept/reject.**
I recommend acceptance due to the paper’s strong theoretical contributions to understanding adversarial robustness in multi-index models and its potential to inform future work in the domain. While there are gaps in practical applicability and empirical analysis, the paper opens up promising avenues for further research and sets a robust theoretical foundation.

**Additional Comments On Reviewer Discussion:**

*Please summarize the discussion and changes during the rebuttal period. What were the points raised by the reviewers? How were each of these points addressed by the authors? How did you weigh in each point in your final decision?*

1. **Clarifications on theoretical assumptions and results**
   - Reviewers requested additional explanations on Assumption 1 and its validity in anisotropic settings. The authors clarified these points, highlighting examples where the assumption holds and identifying directions for future work to relax this assumption.
   - The necessity of squared loss and lack of exploration with logistic loss were discussed. The authors addressed this with additional experiments showing similar trends for cross-entropy loss.

2. **Empirical validation and practical limitations**
   - Concerns about the limited scope of empirical results were raised. The authors provided a small-scale MNIST experiment and emphasized the theoretical focus of the work while acknowledging the need for broader experiments in future research.

3. **Presentation and structure**
   - Several reviewers noted that combining similar theorems and reorganizing the paper could improve readability. The authors agreed to restructure the main text in the final version for clarity.

4. **Relevance to adversarial robustness in practice**
   - Questions about extending the framework to \( \ell_\infty \)-norm perturbations and addressing computational complexity were discussed. While the authors justified their focus on \( \ell_2 \)-norm for theoretical simplicity, they acknowledged the importance of exploring other norms in future work.

**Final Decision**
The authors have satisfactorily addressed most reviewer concerns, reinforcing the theoretical contributions of the work. The decision to accept is based on the novelty and rigor of the results, as well as the potential impact on future adversarial robustness research.

---

### Decision · Program_Chairs · 2025-01-22

Accept (Poster)